# DoRAN: Stabilizing Weight-Decomposed Low-Rank Adaptation via Noise Injection and Auxiliary Networks

## Abstract

Parameter-efficient fine-tuning (PEFT) methods have become the standard paradigm for adapting large-scale models. Among these techniques, Weight-Decomposed Low-Rank Adaptation (DoRA) has been shown to improve both the learning capacity and training stability of the vanilla Low-Rank Adaptation (LoRA) method by explicitly decomposing pre-trained weights into magnitude and directional components. In this work, we propose **DoRAN**, a new variant of DoRA designed to further stabilize training and boost the sample efficiency of DoRA. Our approach includes two key stages: (i) injecting noise into the denominator of DoRA's weight decomposition, which serves as an adaptive regularizer to mitigate instabilities; and (ii) replacing static low-rank matrices with auxiliary networks that generate them dynamically, enabling parameter coupling across layers and yielding better sample efficiency in both theory and practice. Comprehensive experiments on vision and language benchmarks show that DoRAN consistently outperforms LoRA, DoRA, and other PEFT baselines. These results underscore the effectiveness of combining stabilization through noise-based regularization with network-based parameter generation, offering a promising direction for robust and efficient fine-tuning of foundation models.

## 1 Introduction

The rapid growth of large-scale pre-trained models has reshaped modern machine learning, enabling state-of-the-art performance across a wide range of vision and language tasks (Bommasani et al., 2022; Wei et al., 2022). The prohibitive cost of fully fine-tuning these models has spurred significant interest in parameter-efficient fine-tuning (PEFT) methods, which achieve strong downstream performance while modifying only a small fraction of parameters. A variety of PEFT methods have been developed to adapt large-scale pre-trained models without incurring the cost of full fine-tuning. Early approaches, such as adapter layers, insert small trainable modules between frozen transformer blocks, while prefix- and prompt-tuning prepend learnable vectors to the model's input space. More recently, Low-Rank Adaptation (LoRA) has become a foundational technique for efficient fine-tuning of large-scale models across multiple AI domains. Originally proposed by Hu et al. (2022), LoRA introduces low-rank trainable adapters into existing weight matrices, reducing computational overhead while maintaining high task performance. Beyond its success in natural language processing (NLP), LoRA has demonstrated strong adaptability in computer vision (Yang et al., 2024), speech processing, and reinforcement learning (Huan & Shun, 2025). LoRA also plays a pivotal role in federated and distributed learning, enabling efficient personalization while preserving data privacy (Sun et al., 2025; Xu et al., 2025).

Despite its effectiveness in providing a flexible low-rank adaptation, LoRA's rigid low-rank formulation can limit both representational capacity and training stability. Several extensions have sought to address these shortcomings. Notably, DoRA (Liu et al., 2024) introduces a directional-based normalization approach: instead of learning flexible low-rank updates directly, it decomposes pre-trained weights into magnitude and directional components. This design improves representational power relative to vanilla LoRA and mitigates some instability.

While DoRA represents a significant advance, it still inherits two challenges. First, DoRA relies on strict normalization, which makes it sensitive to optimization instabilities: when the adapted weight

norm approaches zero, gradients can explode, destabilizing training. Second, DoRA still employs static low-rank matrices that are optimized independently at each layer, underutilizing cross-layer correlations and often leading to redundant or poorly coordinated updates. These challenges motivate the following central question of our work:

> *(Q) Can we unify the strengths of **direction-based normalization** and **flexible low-rank adaptation** into a single framework that is both stable and expressive?*

To address this question, we propose Weight-**D**ecomposed L**o**w-**R**ank **A**daptation with **N**oise injection (DoRAN), a PEFT approach that simultaneously **stabilizes optimization** and **enriches low-rank representations**. DoRAN introduces two key components. **(i) Stabilization via learnable noise:** We augment DoRA's normalization with a learnable offset in the denominator, which functions as an adaptive regularizer. This simple yet powerful modification eliminates singularities at small norms, ensures gradients remain well-conditioned, and allows the model to smoothly interpolate between two regimes: purely directional updates (as in DoRA) and linear scaling (as in LoRA). In this way, DoRAN adaptively preserves magnitude information of the gradient signal instead of discarding it entirely like DoRA, as we will reveal in Section 3.2, leading to more expressive and stable updates. **(ii) Dynamic generation of low-rank factors:** Instead of directly learning a separate low-rank adapter at each layer, we dynamically generate the adapters through a hypernetwork with a shared backbone and head-specific output heads between query and value matrices for each layer. This coupling encourages information sharing between query and value across layers, enforces structural consistency across the model, and improves sample efficiency by leveraging shared latent structure. This design is motivated by recent studies which demonstrated that hypernetworks enhances generalization and stability by enabling adaptive weight generation conditioned on task representations (Ha et al., 2017). Besides, this design also provides a richer inductive bias that regularizes the adaptation space and mitigates redundancy across layers.

Crucially, as we will show in our experiments, both components contribute significantly to the performance improvements. Moreover, these two components are synergistic: the learnable noise term ensures that gradient signals remain stable and informative, while the hypernetwork ensures that these stabilized signals are used to generate structured low-rank updates. Together, they provide a unified framework that bridges the gap between direction-based normalization and flexible low-rank adaptation, offering both robustness and expressiveness for fine-tuning large-scale models.

We validate DoRAN through extensive experiments on vision (VTAB-1K, FGVC) and language (commonsense reasoning with LLaMA-7B/13B) benchmarks. DoRAN consistently outperforms LoRA, DoRA, and other PEFT baselines, while introducing negligible additional overhead. We also provide theoretical and empirical evidence that DoRAN significantly improves sample efficiency by leveraging shared structure between query and value matrices through applying specific-hypernetworks across layers.

> **Contributions.** In summary, our contributions are threefold:
>
> - We propose a stabilized weight decomposition with learnable noise that removes the singularity of DoRA and adaptively preserves magnitude information, and a hypernetwork-based reparameterization of low-rank factors, enabling parameter coupling between query and value matrices across layers and improved sample efficiency.
> - We provide a theoretical analysis showing that this design significantly enhances the sample efficiency from an *exponential order* to a *polynomial order*.
> - We demonstrate through various experimental settings that DoRAN achieves state-of-the-art parameter efficiency, consistently outperforming LoRA, DoRA, and other PEFT baselines on both vision and language tasks.

Together, we show that combining noise-based stabilization with hypernetwork-driven parameter generation offers a robust and generalizable path forward for fine-tuning foundation models.

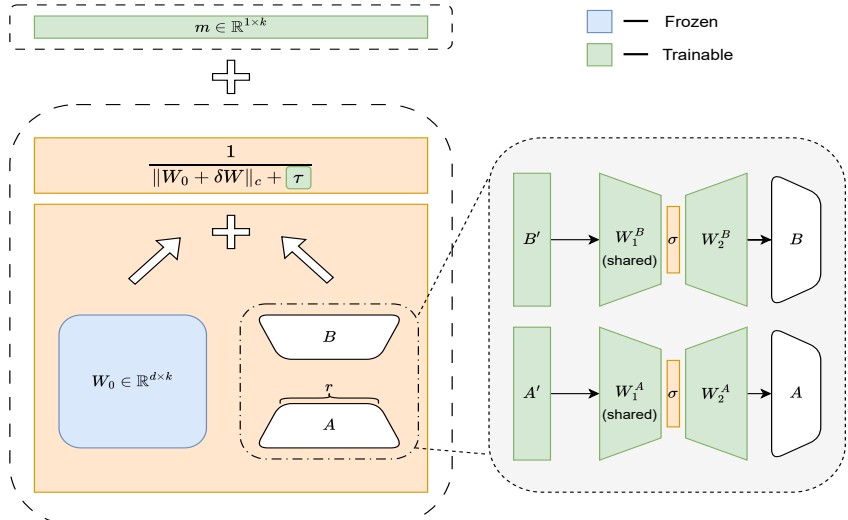

Figure 1: Illustration of DoRAN. The matrices $W_1^A$ and $W_1^B$ are shared across query and value projection layers, as well as across all attention heads within the same Transformer block. Each layer maintains its own independent hypernetwork.

## 2 PRELIMINARIES

**Notation.** We denote $[n] = \{1, 2, \ldots, n\}$ for an integer $n$. For a vector $u \in \mathbb{R}^d$, we will use both notations $u = (u^{(1)}, u^{(2)}, \ldots, u^{(d)})$ and $u = (u_1, u_2, \ldots, u_d)$ interchangeably. Given a multi-index vector $\alpha = (\alpha_1, \alpha_2, \ldots, \alpha_d) \in \mathbb{N}^d$, we write $u^\alpha = u_1^{\alpha_1} u_2^{\alpha_2} \cdots u_d^{\alpha_d}$, $|\alpha| = \alpha_1 + \alpha_2 + \cdots + \alpha_d$, and $\alpha! = \alpha_1! \alpha_2! \cdots \alpha_d!$. $\|u\|$ denotes the Euclidean norm of $u$, while $|S|$ represents the cardinality of a set $S$. For two positive sequences $(a_n)_{n \geq 1}$ and $(b_n)_{n \geq 1}$, we write $a_n = \mathcal{O}(b_n)$ or $a_n \lesssim b_n$ if there exists a constant $C > 0$ such that $a_n \leq C b_n$ for all $n$. We denote $a_n = \mathcal{O}_P(b_n)$ if $a_n/b_n$ is bounded in probability. Next, we use the notation $a_n = \widetilde{\mathcal{O}}_P(b_n)$ when $a_n = \mathcal{O}_P(b_n \log^c(b_n))$ for some $c > 0$. Finally, we define the inner product of two matrices as $\langle \boldsymbol{A}, \boldsymbol{B} \rangle = \text{Trace}(\boldsymbol{A}^\top \boldsymbol{B})$.

**MSA: Multi-head Self-attention.** Self-attention is the central mechanism of the Transformer architecture (Vaswani et al., 2017; Dosovitskiy, 2021a). Given an input sequence $\boldsymbol{X} \in \mathbb{R}^{n \times d}$ with $n$ tokens of $d$ dimension, the model projects $\boldsymbol{X}$ into query, key, and value matrices via learned linear transformations $\boldsymbol{Q} = \boldsymbol{X} \boldsymbol{W}_Q, \boldsymbol{K} = \boldsymbol{X} \boldsymbol{W}_K$ and $\boldsymbol{V} = \boldsymbol{X} \boldsymbol{W}_V$ where $\boldsymbol{W}_Q, \boldsymbol{W}_k, \boldsymbol{W}_V \in \mathbb{R}^{d \times d_h}$. The attention weight is defined as

$$\text{Attention}(\boldsymbol{Q}, \boldsymbol{K}, \boldsymbol{V}) = \text{softmax}(\frac{\boldsymbol{Q} \boldsymbol{K}^\top}{\sqrt{d_k}}) \boldsymbol{V}.$$

Multi-head Self-attention (MSA) extends this mechanism by introducing multiple parallel attention operations or "heads", where each attention head processes the input sequence from a different representational subspace. Formally, for $h$ heads, the output of an MSA layer is calculated as

$$\text{MSA}(\boldsymbol{X}_Q, \boldsymbol{X}_K, \boldsymbol{X}_V) = \text{Concat}(\text{head}_1, \cdots, \text{head}_h) \boldsymbol{W}_O,$$

$$\text{head}_i = \text{Attention}(\boldsymbol{X} \boldsymbol{W}_Q^{(i)}, \boldsymbol{X} \boldsymbol{W}_K^{(i)}, \boldsymbol{X} \boldsymbol{W}_V^{(i)}).$$

**LoRA and DoRA.** Low-rank Adaptation (LoRA) (Hu et al., 2022) is one of the most widely used PEFT approaches. The key idea is to decompose the weight updates into low-rank matrices and fine-tune these low-rank matrices while keeping the original weights frozen. Formally, given a pre-trained weight matrix $\boldsymbol{W} \in \mathbb{R}^{d \times k}$, LoRA freezes $\boldsymbol{W}$ and models the weight update $\Delta \boldsymbol{W} \in \mathbb{R}^{d \times k}$ as two low-rank matrices $\boldsymbol{B}$ and $\boldsymbol{A}$:

$$\boldsymbol{W} = \boldsymbol{W}_0 + \Delta \boldsymbol{W} = \boldsymbol{W}_0 + \boldsymbol{B} \boldsymbol{A}$$

where $\boldsymbol{B} \in \mathbb{R}^{d \times r}$ and $\boldsymbol{A} \in \mathbb{R}^{r \times k}$, with rank $r \ll \min(d, k)$. On performing weight decomposition on fine-tuned weight matrices, Liu et al. (2024) found that LoRA and full fine-tuning show different learning patterns. To resolve this difference, weight-decomposed low-rank adaptation (DoRA)

(Liu et al., 2024) explicitly decomposes the pre-trained weight into its magnitude and directional component and fine-tunes both components. Then, DoRA performs updates as follows:

$$\boldsymbol{W} = m \frac{\boldsymbol{V} + \Delta \boldsymbol{V}}{\|\boldsymbol{V} + \Delta \boldsymbol{V}\|_c} = m \frac{\boldsymbol{W}_0 + \boldsymbol{BA}}{\|\boldsymbol{W}_0 + \boldsymbol{BA}\|_c},$$

where $\boldsymbol{B}$, $\boldsymbol{A}$ are trainable low-rank matrices, $m \in \mathbb{R}^{1 \times k}$ is a learnable vector, and $\| \cdot \|_c$ is the vector-wise norm of a matrix across each column. In this work, we aim to improve DoRA by introducing noise-stabilized normalization to prevent instability and by utilizing auxiliary networks to generate low-rank updates, thereby enhancing both robustness and parameter efficiency.

## 3 METHODOLOGY

**Motivations.** DoRA enhances stability by decomposing pre-trained weights into magnitude and directional components. Nevertheless, adaptation can still suffer from instability. In particular, when the normalization denominator becomes excessively small, DoRA may encounter gradient explosions, as we will discuss in Section 3.2. Moreover, standard DoRA employs static, distinct low-rank matrices between query and value. When each adapter $(\boldsymbol{A}, \boldsymbol{B})$ is optimized independently, they risk diverging toward inconsistent or redundant solutions. To address these limitations, we propose Stabilized Weight-**D**ecomposed **L**ow-**R**ank **A**daptation via **N**oise Injection and Auxiliary Networks (DoRAN), which integrates two complementary components. **First**, DoRAN introduces a learnable stabilization noise term in the normalization step, which provides more stable and expressive updates. **Second**, instead of directly optimizing the low-rank matrices, DoRAN employs hypernetworks to dynamically generate low-rank matrices.

### 3.1 DORAN: DORA WITH NOISE INJECTION AND AUXILIARY NETWORKS

**Noise Injection.** We first enhance stability by introducing a trainable stabilization noise term $\tau$:

$$\boldsymbol{W} = m \frac{\boldsymbol{W}_0 + \boldsymbol{BA}}{\|\boldsymbol{W}_0 + \boldsymbol{BA}\|_c + \tau},$$

where $\tau \in \mathbb{R}^+$ acts as an adaptive regularizer, ensuring stable normalization. We design $\tau$ as a *learnable scalar* rather than a fixed constant for a key reasons: As we will reveal in the gradient analysis (Section 3.2), the value of $\tau$ determines the balance between two adaptation regimes: when $\tau \to 0$, the update reduces to DoRA-style direction learning, while large $\tau$ yields LoRA-like scaling. By making $\tau$ learnable, the model can automatically interpolate between these extremes, adaptively finding the most effective balance between directional learning and stable norm control.

**Auxiliary Networks.** In addition to the term $\tau$, DoRAN further enhances flexibility by implementing the low-rank matrices $\boldsymbol{A}$ and $\boldsymbol{B}$ through auxiliary networks rather than optimizing them directly:

$$\boldsymbol{B} = g_{\boldsymbol{B}}(\boldsymbol{B}') := \boldsymbol{W}_2^{\boldsymbol{B}} \sigma(\boldsymbol{W}_1^{\boldsymbol{B}} \boldsymbol{B}'); \qquad \boldsymbol{A} = g_{\boldsymbol{A}}(\boldsymbol{A}') := \boldsymbol{W}_2^{\boldsymbol{A}} \sigma(\boldsymbol{W}_1^{\boldsymbol{A}} \boldsymbol{A}'),$$

where $\boldsymbol{A}'$, $\boldsymbol{B}'$ are learnable embedding, and $g_{\boldsymbol{B}}, g_{\boldsymbol{A}}$ are two-layer feedforward networks with the activation $\sigma$. In our experiments, the inputs $\boldsymbol{B}'$, $\boldsymbol{A}'$ and the first layers $\boldsymbol{W}_1^{\boldsymbol{A}}, \boldsymbol{W}_1^{\boldsymbol{B}}$ are shared between query and value projections in each layer, while $\boldsymbol{W}_2^{\boldsymbol{A}}, \boldsymbol{W}_2^{\boldsymbol{B}}$ are distinct to emit specific adaptation matrices for the query and value projections. To ensure parameter efficiency and facilitate knowledge sharing across attention heads, this hypernetwork is shared among attention heads in each layer. Bringing these components together, we have the following DoRAN updates, whose demonstration can be found in Figure 1:

$$\boldsymbol{W} = m \frac{\boldsymbol{W}_0 + g_{\boldsymbol{B}}(\boldsymbol{B}') g_{\boldsymbol{A}}(\boldsymbol{A}')}{\|\boldsymbol{W}_0 + g_{\boldsymbol{B}}(\boldsymbol{B}') g_{\boldsymbol{A}}(\boldsymbol{A}')\|_c + \tau}.$$

**Stabilization–Hypernetwork Synergy in DoRAN.** We emphasize that the stabilization noise and the hypernetworks play complementary roles. As we will discuss in Section 3.2, the learnable noise term $\tau$ stabilizes optimization by ensuring that the gradients passed to the low-rank factors remain well-conditioned and preserve both orthogonal and parallel components of the signal. On the other hand, the hypernetwork then ensures that these gradients are not used to produce arbitrary low-rank updates, but rather to generate structured and consistent factors. In particular, the parameter sharing

mechanism acts as an implicit regularizer: rather than producing arbitrary and potentially redundant low-rank solutions, the hypernetwork encourages consistency and reuse of common structure between query and value matrices across layers, which has been shown in prior work to improve generalization of learned weights (Ha et al., 2017; Bertinetto et al., 2016).

## 3.2 GRADIENT ANALYSIS

In this section, we examine the gradient properties of DoRAN to further highlight its theoretical benefits. Let $\boldsymbol{W}' = \boldsymbol{W}_0 + \boldsymbol{B}\boldsymbol{A}$. Let $\mathcal{L}(\boldsymbol{W})$ denote the loss and define $\boldsymbol{G} = \frac{\partial \mathcal{L}}{\partial \boldsymbol{W}}$ as the upstream gradient. By the chain rule, the total gradient with respect to $\boldsymbol{W}'$ can be given by:

$$\frac{\partial \mathcal{L}}{\partial \boldsymbol{W}'} = m\left[\frac{\boldsymbol{G}}{\|\boldsymbol{W}'\|_c + \tau} - \frac{\langle \boldsymbol{G}, \boldsymbol{W}'\rangle}{(\|\boldsymbol{W}'\|_c + \tau)^2}\frac{\boldsymbol{W}'}{\|\boldsymbol{W}'\|_c}\right].$$

Writing $\boldsymbol{G}$ as the sum of its component orthogonal to $\boldsymbol{W}'$ ($\boldsymbol{G}_\perp$) and its projection onto $\boldsymbol{W}'$ ($\mathrm{proj}_{\boldsymbol{W}'}(\boldsymbol{G})$), we obtain the equivalent decomposition:

$$\frac{\partial \mathcal{L}}{\partial \boldsymbol{W}'} = \frac{m}{\|\boldsymbol{W}'\|_c + \tau}\boldsymbol{G}_\perp + \frac{m\tau}{(\|\boldsymbol{W}'\|_c + \tau)^2}\mathrm{proj}_{\boldsymbol{W}'}(\boldsymbol{G}).$$

This formulation reveals that the *orthogonal component* of the gradient, which drives updates in the direction subspace, is scaled by a factor of $m/(\|\boldsymbol{W}'\|_c + \tau)$. On the other hand, the *parallel component* (which controls norm changes) is preserved with a damping factor $\tau/(\|\boldsymbol{W}'\|_c + \tau)$. In other words, the parallel gradient $\mathrm{proj}_{\boldsymbol{W}'}(\boldsymbol{G})$ is not completely discarded (as in DoRA), hence the magnitude information is adaptively preserved. Therefore, in the limit $\tau \to 0$, the parallel component vanishes, recovering the DoRA update rule where only the direction of $\boldsymbol{W}'$ is optimized. Conversely, as $\tau \to \infty$, the update reduces to a rescaled linear adaptation $\boldsymbol{W} \approx (m/\tau)\boldsymbol{W}' = (m/\tau)(\boldsymbol{W}_0 + \boldsymbol{A}\boldsymbol{B})$, which resembles LoRA. The gradients with respect to $m$ and $\tau$ are given by:

$$\frac{\partial \mathcal{L}}{\partial m} = \left\langle \boldsymbol{G}, \frac{\boldsymbol{W}'}{\|\boldsymbol{W}'\|_c + \tau}\right\rangle, \qquad \frac{\partial \mathcal{L}}{\partial \tau} = -m\left\langle \boldsymbol{G}, \frac{\boldsymbol{W}'}{(\|\boldsymbol{W}'\|_c + \tau)^2}\right\rangle.$$

This equation reveals that $m$ governs global scaling, while $\tau$ adaptively interpolates between strict normalization and linear scaling. To sum up, relative to DoRA and LoRA, the adaptive $\tau$ of DoRAN provides the following effects:

- **Stability:** The denominator $\|\boldsymbol{W}'\|_c + \tau$ avoids the singularity at small norms, providing bounded gradients, which leads to more stable updates.

- **Magnitude learning via $\boldsymbol{W}'$:** Unlike DoRA, the parallel component is partially retained, allowing $\boldsymbol{A}, \boldsymbol{B}$ to co-adapt both direction and norm.

- **Continuum of behaviors:** By adjusting $\tau$, the method interpolates smoothly between DoRA ($\tau \to 0$) and LoRA-like scaling ($\tau \gg \|\boldsymbol{W}'\|_c$). Therefore, $\tau$ enables the model to discover the most effective balance between direction learning and stable norm control.

## 4 THEORETICAL ANALYSIS

This section presents the theoretical explanation of the advantages of the reparameterization technique implemented in DoRAN, through its connection to Mixture of Experts (MoE).

**DoRA meets MoE.** Recent works have shown that single-headed self-attention can be reinterpreted as a Mixture of Experts (MoE) model (Truong et al., 2025a), where each head or projection serves as an expert and the attention mechanism acts as the gating function. Building on this view, applying DoRA to an attention head can also be reformulated as an MoE model. Specifically, the attention head before and after applying DoRA can be expressed as follows:

$$\mathrm{head}_{\mathrm{pre}} = \mathrm{softmax}\left(\boldsymbol{X}\boldsymbol{W}_Q\boldsymbol{W}_K^\top\boldsymbol{X}^\top/\sqrt{d_h}\right)\boldsymbol{X}\boldsymbol{W}_V \in \mathbb{R}^{N \times d_h},$$

$$\mathrm{head}_{\mathrm{post}} = \mathrm{softmax}\left(\boldsymbol{X}m_Q\frac{\boldsymbol{W}_Q + \boldsymbol{B}_Q\boldsymbol{A}_Q}{\|\boldsymbol{W}_Q + \boldsymbol{B}_Q\boldsymbol{A}_Q\|\sqrt{d_h}}\boldsymbol{W}_K^\top\boldsymbol{X}^\top\right)\boldsymbol{X}m_V\frac{\boldsymbol{W}_V + \boldsymbol{B}_V\boldsymbol{A}_V}{\|\boldsymbol{W}_V + \boldsymbol{B}_V\boldsymbol{A}_V\|},$$

where the softmax is applied to each row of the matrix inside. Therefore, the $i$-th row of the $\text{head}_{\text{post}}$ can be written as,

$$
\text{head}_{\text{post},i} = \sum_{j=1}^{N} \underbrace{\text{softmax}\left(\left\{\widetilde{\boldsymbol{X}}\boldsymbol{E}_i^\top \frac{m_Q(\boldsymbol{W}_Q + \boldsymbol{B}_Q \boldsymbol{A}_Q)}{\|\boldsymbol{W}_Q + \boldsymbol{B}_Q \boldsymbol{A}_Q\|\sqrt{d_h}}\boldsymbol{W}_K^\top \boldsymbol{E}_k \widetilde{\boldsymbol{X}}\right\}_{k=1}^{N}\right)_j}_{\text{gating score for } j\text{-th expert}}
$$
$$
\underbrace{m_V \frac{(\boldsymbol{W}_V + \boldsymbol{B}_V \boldsymbol{A}_V)^\top}{\|\boldsymbol{W}_V + \boldsymbol{B}_V \boldsymbol{A}_V\|}\boldsymbol{E}_j \widetilde{\boldsymbol{X}}}_{j\text{-th expert}}, \tag{1}
$$

where we define $\widetilde{\boldsymbol{X}} := \text{Vec}(\boldsymbol{X}) \in \mathbb{R}^{Nd}$ and the matrix $\boldsymbol{E}_j \in \mathbb{R}^{d \times Nd}$ that extracts the $j$-th token of $\boldsymbol{X}$, i.e., $\boldsymbol{E}_j \widetilde{\boldsymbol{X}} = \mathbf{x}_j$. Thus, the attention head can be expressed as an MoE with $N$ experts and we will leverage this connection to analyze the properties of DoRA and DoRAN.

**Problem setup:** Let $(\boldsymbol{X}_1, \boldsymbol{Y}_1), \ldots, (\boldsymbol{X}_n, \boldsymbol{Y}_n) \in \mathbb{R}^d \times \mathbb{R}^d$ be i.i.d. samples generated from the following regression model:

$$
\boldsymbol{Y}_i = f_{G_*}(\boldsymbol{X}_i) + \varepsilon_i, \quad i = 1, 2, \ldots, n. \tag{2}
$$

We assume that $\boldsymbol{X}_1, \boldsymbol{X}_2, \ldots, \boldsymbol{X}_n$ are i.i.d. samples from some probability distribution $\mu$ with bounded support; $\varepsilon_1, \varepsilon_2, \ldots, \varepsilon_n \in \mathbb{R}^d$ are independent Gaussian noise with $\mathbb{E}(\varepsilon_i|\boldsymbol{X}_i) = 0$ and $\text{Var}(\varepsilon_i|\boldsymbol{X}_i) = \sigma^2 I_d$ for all $i$. The regression function $f_{G_*}$ takes the form of an MoE model with $L$ unknown experts,

$$
f_{G_*}(\boldsymbol{X}) := \sum_{j=1}^{L} \text{softmax}\left(\left\{\boldsymbol{X}^\top m_{Q,k}\frac{\boldsymbol{C}_Q + \boldsymbol{B}_{Q,k}^* \boldsymbol{A}_{Q,k}^*}{\|\boldsymbol{C}_Q + \boldsymbol{B}_{Q,k}^* \boldsymbol{A}_{Q,k}^*\|}\boldsymbol{C}_K \boldsymbol{X} + c_k^*\right\}_{k=1}^{L}\right)_j
$$
$$
\left(m_{V,j}\frac{\boldsymbol{C}_V + \boldsymbol{B}_{V,j}^* \boldsymbol{A}_{V,j}^*}{\|\boldsymbol{C}_V + \boldsymbol{B}_{V,j}^* \boldsymbol{A}_{V,j}^*\|}\right)\boldsymbol{X}, \tag{3}
$$

and $G_* = \sum_{j=1}^{L} \exp(c_j^*)\delta_{(\boldsymbol{B}_{Q,j}^*, \boldsymbol{A}_{Q,j}^*, \boldsymbol{B}_{V,j}^*, \boldsymbol{A}_{V,j}^*)}$ represents the mixing measure, which is a combination of Dirac measures $\delta$, associated with the true unknown parameters $(c_j^*, \boldsymbol{B}_{Q,j}^*, \boldsymbol{A}_{Q,j}^*, \boldsymbol{B}_{V,j}^*, \boldsymbol{A}_{V,j}^*)_{j=1}^{L}$ in the compact parameter space $\Theta \subset \mathbb{R} \times \mathbb{R}^{d \times r} \times \mathbb{R}^{r \times d} \times \mathbb{R}^{d \times r} \times \mathbb{R}^{r \times d}$. The pre-trained matrices $\boldsymbol{C}_Q, \boldsymbol{C}_K, \boldsymbol{C}_V \in \mathbb{R}^{d \times d}$ are fixed and given (we changed the notation from matrix $\boldsymbol{W}$ to $\boldsymbol{C}$ to avoid confusion in subsequent analysis). To facilitate the analysis, we also assume that the magnitude parameters $\{m_{Q,j}, m_{V,j}\}_j$ are positive and known.

We note that the regression function in Eq. (3) resembles the adaptation strategy of DoRA, with different low-rank matrices added to the query and value matrices of the pre-trained model. We prove that this mechanism causes suboptimal sample complexity in estimating the unknown low-rank matrices $(\boldsymbol{B}_{Q,j}^*, \boldsymbol{A}_{Q,j}^*, \boldsymbol{B}_{V,j}^*, \boldsymbol{A}_{V,j}^*)$ (see Section 4.1). We refer to this as the non-shared structure. On the other hand, we employ the sharing technique in DoRAN, with added noise to the $\ell_2$ norms of the query and value matrices. We prove that these adjustments improve sample efficiency compared to the original technique (see Section 4.2).

### 4.1 NON-SHARED STRUCTURE CAUSES SUBOPTIMAL SAMPLE COMPLEXITY

We proceed to analyze the convergence rate of the low-rank matrices under the non-shared structure of the regression function in Eq. (3). To study the convergence behavior, it is natural to approach the problem from the perspective of estimating the ground-truth mixing measure $G_*$. For this purpose, we adopt the least-squares method (van de Geer, 2000) as follows:

$$
\widehat{G}_n \in \arg\min_{G \in \mathcal{G}_{L'}(\Theta)} \sum_{i=1}^{n}\left(\boldsymbol{Y}_i - f_G(\boldsymbol{X}_i)\right)^2, \tag{4}
$$

where we denote by $\mathcal{G}_{L'}(\Theta) := \{G = \sum_{j=1}^{\ell} \exp(c_j)\delta_{\boldsymbol{B}_{Q,j}, \boldsymbol{A}_{Q,j}, \boldsymbol{B}_{V,j}, \boldsymbol{A}_{V,j}} : 1 \leq \ell \leq L'\}$ the set of all mixing measures with at most $L'$ atoms. Since the true number of experts $L$ is typically

unknown, we assume that the number of fitted experts $L'$ chosen such that it is larger than $L$. To determine the convergence rates of the above estimator $\widehat{G}_n$, we use the following loss function, which is constructed based on the Voronoi cells concept from Manole & Ho (2022).

**Voronoi loss:** For a mixing measure $G$ with $L' > L$ atoms, the Voronoi cell set $\{\mathcal{V}_j \equiv \mathcal{V}_j(G) : j \in [L]\}$ is generated by the atoms of $G_*$ as $\mathcal{V}_j := \{i \in [L'] : \|\boldsymbol{H}_i - \boldsymbol{H}_j^*\| \leq \|\boldsymbol{H}_i - \boldsymbol{H}_\ell^*\|, \forall \ell \neq j\}$, where $\boldsymbol{H} := (\boldsymbol{B}_Q, \boldsymbol{A}_Q, \boldsymbol{B}_V, \boldsymbol{A}_V)$. Then, we define the Voronoi loss for the non-shared structure,

$$
\mathcal{D}_{1,r}(G, G_*) := \sum_{j=1}^{L} \Big| \sum_{i \in \mathcal{V}_j} \exp(c_i) - \exp(c_j^*) \Big|
$$

$$
+ \sum_{j=1}^{L} \sum_{i \in \mathcal{V}_j} \exp(c_i)(\|\Delta \boldsymbol{B}_{Q,ij}\|^r + \|\Delta \boldsymbol{A}_{Q,ij}\|^r + \|\Delta \boldsymbol{B}_{V,ij}\|^r + \|\Delta \boldsymbol{A}_{V,ij}\|^r),
$$

where we use the notation $\Delta \boldsymbol{A}_{ij} := \boldsymbol{A}_i - \boldsymbol{A}_j^*$ for a matrix $\boldsymbol{A}$ to quantify the estimation error. It is evident that once the order of the loss value is determined, the convergence rate of the estimators can be readily inferred. The following theorem presents the minimax rate of the Voronoi loss.

**Theorem 1.** *Under the setting of non-shared structure defined in Eq. (2) and Eq. (3), the following minimax lower bound of estimating $G_*$ using $\widehat{G}_n$ defined in Eq. (4) holds for any $r \in \mathbb{N}$:*

$$
\inf_{\widehat{G}_n \in \mathcal{G}_{L'}} \sup_{G \in \mathcal{G}_{L'}(\Theta) \setminus \mathcal{G}_{L-1}(\Theta)} \mathbb{E}_{f_G}[\mathcal{D}_{1,r}(\widehat{G}_n, G)] \gtrsim \frac{1}{\sqrt{n}},
$$

*where $\mathbb{E}_{f_G}$ denotes the expectation taken w.r.t. the product measure $f_G^n$.*

The proof of Theorem 1 is in Appendix D.1. Since the minimax lower bound above holds for any natural number $r$, which is the order of the estimation error in the Voronoi loss, it follows that the convergence rates of the low-rank matrix estimators $\{\boldsymbol{B}_{Q,j}^n, \boldsymbol{A}_{Q,j}^n, \boldsymbol{B}_{V,j}^n, \boldsymbol{A}_{Q,j}^n\}$ would be slower than any polynomial rates $\mathcal{O}_P(n^{-1/2r})$. Hence, the convergence rate could become as slow as $\mathcal{O}_P(1/\log^a(n))$ for some constant $a > 0$. As a consequence, we need an exponential sample size of order $\mathcal{O}(\exp(\epsilon^{-1/a})$ to obtain the estimation error $\epsilon$. This observation underscores the limited sample efficiency of the non-sharing structure. At a high level, the main technical obstacle to the above convergence rate arises from the fact that normalizing the adapted matrix induces an interaction among the parameters through the following partial differential equation (PDE):

$$
\Big\langle \boldsymbol{C}_Q + \boldsymbol{B}\boldsymbol{A}, \frac{\partial}{\partial(\boldsymbol{B}\boldsymbol{A})} \exp\Big(\boldsymbol{X}^\top m_Q \frac{\boldsymbol{C}_Q + \boldsymbol{B}\boldsymbol{A}}{\|\boldsymbol{C}_Q + \boldsymbol{B}\boldsymbol{A}\|} \boldsymbol{C}_K \boldsymbol{X}\Big)\Big\rangle = 0. \tag{5}
$$

This PDE causes the linear dependence of the terms in the Taylor expansion of the density discrepancy $f_{\widehat{G}_n}(\boldsymbol{X}) - f_{G_*}(\boldsymbol{X})$ that affect the convergence of expert estimation.

## 4.2 SHARED-STRUCTURE WITH ADDED NOISE

Now, we consider the following non-linear reparameterization that shares the parameters within the low-rank adapting matrices:

$$
\boldsymbol{A}_Q = \boldsymbol{A}_V = \sigma_1(\boldsymbol{W}_1 \boldsymbol{A}), \quad \boldsymbol{B}_Q = \boldsymbol{B}_V = \sigma_2(\boldsymbol{W}_2 \boldsymbol{B}), \tag{6}
$$

where $\boldsymbol{A} \in \mathbb{R}^{m \times d}, \boldsymbol{B} \in \mathbb{R}^{m' \times r}, \boldsymbol{W}_1 \in \mathbb{R}^{r \times m}, \boldsymbol{W}_2 \in \mathbb{R}^{d \times m'}$ are learnable matrices with given dimension $m, m'$; $\sigma_1, \sigma_2$ are some non-linear activation functions. Moreover, we perturb the $\ell_2$-norm components of the query and value matrices by adding a small noise term in the denominator. In addition to the aforementioned benefits of the added noise, we note that it also helps avoid the interaction in Eq. (5). With these adjustments, the ground-truth regression function is:

$$
f_{\widetilde{G}_*}(\boldsymbol{X}) := \sum_{j=1}^{L} \mathrm{softmax}\Big( \Big\{ \boldsymbol{X}^\top m_{Q,k} \frac{\boldsymbol{C}_Q + \sigma_2(\boldsymbol{W}_{2,k}^* \boldsymbol{B}_k^*) \sigma_1(\boldsymbol{W}_{1,k}^* \boldsymbol{A}_k^*)}{\|\boldsymbol{C}_Q + \sigma_2(\boldsymbol{W}_{2,k}^* \boldsymbol{B}_k^*) \sigma_1(\boldsymbol{W}_{1,k}^* \boldsymbol{A}_k^*)\| + \tau_Q} \boldsymbol{C}_K \boldsymbol{X} + c_k^* \Big\}_{k=1}^{L} \Big)_j
$$

$$
\times \Big( m_{V,j} \frac{\boldsymbol{C}_V + \sigma_2(\boldsymbol{W}_{2,j}^* \boldsymbol{B}_j^*) \sigma_1(\boldsymbol{W}_{1,j}^* \boldsymbol{A}_j^*)}{\|\boldsymbol{C}_V + \sigma_2(\boldsymbol{W}_{2,j}^* \boldsymbol{B}_j^*) \sigma_1(\boldsymbol{W}_{1,j}^* \boldsymbol{A}_j^*)\| + \tau_V} \Big) \boldsymbol{X}, \tag{7}
$$

here the ground-truth mixing measure is $\widetilde{G}_* := \sum_{j=1}^L \exp(c_j^*)\delta_{(\boldsymbol{W}_{2,j}^*\boldsymbol{B}_j^*,\boldsymbol{W}_{1,j}^*\boldsymbol{A}_j^*)}$. Similarly, we consider the least-square method for our estimator, i.e., $\widetilde{G}_n \in \arg\min_{\widetilde{G}\in\widetilde{\mathcal{G}}_{L'}(\Theta)} \sum_{i=1}^n \left(\boldsymbol{Y}_i - f_{\widetilde{G}}(\boldsymbol{X}_i)\right)^2$, where $\widetilde{\mathcal{G}}_{L'}(\Theta) := \{G = \sum_{i=1}^\ell \exp(c_i)\delta_{(\boldsymbol{W}_{2,i}\boldsymbol{B}_i,\boldsymbol{W}_{1,i}\boldsymbol{A}_i)} : 1 \le \ell \le L'\}$.

The Voronoi loss function in this setting is constructed as

$$\mathcal{D}_2(\widetilde{G},\widetilde{G}_*) := \sum_{j=1}^L \Big| \sum_{i\in\mathcal{V}_j}\exp(c_i) - \exp(c_j^*)\Big| + \sum_{j\in[L]:|\mathcal{V}_j|=1}\sum_{i\in\mathcal{V}_j}\exp(c_i)(\|\Delta(\boldsymbol{W}_2\boldsymbol{B})_{ij}\| + \|\Delta(\boldsymbol{W}_1\boldsymbol{A})_{ij}\|)$$
$$+ \sum_{j\in[L]:|\mathcal{V}_j|>1}\sum_{i\in\mathcal{V}_j}\exp(c_i)(\|\Delta(\boldsymbol{W}_2\boldsymbol{B})_{ij}\|^2 + \|\Delta(\boldsymbol{W}_1\boldsymbol{A})_{ij}\|^2),$$

where we denote, for example, $\Delta(\boldsymbol{W}_2\boldsymbol{B})_{ij} := \boldsymbol{W}_{2,i}\boldsymbol{B}_i - \boldsymbol{W}_{2,j}^*\boldsymbol{B}_j^*$. Before presenting the main theorem, we introduce some mild assumptions on the activation functions $\sigma_1$ and $\sigma_2$. Due to the space constraint, we defer these assumptions until the proof of Theorem 2 in Appendix D.2.

**Theorem 2.** *Assume that the activation functions $\sigma_1$ and $\sigma_2$ meet the assumptions specified in Appendix D.2. Then, the estimator $\widetilde{G}_n$ converges to the true mixing measure $\widetilde{G}_*$ at the following rate: $\mathcal{D}_2(\widetilde{G}_n,\widetilde{G}_*) = \mathcal{O}_P(\sqrt{\log(n)/n})$.*

The proof of Theorem 2 is Appendix D.2. Based on the construction of the loss $\mathcal{D}_2(\widetilde{G},\widetilde{G}_*)$, Theorem 2 suggests that the convergence rate of the estimators $\boldsymbol{W}_{2,j}\boldsymbol{B}_j$ and $\boldsymbol{W}_{1,j}\boldsymbol{A}_j$ to the true matrices $\boldsymbol{W}_{2,j}^*\boldsymbol{B}_j^*$ and $\boldsymbol{W}_{1,j}^*\boldsymbol{A}_j^*$ are ranging from order $\mathcal{O}_P([\log(n)/n]^{\frac{1}{2}})$, if the $j$-th Voronoi cell $\mathcal{V}_j$ has one element, to order $\mathcal{O}_P([\log(n)/n]^{\frac{1}{4}})$, if the $j$-th Voronoi cell $\mathcal{V}_j$ has more than one element. In both cases, we only need the sample size with the order at most $\mathcal{O}(\epsilon^{-4})$ to achieve the desired estimation error $\epsilon$, compared with the required sample size of order $\mathcal{O}(\exp(\epsilon^{-1/a}))$ of the non-shared structure analyzed in the previous section.

## 5 EXPERIMENTS

**Experimental Settings.** This section presents experimental results evaluating the effectiveness of our proposed methods: $\tau-$DoRA, which refers to DoRA with our stabilization term $\tau$, and Do-RAN. We consider two tasks: Image Classification on VTAB-1K and FGVC, and Commonsense Reasoning. We compare our method against several PEFT methods, including *LoRA* (Hu et al., 2022), *Prefix Tuning* (Li & Liang, 2021), *Parallel Adapter* (He et al., 2022), PiSSA (Meng et al., 2024), and *DoRA* Liu et al. (2024). We also conduct a Sample Efficiency task comparing DoRAN and DoRA on the Commonsense Reasoning task. Baseline results for the vision tasks are directly adapted from Xin et al. (2024), while those for the language tasks are taken from Liu et al. (2024). For a fair comparison, DoRAN is evaluated under the same experimental settings as DoRA and LoRA, including using the same ranks and identical data augmentation strategies. For consistency with the theoretical setting, we fine-tune the query and value projection matrices of each attention layer. Additionally, we present an extended variant of DoRAN that applies low-rank matrices to the proj_up and proj_down matrices in both LLaMA-7B and LLaMA-13B, as described in Ablation B.5. Full hyperparameter details are in Appendix B.1.

**Image Classification.** We first consider the vision domain, where we aim to fine-tune the ViT-B/16 variant of the Vision Transformer architecture (Dosovitskiy, 2021b), which was fine-tuned on the ImageNet-21K (Deng et al., 2009) dataset.

This experiment consists of two benchmarks. The first is the Visual Task Adaptation Benchmark (VTAB-1K), a transfer-learning benchmark that evaluates vision models on 19 image classification tasks across three domains—Natural, Specialized, and Structured—using only 1000 labeled examples per task. Per-domain results are reported in Table 1, with detailed results in Appendix B.3. When incorporating stabilization into DoRA ($\tau-$DoRA), as described in Section 3.1, the resulting $\tau-$DoRA improves overall performance by $0.5\%$ compared to the original DoRA with nearly no additional computational overhead, suggesting the practical benefits of including this stabilization term. Moreover, with hypernetwork-based parameter generation, our method DoRAN significantly outperforms all baselines, notably improving over DoRA by 1.1% on Natural, 1.6% on Specialized,

and 1.8% on Structured tasks. While hypernetworks inevitably introduce additional parameters, sharing half of the architecture across attention heads and query/value projections greatly reduces this overhead. As a result, DoRAN requires only 0.09% more trainable parameters relative to total parameters of ViT compared to DoRA, yet achieves significant performance gains.

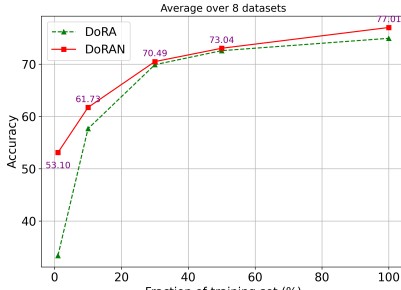

Table 1: Average Image Classification results on VTAB-1K domains.

| Method | #Params (%) | Natural | Specialized | Structured | AVG |
|--------|-------------|---------|-------------|------------|-----|
| FFT | - | 75.89 | 83.38 | 47.64 | 65.6 |
| LoRA | 0.39 | 79.40 | 84.55 | 59.78 | 72.2 |
| DoRA | 0.40 | 80.33 | 85.15 | 60.11 | 72.8 |
| PiSSA | 0.39 | 80.33 | 85.25 | 60.20 | 72.9 |
| Adapter | 0.18 | 79.01 | 84.08 | 58.49 | 71.4 |
| Prefix | 0.16 | 77.06 | 82.30 | 52.00 | 67.6 |
| $\tau-$**DoRA** | 0.41 | 80.89 | 85.57 | 60.56 | 73.3 |
| **DoRAN** | 0.49 | **81.46** | **86.78** | **61.98** | **74.4** |

Figure 2: Average sample efficiency on the commonsense reasoning datasets.

The second benchmark is Fine-Grained Visual Classification (FGVC), a family of datasets designed to distinguish highly similar categories within a domain. These datasets evaluate a model's ability to capture subtle, local visual cues such as textures, parts, and patterns, and are widely used to assess recognition beyond coarse object categories. As reported in Table 2, $\tau-$DoRA significantly improves performance over standard DoRA across all datasets, with an average gain of 1.6%. Furthermore, DoRAN, which combines added noise with hypernetwork-based reparameterization, achieves the best results on nearly all datasets (except Stanford Cars), with notable improvements—+2.3% over DoRA and +4.7% over LoRA—while introducing only 0.09% additional trainable parameters relative to the total parameters of ViT. These results highlight the effectiveness of integrating added noise with the reparameterization mechanism. .

Table 2: Image Classification results on the FGVC datasets

| Method | #Params (%) | CUB-200 -2011 | NABirds | Oxford Flowers | Stanford Dogs | Stanford Cars | Average |
|--------|-------------|---------------|---------|----------------|---------------|---------------|---------|
| FFT | - | 87.3 | 82.7 | 98.8 | 89.4 | 84.5 | 88.54 |
| LoRA | 0.55 | 84.6 | 78.2 | 98.9 | 85.1 | 77.1 | 84.78 |
| DoRA | 0.57 | 87.3 | 80 | 99.1 | 87.6 | 81.9 | 87.18 |
| PiSSA | 0.55 | 87.1 | 80.5 | 99.1 | 84.8 | 82.3 | 86.66 |
| Adapter | 0.47 | 87.1 | 84.3 | 98.5 | 89.8 | 68.6 | 85.66 |
| Prefix | 0.42 | 87.5 | 82 | 98 | 74.2 | **90.2** | 86.38 |
| $\tau-$**DoRA** | 0.59 | 88.3 | 83.4 | 99.2 | 90.2 | 82.9 | 88.8 |
| **DoRAN** | 0.66 | **88.5** | **85.3** | **99.2** | **90.8** | 83.7 | **89.5** |

**Commonsense Reasoning.** Having shown the effectiveness of both $\tau-$DoRA and DoRAN on vision tasks, we now evaluate them on language tasks using the commonsense reasoning benchmark, which includes eight sub-tasks. Following Hu et al. (2023), we merge all datasets into a 150k training set and test on LLaMA-7B and 13B (Touvron et al., 2023). As shown in Table 3, $\tau-$DoRA outperforms most datasets on LLaMA-7B and improves DoRA by +1.9% (7B) and +0.2% (13B), demonstrating the benefit of added noise. DoRAN, combining noise with hypernetwork-based reparameterization, achieves the best average accuracy on both scales: 77.01% on 7B (+2.0% over DoRA, ≈ +3.0% over LoRA) and 80.76% on 13B (+0.7% over DoRA). On LLaMA-7B, it performs especially well on HellaSwag (83.45%), ARC-c (65.02%), and OBQA (79.6%, best in block). Overall, DoRAN consistently improves commonsense reasoning, with gains persisting on harder benchmarks (e.g., ARC-c), while adding only negligible parameters compared to DoRA and LoRA.

**Sample Efficiency.** In Section 4, we outlined the theoretical advantages of incorporating a shared structure with added noise to improve sample efficiency. Here, we empirically validate this claim by comparing DoRAN with DoRA on the commonsense reasoning task using the LLaMA-7B setting. Following d'Ascoli et al. (2021), we subsample each class at fractions $f = \{1\%, 10\%, 30\%, 50\%, 100\%\}$ and scale the number of training epochs by $1/f$ to keep the total number of training examples constant. The results, presented in Figure 2 and detailed in Appendix B.2, show that DoRAN consistently outperforms DoRA across all fractions. The improvement is especially pronounced in the low-data regime, exceeding 20% when only 1% of the dataset is used, thereby demonstrating the superior sample efficiency of DoRAN over vanilla DoRA.

Table 3: Performance on the Commonsense Reasoning task

| Model | Method | #Params (%) | BoolQ | PIQA | SIQA | HellaSwag | WinoGrande | ARC-e | ARC-c | OBQA | Average |
|-------|--------|-------------|-------|------|------|-----------|------------|-------|-------|------|---------|
| LLaMA-7B | Prefix | 0.11 | 64.3 | 76.8 | 79.3 | 42.1 | 72.1 | 72.9 | 54 | 60.6 | 65.26 |
| | LoRA | 0.25 | 67.2 | 79.4 | 76.6 | 78.3 | 78.4 | 77.1 | 61.5 | 74.2 | 74.09 |
| | DoRA | 0.25 | 67.22 | 79.98 | 76.66 | 80.66 | **79.72** | 79.5 | 61.01 | 74.8 | 74.94 |
| | Adapter | 0.99 | 63 | 79.2 | 76.3 | 67.9 | 75.7 | 74.5 | 57.1 | 72.4 | 70.76 |
| | $\tau-$**DoRA** | 0.25 | 69.45 | 81.39 | 77.18 | 83.76 | 79.56 | **80.26** | 64.59 | 78.6 | 76.85 |
| | **DoRAN** | 0.26 | **69.82** | **81.01** | **77.89** | **83.45** | 79.56 | 79.76 | **65.02** | **79.6** | **77.01** |
| LLaMA-13B | Prefix | 0.03 | 65.3 | 75.4 | 72.1 | 55.2 | 68.6 | 79.5 | 62.9 | 68 | 68.38 |
| | LoRA | 0.2 | 71.7 | 82.4 | 79.6 | **90.4** | 83.6 | 83.1 | 68.5 | 82.1 | 80.18 |
| | DoRA | 0.2 | 72.2 | 83.19 | 80.81 | 88.92 | 81.93 | 82.95 | 69.37 | 81 | 80.05 |
| | Adapter | 0.8 | 71.8 | 83 | 79.2 | 88.1 | 82.4 | 82.5 | 67.3 | 81.8 | 79.51 |
| | $\tau-$**DoRA** | 0.2 | 71.01 | 84.39 | **80.96** | 89.65 | **83.74** | 83 | 67.24 | 82.2 | 80.27 |
| | **DoRAN** | 0.21 | **71.8** | **84.5** | 80.6 | 89.79 | 83.19 | **83.29** | 68.69 | **84.2** | **80.76** |

## 6 CONCLUSION

We propose **DoRAN**, a stable yet efficient variant of DoRA. By introducing a noise-stabilized normalization and low-rank updates generated by auxiliary networks, DoRAN addresses key limitations of standard DoRA. Our theoretical analysis established improved convergence guarantees, and empirical results confirmed gains in both stability and sample efficiency.

**Limitations and future works:** The auxiliary network design introduces additional hyperparameters and structural choices that may complicate practical deployment. Moreover, DoRAN inherits the computational overhead associated with DoRA's normalization term, as computing the matrix norm $\|W_0 + BA\|$ requires materializing the full matrix, which can be inefficient in certain cases and applications. It is important to note, however, that DoRAN is designed to address an orthogonal limitation of DoRA: its instability and sample inefficiency during training. Improving computational efficiency is therefore complementary to our contribution. Future work may incorporate techniques that mitigate DoRA's computational cost into the DoRAN framework, yielding a more efficient and robust fine-tuning method. Additionally, we will explore improved hypernetwork architectures to extend DoRAN to broader model families and application domains.

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

# Supplement to "DoRAN: Stabilizing Weight-Decomposed Low-Rank Adaptation via Noise Injection and Auxiliary Networks"

In this supplementary material, firstly, we provide the deferred related works and notations in Appendix A. Next, we present the additional experiment details in Appendix B.1. Next, detailed proofs for Theorems 1 and 2 are provided in Appendix D.1 and D.2, respectively. Lastly, we discuss the use of large language models in this paper in Appendix E.

## A  RELATED WORKS

**Parameter-Efficient Fine-Tuning.** Fine-tuning large pre-trained models has become increasingly expensive with the growing size of these models. Thus, it has motivated a wide range of parameter-efficient fine-tuning (PEFT) methods that only update a relatively small number of parameters. Existing PEFT methods can be divided into three categories. The first category is referred to as *adapter-based* methods, which insert lightweight modules into the frozen backbone. For example, (Houlsby et al., 2019) proposed adding linear modules to the existing layers in sequence, while (He et al., 2022) proposed integrating these modules in parallel to improve performance.

The second category is *prompt-based* methods, which add learnable soft tokens (prompts) to the initial input. As proposed by (Jia et al., 2022; Le et al., 2025a; Lester et al., 2021; Razdaibiedina et al., 2023; Zhang et al., 2024a; Diep et al., 2025), the methods of this paradigm optimize only the additional prompt embeddings while keeping the backbone parameters frozen. This design enables efficient transfer learning with minimal parameter updates. However, despite their efficiency, prompt-based PEFT approaches introduce additional context tokens during inference, thereby increasing computational cost and latency compared to the original frozen backbone models—a limitation that constrains their practical deployment in latency-sensitive applications.

The third category of PEFT is based on low-rank adaptation, which is pioneered by LoRA (Hu et al., 2022) and its variants. LoRA uses low-rank matrices to approximate the weight updates while keeping the pre-trained weights frozen. A key advantage of LoRA is that these low-rank updates can be merged into the original weights before inference, so no extra inference latency is added compared to the original models. Later works have aimed to improve the stability, efficiency, and performance of LoRA. For example, DyLoRA and AutoLoRA (Valipour et al., 2023; Zhang et al., 2024b) improve the performance on downstream tasks by finding the optimal rank of LoRA matrices using various methods. In the parameter efficiency aspect, VeRA (Kopiczko et al., 2024) employs a single pair of shared low-rank matrices across all layers, which further reduces the number of parameters while still maintaining competitive performance. Liu et al. (2024) found that LoRA's magnitude and direction updates differ significantly from full fine-tuning, which might limit its learning capacity. Thus, DoRA (Liu et al., 2024) was proposed, which decomposes pre-trained weights into magnitude and direction components and uses LoRA to update the direction component, to better approximate full fine-tuning. In parallel, Truong et al. (2025b) and Pham et al. (2025) explore the integration of Bayesian inference and sharpness-aware techniques into LoRA-based fine-tuning frameworks to enhance robustness and generalization. By introducing uncertainty-aware adaptation mechanisms and probabilistic regularization, these methods provide a principled approach to mitigating overfitting and improving the stability of parameter-efficient model tuning. In the parameter efficiency aspect, VeRA Kopiczko et al. (2024) employs a single pair of shared low-rank matrices across all layers, which further reduces the number of parameters while still maintaining competitive performance.

**Hypernetworks.** The HyperNetwork framework Ha et al. (2017) introduced a paradigm in which the parameters of a target model are dynamically generated by an auxiliary neural network, termed a HyperNetwork, rather than being directly learned. Initially explored in the context of recurrent neural networks, HyperNetworks demonstrated improved adaptability and generalization by producing context-dependent weight updates (Ha et al., 2017). Subsequent research extended this idea to continual learning, where task-specific weight generation via HyperNetworks helped alleviate catastrophic forgetting (von Oswald et al., 2022). In the realm of parameter-efficient fine-tuning (PEFT), HyperNetworks have proven particularly useful for enabling cross-task adaptation and reducing parameter redundancy. For instance, Mahabadi et al. (2021) proposed generating task-specific adapter weights through a shared HyperNetwork, achieving significant parameter savings while maintain-

ing competitive performance. Similarly, Li & Liang (2021) leveraged HyperNetworks to enhance prompt tuning, replacing direct parameter optimization with a meta-network that predicts prompt parameters. More recent works, such as (Le et al., 2025b) and (Truong et al., 2025a), have analyzed the theoretical benefits of HyperNetwork-driven PEFT, showing improved sample efficiency and generalization, thereby motivates novel fine-tuning techniques.

**Mixture of Experts.** The Mixture of Experts (MoE) framework (Jacobs et al., 1991; Jordan & Jacobs, 1994) has been widely adopted to scale model capacity without proportional increases in computation (Eigen et al., 2014; Shazeer et al., 2017). Modern implementations such as sparsely-gated MoE layers activate only a small set of experts per token, enabling trillion-parameter models with tractable training cost (Shazeer et al., 2017; Lepikhin et al., 2021; Fedus et al., 2022). Efficient routing and load-balancing are central challenges, addressed via auxiliary losses, stochastic routing, and expert capacity constraints (Lepikhin et al., 2021; Fedus et al., 2022; Dai et al., 2022). From a theoretical side, the convergence behavior of MoE and the sample complexity of estimating parameters and experts in MoE have been extensively explored in (Nguyen et al., 2023; 2024a;c;b). MoE has also been linked to the transformer attention mechanism, where attention heads can be viewed as experts and the softmax query-key interactions as gating distributions (Jin et al., 2025; Csordás et al., 2024). RepLoRA (Truong et al., 2025a) formalizes multi-head self-attention as an MoE and shows that LoRA fine-tunes these embedded experts via low-rank updates. Systems-oriented work has improved scalability with hierarchical routing (Du et al., 2022), hashing (Roller et al., 2021), and specialized runtimes (Rajbhandari et al., 2022; He et al., 2021).

# B    ADDITIONAL EXPERIMENTAL DETAILS

## B.1    IMPLEMENTATION DETAILS

For vision tasks, we experiment with ViT-B/16 (Vaswani et al., 2017) for 100 epochs using 100 warmup steps, a batch size of 64, a Low-Rank Matrix rank of 8, and $\alpha = 8$. Optimization is performed with AdamW and a cosine learning rate scheduler. Learning rate and weight decay are tuned via grid search over $\{0.001, 0.005, 0.01, 0.05, 0.1\}$ and $\{0.0001, 0.0005, 0.001, 0.01, 0.1\}$, respectively. For the hypernetwork in a low-rank matrix $B$, we set the input dimension to 64, the hidden dimension to 16, and use leaky-ReLU activation.

For commonsense reasoning tasks, we use LLaMA-7B (32 layers) and LLaMA-13B (40 layers) Touvron et al. (2023). Training runs for 3 epochs on a single A100 GPU with 100 warmup steps, batch size of 32, learning rate $1e-4$, dropout 0.05, rank 32, and $\alpha = 64$. We optimize the models using the AdamW optimizer with a linear learning rate scheduler. In LLaMA-7B, the hypernetwork for the low-rank matrix $B$ has an input dimension of 64, a hidden dimension of 32, and leaky-ReLU activation; in LLaMA-13B, the hidden dimension is 40 with the same activation.

## B.2    DETAIL OF SAMPLE EFFICIENCY

We provide in Figure 3 the details of the sample efficiency problem in each commonsense reasoning dataset with the LLaMA-7B setting.

## B.3    DETAIL OF RESULTS ON VTAB-1K DATASETS

In Table 4, we provide the results of DoRAN in detail for each dataset in the VTAB-1K domain. Compared to DoRA, DoRAN delivers consistent improvements across all datasets and achieves state-of-the-art results on nearly all of them—except for CIFAR100 and sNORB-ele—while requiring only a modest increase in parameters. This highlights the benefit of sharing query and value matrices and using noise, as shown in Section 4.

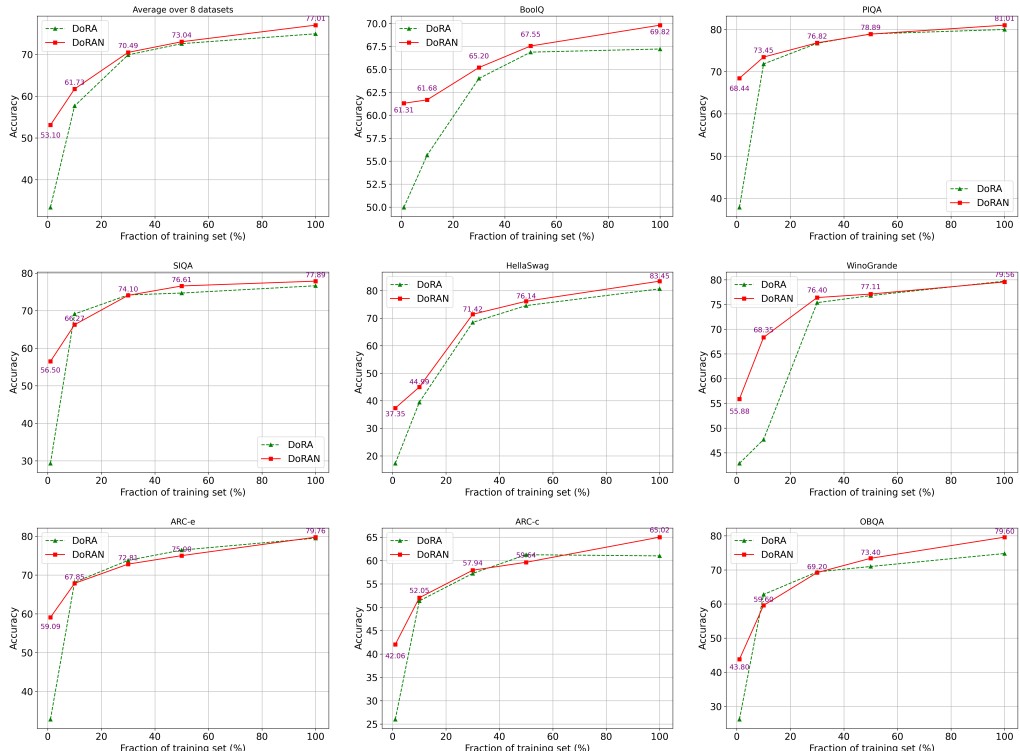

Figure 3: The detail of sample efficiency on each commonsense reasoning dataset with LLaMA-7B settings.

Table 4: Image Classification results on the VTAB-1K dataset

| | **Natural** | | | | | | | **Specialized** | | | | **Structured** | | | | | | | | |
| Method | CIFAR100 | Caltech101 | DTD | Flower102 | Pets | SVHN | Sun397 | Camelyon | EuroSAT | Resisc45 | Retinopathy | Clevr-Count | Clevr-Dist | DMLab | KITTI | dSpr-Loc | dSpr-ori | sNORB-Azim | sNORB-Ele | AVG |
|---|---|---|---|---|---|---|---|---|---|---|---|---|---|---|---|---|---|---|---|---|
| FFT | 68.9 | 87.7 | 64.3 | 97.2 | 86.9 | 87.4 | 38.8 | 79.7 | 95.7 | 84.2 | 73.9 | 56.3 | 58.6 | 41.7 | 65.5 | 57.5 | 46.7 | 25.7 | 29.1 | 65.6 |
| LoRA | 67.1 | 91.4 | 69.4 | 98.2 | 90.4 | 85.3 | 54 | 84.9 | 95.3 | 84.4 | 73.6 | 82.9 | 69.2 | 49.8 | 78.5 | 75.7 | 47.1 | 31 | **44** | 72.2 |
| DoRA | 67.9 | 90.4 | 70.6 | 99 | 90.2 | 89.6 | 54.6 | 83.9 | 95.5 | 85.3 | 75.9 | 80.8 | 69.8 | 50.5 | 80.9 | 79.1 | 47.7 | 32.5 | 39.6 | 72.8 |
| PiSSA | 68.9 | 90.7 | 71.2 | 98.7 | 90.2 | 87.9 | 54.7 | 84.2 | 95.3 | 85.5 | 76 | 81.4 | 69.7 | 51 | 80.5 | 78.9 | 47.2 | 31.6 | 41.3 | 72.9 |
| Adapter | 69.2 | 90.1 | 68 | 98.8 | 89.9 | 82.8 | 54.3 | 84 | 94.9 | 81.9 | 75.5 | 80.9 | 65.3 | 48.6 | 78.3 | 74.8 | 48.5 | 29.9 | 41.6 | 71.4 |
| Prefix | **75.5** | 90.7 | 65.4 | 96.6 | 86 | 78.5 | 46.7 | 79.5 | 95.1 | 80.6 | 74 | 69.9 | 58.2 | 40.9 | 69.5 | 72.4 | 46.8 | 23.9 | 34.4 | 67.6 |
| $\tau-$DoRA | 69.4 | 91.2 | 71.6 | 99 | 90.3 | 89.6 | 55.1 | 84.9 | 95.5 | 85.8 | 76.1 | 81.9 | 69.6 | 52.5 | 81.0 | 79.4 | 47.7 | 32.8 | 39.5 | 73.3 |
| **DoRAN** | 70.2 | **92.4** | **71.9** | **99.1** | **91.3** | **90.3** | 55 | **86.9** | **96.4** | **87.4** | **76.4** | **84** | **70** | **53.2** | **81.9** | **79.9** | **49.2** | **35.6** | 42 | **74.4** |

## B.4 ABLATION ON THE ROLE OF NOISE

In this ablation study, we assess the role of $\tau$ to the performance of DoRAN. To do so, we consider experiments on the FGVC dataset on the following settings: (1) DoRA without hypernetwork and with learnable $\tau$, which corresponds to DoRA-$\tau$ in the main experiment; (2) DoRA with the hypernetwork and Gaussian $\tau$, (3) DoRA with the hypernetwork and no noise, and (4) DoRAN. The results are reported in Table 5.

## B.5 ABLATION ON FINE-TUNING MODULES

Beyond applying low-rank matrices to the query and value matrices in each layer, we also explore whether our DoRA-based design can generalize when extended to additional modules. We assess the performance of DoRAN on two settings: when it is applied to the query, value, proj_up and proj_down matrices, and when it is applied to every projection matrices while maintaining our pro-

Table 5: Performance with different types of $\tau$ on the FGVC datasets

| Method | CUB-200 -2011 | NABirds | Oxford Flowers | Stanford Dogs | Stanford Cars | AVG |
|---|---|---|---|---|---|---|
| Vanilla DoRA | 87.3 | 80 | 99.1 | 87.6 | 81.9 | 87.18 |
| DoRAN w/o hypernetwork | 88.3 | 83.4 | 99.2 | 90.2 | 82.9 | 88.8 |
| DoRAN + Gaussian $\tau$ | 88.3 | 85.2 | 99.0 | 90.5 | 83.3 | 89.3 |
| DoRAN w/o $\tau$ | 87.9 | 85.4 | 99.2 | 90.5 | 83.5 | 89.3 |
| DoRAN | **88.5** | **85.3** | **99.2** | **90.8** | **83.7** | **89.5** |

posed design for the query and value matrices and using the standard low-rank formulation for the proj_up and proj_down. As in Table 6 and Table 7, in both settings DoRAN achieves the best performance in both LLaMA-7B and LLaMA-13B. These results indicate that our method, originally designed for query and value matrices in multi-head attention, remains effective even when low-rank matrices are applied more broadly across the model.

Table 6: Ablation Study on Low-Rank Matrices in Query, Value, Up, and Down Weights.

| Model | Method | #Params (%) | BoolQ | PIQA | SIQA | HellaSwag | WinoGrande | ARC-e | ARC-c | OBQA | Average |
|---|---|---|---|---|---|---|---|---|---|---|---|
| LLaMA-7B | LoRA | 0.7 | 69.82 | **83.51** | 78.66 | **85.77** | 74.27 | 81.69 | **66.81** | 77.8 | 77.29 |
| | DoRA | 0.71 | 68.69 | 81.66 | 78.3 | 84.24 | 80.51 | 81.36 | 65.44 | 79.2 | 77.43 |
| | DoRAN | 0.72 | **70.46** | 82.1 | **78.81** | 84.78 | **81.61** | 82.28 | 66.55 | **80.6** | **78.4** |
| LLaMA-13B | LoRA | 0.57 | 72.11 | 83.73 | 80.5 | 90.5 | 83.74 | 82.11 | 68.09 | 82.4 | 80.4 |
| | DoRA | 0.58 | **72.42** | 84.98 | **81.17** | 91.81 | 84.61 | 84.22 | 69.88 | 82.8 | 81.49 |
| | DoRAN | 0.58 | 72.17 | **85.91** | 80.19 | **92.77** | **85.71** | **85.82** | **71.93** | **84** | **82.31** |

Table 7: Ablation Study on Low-Rank Matrices in Query, Key, Value, Up, and Down Weights.

| Model | Method | #Params (%) | BoolQ | PIQA | SIQA | HellaSwag | WinoGrande | ARC-e | ARC-c | OBQA | Average |
|---|---|---|---|---|---|---|---|---|---|---|---|
| LLaMA-7B | DoRA | 0.84 | 69.7 | 82.3 | 79.5 | 86.3 | 81 | 81.9 | 66.2 | 79.2 | 78.26 |
| | DoRAN | 0.84 | 70.2 | 83.8 | 78.6 | 85 | 81.1 | **82** | **66.7** | **80.4** | **78.5** |
| LLaMA-13B | DoRA | 0.68 | **72.4** | 85.1 | **80.1** | 91.9 | 84.3 | 84.8 | 70.9 | 82.8 | 81.54 |
| | DoRAN | 0.68 | 72.2 | **86.1** | 80.4 | **92.9** | **84.9** | 84.6 | 71.8 | 83.8 | **82.1** |

## B.6 EXPERIMENTS ON RECENT MODEL

To further evaluate the effectiveness of DoRAN on recent and competitive large language models, we conducted an additional experiment using LLaMA3-8B. As shown in Table 8, DoRAN continues to provide measurable benefits at this scale, yielding an average improvement of +0.6% over vanilla DoRA.

## B.7 ANALYSIS ON THE VALUES OF $\tau$

To analyze the roles and dynamics of $\tau$, we conducted an experiment on the CIFAR100 dataset and visualize the violin plots of the learned $\tau$ values across the 12 ViT layers—both after the first 10 epochs and at the end of training.

At epoch 10, as shown in Figure 4 except the last layer, the distribution of $\tau$ values is relatively uniform across layers: both the mean and variance remain relatively similar throughout the depth of the network. This indicates that, early in training, the model has not yet differentiated the roles of different layers in terms of how strongly they should rely on direction learning versus norm control.

By contrast, in the final epoch, as shown in Figure 5 clear structural patterns emerge. The early layers exhibit larger variance in $\tau$, suggesting that these layers learn more diverse or specialized transformations and therefore benefit from a wider range of direction-vs-norm tradeoffs. The later layers, in comparison, converge to much tighter $\tau$ distributions, suggesting that they operate in a more homogeneous regime.

Moreover, the mean $\tau$ values increase steadily over training. This trend shows that the model gradually shifts toward stronger direction learning rather than relying solely on norm control, differentiating DoRAN from DoRA and highlighting the additional flexibility introduced by the $\tau$ parameter.

Table 8: Performance on the commonsense reasoning task with LLaMA3-8B

| Model | Method | BoolQ | PIQA | SIQA | HellaSwag | WinoGrande | ARC-e | ARC-c | OBQA | Average |
|-------|--------|-------|------|------|-----------|------------|-------|-------|------|---------|
| LLaMA3-8B | DoRA | **73.5** | 87.9 | 79 | 93.4 | 83.7 | 89.8 | **78.9** | 84.4 | 83.8 |
| | DoRAN | 72.9 | **88.7** | **80.5** | **94.7** | **84.1** | **90.3** | 78 | **85.6** | **84.4** |

Overall, these results support our claim that $\tau$ adapts meaningfully across depth and provides structural benefits.

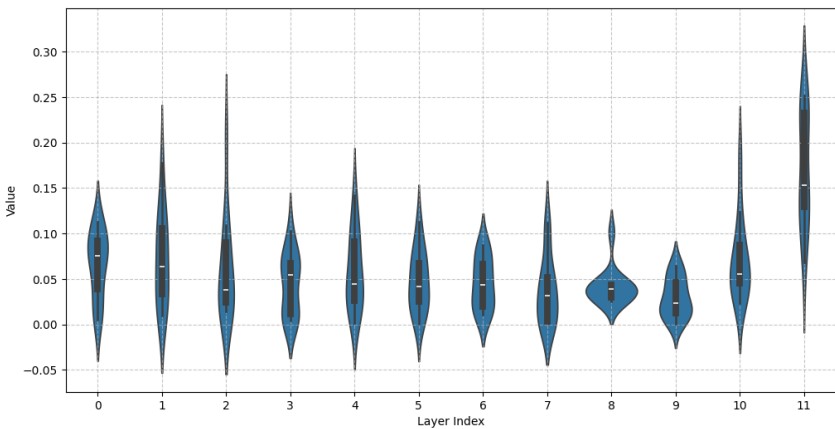

Figure 4: Distribution of $\tau$ after 10 epochs across 12 layers of the ViT

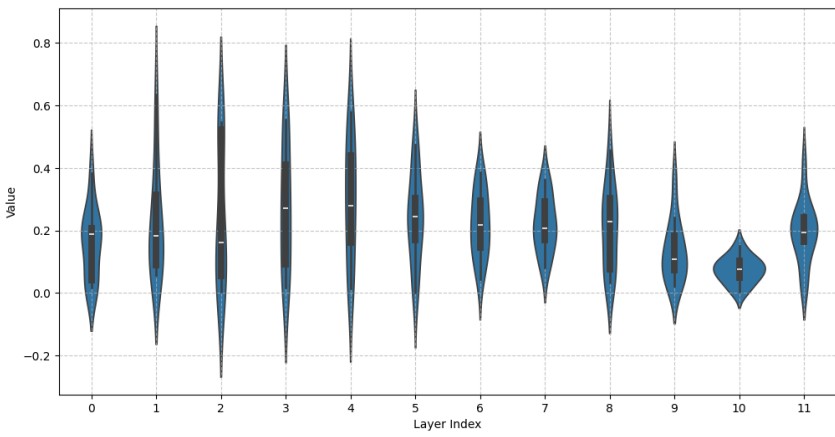

Figure 5: Distribution of final $\tau$ after 100 epochs across 12 layers of the ViT

## C   ADDITIONAL EXPERIMENTS

In this appendix, we perform additional experiments to illustrate how our proposed method DoRAN compares with a strong baseline in the vision domain known as MLAE (Wang et al., 2024).

**The configurations for comparing with MLAE:** To ensure a fair comparison among low-rank adaptation methods reported in our paper, we emphasize that we apply low-rank fine-tuning to the query and value projection weights for MLAE in both vision and language tasks. This differs from the original implementations of the MLAE paper, which apply low-rank updates to all the projection matrices including the query, key, and value weights.

Because Wang et al. (2024) do not provide hyperparameters and optimization configurations on the FGVC benchmark, we adopt a dropout rate of 0.5 and set the coefficient initialization value to 1.0

for MLAE on this benchmark. All other training settings—including the learning rate, weight decay, and other optimization configurations—are kept consistent with those used in our proposed method. For the VTAB-1K experiments, we follow the hyperparameters and optimizer settings reported in the original paper.

For the Commonsense Reasoning task, we reuse the FGVC hyperparameters for reproducing MLAE: a dropout rate of 0.5 and a coefficient initialization value of 1.0.

**Comparision with MLAE:** The results are summarized in the tables below. These results show that although MLAE achieves a 0.5% performance gain over DoRAN on FGVC, DoRAN surpasses MLAE by 1.7% on the VTAB-1K benchmark and further outperforms it on the language tasks by 0.91% for LLaMA-7B and 1.04% for LLaMA-13B. This demonstrates that DoRAN is competitive with state-of-the-art approaches including MLAE, while also exhibiting strong flexibility and applicability across multiple domains.

Table 9: Image Classification results on the FGVC datasets

| Method | CUB-200 -2011 | NABirds | Oxford Flowers | Stanford Dogs | Stanford Cars | Average |
|---|---|---|---|---|---|---|
| LoRA | 84.6 | 78.2 | 98.9 | 85.1 | 77.1 | 84.78 |
| DoRA | 87.3 | 80 | 99.1 | 87.6 | 81.9 | 87.18 |
| MLAE | **89.5** | **85.5** | 99.2 | **91.2** | **84.4** | **90** |
| **DoRAN** | 88.5 | 85.3 | **99.2** | 90.8 | 83.7 | 89.5 |

Table 10: Image Classification results on the VTAB-1K dataset

| | Natural | | | | | | | Specialized | | | | Structured | | | | | | | | |
|---|---|---|---|---|---|---|---|---|---|---|---|---|---|---|---|---|---|---|---|---|
| Method | CIFAR100 | Caltech101 | DTD | Flower102 | Pets | SVHN | Sun397 | Camelyon | EuroSAT | Resisc45 | Retinopathy | Clevr-Count | Clevr-Dist | DMLab | KITTI | dSpr-Loc | dSpr-ori | sNORB-Azim | sNORB-Ele | AVG |
| LoRA | 67.1 | 91.4 | 69.4 | 98.2 | 90.4 | 85.3 | 54 | 84.9 | 95.3 | 84.4 | 73.6 | 82.9 | 69.2 | 49.8 | 78.5 | 75.7 | 47.1 | 31 | **44** | 72.2 |
| DoRA | 67.9 | 90.4 | 70.6 | 99 | 90.2 | 89.6 | 54.6 | 83.9 | 95.5 | 85.3 | 75.9 | 80.8 | 69.8 | 50.5 | 80.9 | 79.1 | 47.7 | 32.5 | 39.6 | 72.8 |
| MLAE | 71.1 | 91.4 | **72.7** | 99.1 | **91.8** | 87.8 | 55.7 | 83 | 95.7 | 83.7 | 75.8 | 81.3 | 69.1 | 50.6 | 79.9 | 77.5 | 44.9 | 28.7 | 41.2 | 72.7 |
| **DoRAN** | 70.2 | **92.4** | 71.9 | **99.1** | 91.3 | **90.3** | 55 | **86.9** | **96.4** | **87.4** | **76.4** | **84** | 70 | **53.2** | 81.9 | **79.9** | **49.2** | 35.6 | 42 | **74.4** |

Table 11: Performance on the Commonsense Reasoning task

| Model | Method | #Params (%) | BoolQ | PIQA | SIQA | HellaSwag | WinoGrande | ARC-e | ARC-c | OBQA | Average |
|---|---|---|---|---|---|---|---|---|---|---|---|
| LLaMA-7B | LoRA | 0.25 | 67.2 | 79.4 | 76.6 | 78.3 | 78.4 | 77.1 | 61.5 | 74.2 | 74.09 |
| | DoRA | 0.25 | 67.22 | 79.98 | 76.66 | 80.66 | 79.72 | 79.5 | 61.01 | 74.8 | 74.94 |
| | MLAE | 0.25 | 69.02 | **81.18** | 77.02 | 78.8 | **80.51** | **80.6** | 63.05 | 78.6 | 76.1 |
| | **DoRAN** | 0.26 | **69.82** | 81.01 | **77.89** | **83.45** | 79.56 | 79.76 | **65.02** | **79.6** | **77.01** |
| LLaMA-13B | LoRA | 0.2 | 71.7 | 82.4 | 79.6 | **90.4** | **83.6** | 83.1 | 68.5 | 82.1 | 80.18 |
| | DoRA | 0.2 | **72.2** | 83.19 | **80.81** | 88.92 | 81.93 | 82.95 | **69.37** | 81 | 80.05 |
| | MLAE | 0.2 | 70.12 | 83.13 | 79.22 | 89.04 | 81.77 | 83.25 | 67.83 | 83.4 | 79.72 |
| | **DoRAN** | 0.21 | 71.8 | **84.5** | 80.6 | 89.79 | 83.19 | **83.29** | 68.69 | **84.2** | **80.76** |

**Future direction.** To further highlight the flexibility of DoRAN, we note that DoRAN can be naturally combined with MLAE. In MLAE, the stochastic masking mechanism is implemented as

$$\Delta W = M \odot \Lambda \odot \varepsilon$$

Equivalently, for each layer $l \in [L]$, the forward computation can be written as

$$Y = Y_{\text{pretrained}} + Y_{\text{MLAE}}, Y_{\text{MLAE}} = xA^{\top} \text{diag}(\text{Dropout}_p(\lambda))B$$

Importantly, MLAE only introduces a masking mechanism — it does not modify the architecture of the underlying low-rank matrices. This makes it straightforward to incorporate MLAE into DoRAN. A natural combination strategy is to first generate the low-rank matrices $A$ and $B$ using the DoRAN hypernetworks, and then apply MLAE's masked update within DoRAN's normalized forward pass:

$$Y_{\text{DoRAN + MLAE}} = x \frac{W_0 + W'}{\|W_0 + W'\| + \tau}, W' = A^\top \text{diag}(\text{Dropout}_p(\lambda))B$$

We believe this hybrid approach could leverage the strengths of both DoRAN and MLAE. Exploring this direction is promising and we leave a thorough investigation of its benefits for future work.

# D  PROOFS

## D.1  PROOF OF THEOREM 1

Firstly, we demonstrate that the following limit holds for any $r \geq 1$:

$$\lim_{\varepsilon \to 0} \inf_{G \in \mathcal{G}_{L'}(\Theta) : \mathcal{D}_{1,r}(G, G_*) \leq \varepsilon} \frac{\|f_G - f_{G_*}\|_{L^2(\mu)}}{\mathcal{D}_{1,r}(G, G_*)} = 0. \tag{8}$$

To prove this, we construct a mixing measure sequence $(G_n)_n$ that satisfies both $\mathcal{D}_{1,r}(G_n, G_*) \to 0$ and $\|f_{G_n} - f_{G_*}\|_{L^2(\mu)}/\mathcal{D}_{1,r}(G_n, G_*) \to 0$, as $n \to \infty$. Specifically, we consider the sequence of mixing measure with $L+1$ atoms, $G_n = \sum_{i=1}^{L+1} \exp(c_i^n) \delta_{(B_{Q,i}^n, A_{Q,i}^n, B_{V,i}^n, A_{V,i}^n)}$, where,

- $\exp(c_1^n) = \exp(c_2^n) = \frac{1}{2}\exp(c_1^*) + \frac{1}{2n^{r+1}}$ and $\exp(c_i^n) = \exp(c_{i-1}^*)$ for any $3 \leq i \leq L+1$;
- $B_{Q,1}^n = \left(1 + \frac{1}{n}\right)B_{Q,1}^* + \frac{1}{\sqrt{n}}S$, $B_{Q,2}^n = \left(1 + \frac{1}{n}\right)B_{Q,1}^* - \frac{1}{\sqrt{n}}S$ and $B_{Q,i}^n = B_{Q,i-1}^*$ for any $3 \leq i \leq L+1$;
- $A_{Q,1}^n = A_{Q,1}^* + \frac{1}{\sqrt{n}}T$, $A_{Q,2}^n = A_{Q,1}^* - \frac{1}{\sqrt{n}}T$ and $A_{Q,i}^n = A_{Q,i-1}^*$ for any $3 \leq i \leq L+1$;
- $B_{V,1}^n = B_{V,2}^n = B_{V,1}^*$ and $B_{V,i}^n = B_{V,i-1}^*$ for any $3 \leq i \leq L+1$,
- $A_{V,1}^n = A_{V,2}^n = A_{V,1}^*$ and $A_{V,i}^n = A_{V,i-1}^*$ for any $3 \leq i \leq L+1$,

where $S \in \mathbb{R}^{d \times r}$ and $T \in \mathbb{R}^{r \times d}$ are chosen to satisfy $ST = C_Q$ if rank of $C_Q$ is less than or equal $r$. We note that choosing $S$ and $T$ so that $ST = C_Q$ can be done easily using the singular vectors and singular values of $C_Q$. Then, if we have $ST = C_Q$, it is clear that,

$$B_{Q,1}^n A_{Q,1}^n + B_{Q,2}^n A_{Q,2}^n = 2\left(1 + \frac{1}{n}\right)B_{Q,1}^* A_{Q,1}^* + \frac{2}{n}C_Q.$$

This identity is necessary for the later part of our proof. Now, if rank of $C_Q$ is larger than $r$, we can write $C_Q$ as the sum of multiple matrices, let's say $C_{Q,1}$ and $C_{Q,2}$), each with rank less than or equal to $r$ and build the mixing measure similarly with more atoms, so that we can recover the sum $C_{Q,1} + C_{Q,2} = C_Q$. Just for this example, the goal is to have the following,

$$B_{Q,1}^n A_{Q,1}^n + B_{Q,2}^n A_{Q,2}^n = 2\left(1 + \frac{1}{2n}\right)B_{Q,1}^* A_{Q,1}^* + \frac{2}{n}C_{Q,1},$$

$$B_{Q,3}^n A_{Q,3}^n + B_{Q,4}^n A_{Q,4}^n = 2\left(1 + \frac{1}{2n}\right)B_{Q,1}^* A_{Q,1}^* + \frac{2}{n}C_{Q,2}.$$

We go back to the original chosen values above. Computing the loss function $\mathcal{D}_{1,r}(G_n, G_*)$ yields

$$\mathcal{D}_{1,r}(G_n, G_*) = \frac{1}{n^{r+1}} + \left[\exp(c_1^*) + \frac{1}{n^{r+1}}\right] \cdot \frac{1}{n^r}\|B_{Q,1}^*\|^r = \mathcal{O}(n^{-r}). \tag{9}$$

It can be seen that $\mathcal{D}_{1,r}(G_n, G_*) \to 0$ as $n \to \infty$.

Now we show that $\|f_{G_n} - f_{G_*}\|_{L^2(\mu)}/\mathcal{D}_{1,r}(G_n, G_*) \to 0$. Indeed, consider the below quantity,

$$T_n(X) := \left(\sum_{j=1}^{L} \exp\left(X^\top m_{Q,j} \frac{C_Q + B_{Q,j}^* A_{Q,j}^*}{\|C_Q + B_{Q,j}^* A_{Q,j}^*\|} C_K X + c_j^*\right)\right) \cdot [f_{G_n}(X) - f_{\bar{G}_*}(X)],$$

which can be decomposed as follows:

$$
T_n(\boldsymbol{X}) = \sum_{j=1}^{L} \sum_{i \in \mathcal{V}_j} \exp(c_i^n) \Bigg[ \exp\left( \boldsymbol{X}^\top m_{Q,j} \frac{\boldsymbol{C}_Q + \boldsymbol{B}_{Q,j}^n \boldsymbol{A}_{Q,j}^n}{\|\boldsymbol{C}_Q + \boldsymbol{B}_{Q,j}^n \boldsymbol{A}_{Q,j}^n\|} \boldsymbol{C}_K \boldsymbol{X} \right) \left( m_{V,j} \frac{\boldsymbol{C}_{V,j} + \boldsymbol{B}_{V,j}^n \boldsymbol{A}_{V,j}^n}{\|\boldsymbol{C}_{V,j} + \boldsymbol{B}_{V,j}^n \boldsymbol{A}_{V,j}^n\|} \right) \boldsymbol{X}
$$

$$
- \exp\left( \boldsymbol{X}^\top m_{Q,j}^* \frac{\boldsymbol{C}_Q + \boldsymbol{B}_{Q,j}^* \boldsymbol{A}_{Q,j}^*}{\|\boldsymbol{C}_Q + \boldsymbol{B}_{Q,j}^* \boldsymbol{A}_{Q,j}^*\|} \boldsymbol{C}_K \boldsymbol{X} \right) \left( m_{V,j} \frac{\boldsymbol{C}_V + \boldsymbol{B}_{V,j}^* \boldsymbol{A}_{V,j}^*}{\|\boldsymbol{C}_V + \boldsymbol{B}_{V,j}^* \boldsymbol{A}_{V,j}^*\|} \right) \boldsymbol{X} \Bigg]
$$

$$
- \sum_{j=1}^{L} \sum_{i \in \mathcal{V}_j} \exp(c_i^n) \Bigg[ \exp\left( \boldsymbol{X}^\top m_{Q,j} \frac{\boldsymbol{C}_Q + \boldsymbol{B}_{Q,j}^n \boldsymbol{A}_{Q,j}^n}{\|\boldsymbol{C}_Q + \boldsymbol{B}_{Q,j}^n \boldsymbol{A}_{Q,j}^n\|} \boldsymbol{C}_K \boldsymbol{X} \right)
$$

$$
- \exp\left( \boldsymbol{X}^\top m_{Q,j} \frac{\boldsymbol{C}_Q + \boldsymbol{B}_{Q,j}^* \boldsymbol{A}_{Q,j}^*}{\|\boldsymbol{C}_Q + \boldsymbol{B}_{Q,j}^* \boldsymbol{A}_{Q,j}^*\|} \boldsymbol{C}_K \boldsymbol{X} \right) \Bigg] f_{G_n}(\mathbb{X})
$$

$$
+ \sum_{j=1}^{L} \Big( \sum_{i \in \mathcal{V}_j} \exp(c_i^n) - \exp(c_j^*) \Big) \exp\left( \boldsymbol{X}^\top m_{Q,j} \frac{\boldsymbol{C}_Q + \boldsymbol{B}_{Q,j}^* \boldsymbol{A}_{Q,j}^*}{\|\boldsymbol{C}_Q + \boldsymbol{B}_{Q,j}^* \boldsymbol{A}_{Q,j}^*\|} \boldsymbol{C}_K \boldsymbol{X} \right) \Big[ m_{V,j} \frac{\boldsymbol{C}_V + \boldsymbol{B}_{V,j}^* \boldsymbol{A}_{V,j}^*}{\|\boldsymbol{C}_V + \boldsymbol{B}_{V,j}^* \boldsymbol{A}_{V,j}^*\|} \boldsymbol{X} - f_{G_n}(\boldsymbol{X}) \Big]
$$

$$
:= A_n(\boldsymbol{X}) - B_n(\boldsymbol{X}) + C_n(\boldsymbol{X}).
$$

It follows from the choices of $\boldsymbol{B}_{Q,i}^n, \boldsymbol{A}_{Q,i}^n, \boldsymbol{B}_{V,i}^n, \boldsymbol{A}_{V,i}^n$ and $c_i^n$ that

$$
A_n(\boldsymbol{X}) = \sum_{k=1}^{2} \frac{1}{2} \Big[ \exp(c_1^*) + \frac{1}{n^{r+1}} \Big] \left( m_{V,1} \frac{\boldsymbol{C}_V + \boldsymbol{B}_{V,1}^* \boldsymbol{A}_{V,1}^*}{\|\boldsymbol{C}_V + \boldsymbol{B}_{V,1}^* \boldsymbol{A}_{V,1}^*\|} \right) \boldsymbol{X}
$$

$$
\times \left( \exp\left( \boldsymbol{X}^\top m_{Q,k} \frac{\boldsymbol{C}_Q + \boldsymbol{B}_{Q,k}^n \boldsymbol{A}_{Q,k}^n}{\|\boldsymbol{C}_Q + \boldsymbol{B}_{Q,k}^n \boldsymbol{A}_{Q,k}^n\|} \boldsymbol{C}_K \boldsymbol{X} \right) - \exp\left( \boldsymbol{X}^\top m_{Q,k} \frac{\boldsymbol{C}_Q + \boldsymbol{B}_{Q,k}^* \boldsymbol{A}_{Q,k}^*}{\|\boldsymbol{C}_Q + \boldsymbol{B}_{Q,k}^* \boldsymbol{A}_{Q,k}^*\|} \boldsymbol{C}_K \boldsymbol{X} \right) \right)
$$

We denote $L(\boldsymbol{Z}) := \exp\left( \boldsymbol{X}^\top m_Q \frac{\boldsymbol{C}_Q + \boldsymbol{Z}}{\|\boldsymbol{C}_Q + \boldsymbol{Z}\|} \boldsymbol{C}_K \boldsymbol{X} \right)$. Then, by using first-order Taylor expansion, we have that,

$$
L(\boldsymbol{B}_{Q,1}^n \boldsymbol{A}_{Q,1}^n) - L(\boldsymbol{B}_{Q,1}^* \boldsymbol{A}_{Q,1}^*) = \langle \boldsymbol{B}_{Q,1}^n \boldsymbol{A}_{Q,1}^n - \boldsymbol{B}_{Q,1}^* \boldsymbol{A}_{Q,1}^*, \frac{\partial L}{\partial \boldsymbol{BA}}(\boldsymbol{B}_{Q,1}^* \boldsymbol{A}_{Q,1}^*) \rangle + R_1(\boldsymbol{X}),
$$

$$
L(\boldsymbol{B}_{Q,2}^n \boldsymbol{A}_{Q,2}^n) - L(\boldsymbol{B}_{Q,1}^* \boldsymbol{A}_{Q,1}^*) = \langle \boldsymbol{B}_{Q,2}^n \boldsymbol{A}_{Q,2}^n - \boldsymbol{B}_{Q,1}^* \boldsymbol{A}_{Q,1}^*, \frac{\partial L}{\partial \boldsymbol{BA}}(\boldsymbol{B}_{Q,1}^* \boldsymbol{A}_{Q,1}^*) \rangle + R_2(\boldsymbol{X}),
$$

where $R_1(\boldsymbol{X})$ and $R_2(\boldsymbol{X})$ are Taylor remainder such that $R_1/\mathcal{D}_1(G_n, G_*) \to 0$ and $R_2/\mathcal{D}_1(G_n, G_*) \to 0$ as $n \to \infty$. It follows based on the chosen $\boldsymbol{B}_{Q,1}^n, \boldsymbol{A}_{Q,1}^n, \boldsymbol{B}_{Q,2}^n, \boldsymbol{A}_{Q,2}^n$,

$$
L(\boldsymbol{B}_{Q,1}^n \boldsymbol{A}_{Q,1}^n) - L(\boldsymbol{B}_{Q,1}^* \boldsymbol{A}_{Q,1}^*) + L(\boldsymbol{B}_{Q,2}^n \boldsymbol{A}_{Q,2}^n) - L(\boldsymbol{B}_{Q,1}^* \boldsymbol{A}_{Q,1}^*)
$$

$$
= \frac{1}{n} \langle \boldsymbol{B}_{Q,1}^* \boldsymbol{A}_{Q,1}^* + \boldsymbol{C}_Q, \frac{\partial L}{\partial \boldsymbol{BA}}(\boldsymbol{B}_{Q,1}^* \boldsymbol{A}_{Q,1}^*) \rangle + R_1 + R_2,
$$

$$
= R_1 + R_2,
$$

where the inner product is 0 due to Lemma .... Therefore, we have that $A_n(\boldsymbol{X})/\mathcal{D}_1(G_n, G_*) \to 0$ as $n \to \infty$.

Additionally, it is similar to verify that $B_n(\boldsymbol{X})/\mathcal{D}_1(G_n, G_*) \to 0$, and $C_n(\boldsymbol{X}) = \mathcal{O}(n^{-(r+1)})$. Thus, $T_n(\boldsymbol{X})/\mathcal{D}_{1,r}(G_n, G_*) \to 0$ as $n \to \infty$ for almost every $\boldsymbol{X}$.

Since the term $\sum_{k=1}^{L} \exp\left( \boldsymbol{X}^\top m_{Q,k} \frac{\boldsymbol{C}_Q + \boldsymbol{B}_{Q,k}^* \boldsymbol{A}_{Q,k}^*}{\|\boldsymbol{C}_Q + \boldsymbol{B}_{Q,k}^* \boldsymbol{A}_{Q,k}^*\|} \boldsymbol{C}_K \boldsymbol{X} + c_k^* \right)$ is bounded, we have $[f_{G_n}(\boldsymbol{X}) - f_{G_*}(\boldsymbol{X})]/\mathcal{D}_{1,r}(G_n, G_*) \to 0$ for almost every $\mathbb{X}$. Therefore, we have

$$
\|f_{G_n} - f_{G_*}\|_{L^2(\mu)}/\mathcal{D}_{1,r}(G_n, G_*) \to 0
$$

as $n \to \infty$. We obtain the claim in equation (8).

**Step 2.** Now, we are ready to prove the desired result, i.e.,

$$
\inf_{G_n \in \mathcal{G}_{L'}(\Theta)} \sup_{G \in \mathcal{G}_{L'}(\Theta) \setminus \mathcal{G}_{L-1}(\Theta)} \mathbb{E}_{f_G}[\mathcal{D}_{1,r}(G_n, G)] \gtrsim n^{-1/2}. \tag{10}
$$

Since the noise variables $\varepsilon_i$ follow from the Gaussian distribution, we have that $Y_i | \boldsymbol{X}_i \sim \mathcal{N}(f_{G_*}(\boldsymbol{X}_i), \sigma^2)$ for all $i \in [n]$. For sufficiently small $\varepsilon > 0$ and a fixed constant $C_1 > 0$ which we will choose later, there exists a mixing measure $G'_* \in \mathcal{G}_{L'}(\Theta)$ such that $\mathcal{D}_{1,r}(G'_*, G_*) = 2\varepsilon$ and $\|f_{G'_*} - f_{G_*}\|_{L^2(\mu)} \leq C_1 \varepsilon$ thanks to the result in equation (8). Due to the Le Cam's lemma Yu (1997) and the fact that the Voronoi loss function $\mathcal{D}_{1,r}$ satisfies the weak triangle inequality, we can derive that

$$\inf_{G_n \in \mathcal{G}_{L'}(\Theta)} \sup_{G \in \mathcal{G}_{L'}(\Theta) \setminus \mathcal{G}_{L-1}(\Theta)} \mathbb{E}_{f_G}[\mathcal{D}_{1,r}(G_n, G)]$$

$$\gtrsim \frac{\mathcal{D}_{1,r}(G'_*, G_*)}{8} \exp(-n \mathbb{E}_{\boldsymbol{X} \sim \mu}[\mathrm{KL}(\mathcal{N}(f_{G'_*}(\boldsymbol{X}), \sigma^2), \mathcal{N}(f_{G_*}(\boldsymbol{X}), \sigma^2))])$$

$$\gtrsim \varepsilon \cdot \exp(-n \|f_{G'_*} - f_{G_*}\|_{L^2(\mu)}^2)$$

$$\gtrsim \varepsilon \cdot \exp(-C_1 n \varepsilon^2), \tag{11}$$

where the second inequality follows from the KL distance of two multivariate Gaussians, i.e.,

$$\mathrm{KL}(\mathcal{N}(f_{G'_*}(\boldsymbol{X}), \sigma^2), \mathcal{N}(f_{G_*}(\boldsymbol{X}), \sigma^2)) = \frac{(f_{G'_*}(\boldsymbol{X}) - f_{G_*}(\boldsymbol{X}))^2}{2\sigma^2}.$$

Let $\varepsilon = n^{-1/2}$, then we get that the RHS $\varepsilon \cdot \exp(-C_1 n \varepsilon^2) = n^{-1/2} \exp(-C_1)$. Thus, we achieve the desired minimax lower bound in equation (10).

**Lemma 1.** *Let* $L(\boldsymbol{Z}) := \exp\left(\boldsymbol{X}^\top m_Q \frac{\boldsymbol{C}_Q + \boldsymbol{Z}}{\|\boldsymbol{C}_Q + \boldsymbol{Z}\|} \boldsymbol{C}_K \boldsymbol{X}\right)$. *We have that,*

$$\langle \boldsymbol{C}_Q + \boldsymbol{Z}, \frac{\partial L}{\partial \boldsymbol{Z}} \rangle = 0.$$

Proof of Lemma 1: It follows from direct calculation,

$$\frac{\partial L}{\partial \boldsymbol{Z}} = m_Q \frac{\boldsymbol{X}(\boldsymbol{C}_K \boldsymbol{X})^\top}{\|\boldsymbol{C}_Q + \boldsymbol{Z}\|} - m_Q(\boldsymbol{X}^\top(\boldsymbol{C}_Q + \boldsymbol{Z})\boldsymbol{C}_K \boldsymbol{X}) \frac{\boldsymbol{C}_Q + \boldsymbol{Z}}{\|\boldsymbol{C}_Q + \boldsymbol{Z}\|^3}.$$

$$\langle \boldsymbol{C}_Q + \boldsymbol{Z}, \frac{\partial L}{\partial \boldsymbol{Z}} \rangle = \frac{m_Q}{\|\boldsymbol{C}_Q + \boldsymbol{Z}\|} \mathrm{Trace}\left((\boldsymbol{C}_Q + \boldsymbol{Z})^\top \boldsymbol{X} \boldsymbol{X}^\top \boldsymbol{C}_K^\top - \boldsymbol{X}^\top(\boldsymbol{C}_Q + \boldsymbol{Z})\boldsymbol{C}_K \boldsymbol{X}\right) = 0.$$

### D.2 PROOF OF THEOREM 2

Before going to the proof, without loss of generality, we can assume $\boldsymbol{W}_{1,j}, \boldsymbol{W}_{2,j}$ are identity matrices for each $j$. In particular, we may denote $\sigma_1(\boldsymbol{W}_1 \boldsymbol{A})$ as $\sigma_1(\boldsymbol{A})$ for an input matrix $\boldsymbol{A}$. Throughout this proof, we also assume without loss of generality that $C_{K,j} = I_d$ with a note that our techniques can be extended to the general setting of that matrix.

We first start with the following result regarding the convergence rate of the regression function estimation $f_{\widetilde{G}_n}$ to the true regression function $f_{\widetilde{G}_*}$:

**Proposition 1.** *Given the least square estimator* $\widetilde{G}_n$, *the convergence rate of the regression function estimator* $f_{\widetilde{G}_n}$ *to the true regression function* $f_{\widetilde{G}_*}$ *under the* $L^2(\mu)$ *norm is,*

$$\|f_{\widetilde{G}_n} - f_{\widetilde{G}_*}\|_{L^2(\mu)} = \mathcal{O}_P(\sqrt{\log(n)/n}). \tag{12}$$

Our goal now is to demonstrate the following inequality:

$$\inf_{\widetilde{G} \in \mathcal{G}_{L'}(\Theta)} \|f_{\widetilde{G}} - f_{\widetilde{G}_*}\|_{L^2(\mu)} / \mathcal{D}_2(\widetilde{G}, \widetilde{G}_*) > 0,$$

and since from Proposition 1, the rate of $\|f_{\widetilde{G}} - f_{\widetilde{G}_*}\|_{L^2(\mu)}$ is $\mathcal{O}_P(\sqrt{\log(n)/n})$, we deduce that $\mathcal{D}_2(\widetilde{G}, \widetilde{G}_*)$ is also $\mathcal{O}_P(\sqrt{\log(n)/n})$. We will prove this inequality by considering separately the local part when the Voronoi loss is small and the global part when the Voronoi loss is large. Before delving into the proof, we first state some essential assumptions on the activation function $\sigma_1$ and $\sigma_2$.

**Assumptions.** We impose the following assumptions on the activation functions $\sigma_1$ and $\sigma_2$:

*(A.1) (Algebraic Independence)* For any pair of matrices $(\boldsymbol{B}_1, \boldsymbol{A}_1)$ and $(\boldsymbol{B}_2, \boldsymbol{A}_2)$ such that $\sigma_2(\boldsymbol{B}_1)\sigma_1(\boldsymbol{A}_1) = \sigma_2(\boldsymbol{B}_2)\sigma_1(\boldsymbol{A}_2)$, then it follows that $\boldsymbol{B}_1 = \boldsymbol{B}_2$ and $\boldsymbol{A}_1 = \boldsymbol{A}_2$.

*(A.2) (Uniform Lipschitz)* Denote

$$\boldsymbol{F}(\boldsymbol{X}, \boldsymbol{A}, \boldsymbol{B}) := \exp\left(\boldsymbol{X}^\top m_{Q,k} \frac{\boldsymbol{C}_Q + \sigma_2(\boldsymbol{B})\sigma_1(\boldsymbol{A})}{\|\boldsymbol{C}_Q + \sigma_2(\boldsymbol{B})\sigma_1(\boldsymbol{A})\| + \tau_Q} \boldsymbol{X}\right) \left(m_{V,j}^* \frac{\boldsymbol{C}_V + \sigma_2(\boldsymbol{B})\sigma_1(\boldsymbol{A})}{\|\boldsymbol{C}_V + \sigma_2(\boldsymbol{B})\sigma_1(\boldsymbol{A})\| + \tau_V}\right) \boldsymbol{X},$$

then for any $\tau \in \{1, 2\}$,

$$\sum_{|\alpha|=\tau} \left\| \left(\frac{\partial^{|\alpha|}\boldsymbol{F}}{\partial \boldsymbol{A}^{\alpha_1}\partial \boldsymbol{B}^{\alpha_2}}(\mathbb{X}, \boldsymbol{A}_1, \boldsymbol{B}_1) - \frac{\partial^{|\alpha|}\boldsymbol{F}}{\partial \boldsymbol{A}^{\alpha_1}\partial \boldsymbol{B}^{\alpha_2}}(\mathbb{X}, \boldsymbol{A}_2, \boldsymbol{B}_2)\right)\gamma^\alpha \right\| \le C\|(\boldsymbol{A}_1, \boldsymbol{B}_1) - (\boldsymbol{A}_2, \boldsymbol{B}_2)\|^a \|\gamma\|^a,$$

for any vector $\gamma \in \mathbb{R}^{2dr}$ and for some positive constant $a$ and $C$ independent of input $\mathbb{X}$ and parameter $\boldsymbol{A}_1, \boldsymbol{B}_1, \boldsymbol{A}_2, \boldsymbol{B}_2$. We denote the index $\alpha = (\alpha_1, \alpha_2) \in \mathbb{N}^{r \times d} \times \mathbb{N}^{d \times r}$ and $\boldsymbol{A}^{\alpha_1}$ will return the entries of $\boldsymbol{A}$ at the position that the element of $\alpha_1$ is non-zero.

*(A.3) (Strong identifiability)* We denote,

$$\boldsymbol{M} := \frac{\boldsymbol{C}_Q + \sigma_2(\boldsymbol{B})\sigma_1(\boldsymbol{A})}{\|\boldsymbol{C}_Q + \sigma_2(\boldsymbol{B})\sigma_1(\boldsymbol{A})\| + \tau_Q},$$

$$\boldsymbol{N} := \frac{\boldsymbol{C}_V + \sigma_2(\boldsymbol{B})\sigma_1(\boldsymbol{A})}{\|\boldsymbol{C}_V + \sigma_2(\boldsymbol{B})\sigma_1(\boldsymbol{A})\| + \tau_V},$$

Then, for any non-positive integer $\ell$ and distinct matrices $\{(\boldsymbol{B}_j, \boldsymbol{A}_j)\}_{j \in [\ell]}$, the functions in the set below are linear independent for almost sure $\boldsymbol{X}$:

$$\begin{aligned}
&\Bigg\{ \left(\boldsymbol{X}^\top \frac{\partial \boldsymbol{M}_j^*}{\partial \boldsymbol{A}^{(u)}}\boldsymbol{X}\right)\boldsymbol{N}_j^*\boldsymbol{X}, \quad \left(\boldsymbol{X}^\top \frac{\partial \boldsymbol{M}_j^*}{\partial \boldsymbol{B}^{(u)}}\boldsymbol{X}\right)\boldsymbol{N}_j^*\boldsymbol{X}, \quad \frac{\partial \boldsymbol{N}_j^*}{\partial \boldsymbol{A}^{(u)}}\boldsymbol{X}, \quad \frac{\partial \boldsymbol{N}_j^*}{\partial \boldsymbol{B}^{(u)}}\boldsymbol{X}, \\
&\left(\boldsymbol{X}^\top \frac{\partial^2 \boldsymbol{M}_j^*}{\partial \boldsymbol{A}^{(u)}\partial \boldsymbol{A}^{(v)}}\boldsymbol{X}\right)\boldsymbol{N}_j^*\boldsymbol{X}, \quad \left(\boldsymbol{X}^\top \frac{\partial \boldsymbol{M}_j^*}{\partial \boldsymbol{A}^{(u)}}\boldsymbol{X}\boldsymbol{X}^\top \frac{\partial \boldsymbol{M}_j^*}{\partial \boldsymbol{A}^{(v)}}\boldsymbol{X}\right)\boldsymbol{N}_j^*\boldsymbol{X}, \\
&\left(\boldsymbol{X}^\top \frac{\partial^2 \boldsymbol{M}_j^*}{\partial \boldsymbol{B}^{(u)}\partial \boldsymbol{B}^{(v)}}\boldsymbol{X}\right)\boldsymbol{N}_j^*\boldsymbol{X}, \quad \left(\boldsymbol{X}^\top \frac{\partial \boldsymbol{M}_j^*}{\partial \boldsymbol{B}^{(u)}}\boldsymbol{X}\boldsymbol{X}^\top \frac{\partial \boldsymbol{M}_j^*}{\partial \boldsymbol{B}^{(v)}}\boldsymbol{X}\right)\boldsymbol{N}_j^*\boldsymbol{X}, \\
&\left(\boldsymbol{X}^\top \frac{\partial^2 \boldsymbol{M}_j^*}{\partial \boldsymbol{A}^{(u)}\partial \boldsymbol{B}^{(v)}}\boldsymbol{X}\right)\boldsymbol{N}_j^*\boldsymbol{X}, \quad \left(\boldsymbol{X}^\top \frac{\partial \boldsymbol{M}_j^*}{\partial \boldsymbol{A}^{(u)}}\boldsymbol{X}\boldsymbol{X}^\top \frac{\partial \boldsymbol{M}_j^*}{\partial \boldsymbol{B}^{(v)}}\boldsymbol{X}\right)\boldsymbol{N}_j^*\boldsymbol{X}, \\
&\frac{\partial^2 \boldsymbol{N}_j^*}{\partial \boldsymbol{A}^{(u)}\partial \boldsymbol{A}^{(v)}}\boldsymbol{X}, \quad \frac{\partial^2 \boldsymbol{N}_j^*}{\partial \boldsymbol{B}^{(u)}\partial \boldsymbol{B}^{(v)}}, \quad \frac{\partial^2 \boldsymbol{N}_j^*}{\partial \boldsymbol{A}^{(u)}\partial \boldsymbol{B}^{(v)}}, \\
&\left(\boldsymbol{X}^\top \frac{\partial \boldsymbol{M}_j^*}{\partial \boldsymbol{A}^{(u)}}\boldsymbol{X}\right)\frac{\partial \boldsymbol{N}_j^*}{\partial \boldsymbol{A}^{(v)}}\boldsymbol{X}, \quad \left(\boldsymbol{X}^\top \frac{\partial \boldsymbol{M}_j^*}{\partial \boldsymbol{B}^{(u)}}\boldsymbol{X}\right)\frac{\partial \boldsymbol{N}_j^*}{\partial \boldsymbol{B}^{(v)}}\boldsymbol{X}, \\
&\left(\boldsymbol{X}^\top \frac{\partial \boldsymbol{M}_j^*}{\partial \boldsymbol{A}^{(u)}}\boldsymbol{X}\right)\frac{\partial \boldsymbol{N}_j^*}{\partial \boldsymbol{B}^{(v)}}\boldsymbol{X}, \quad \left(\boldsymbol{X}^\top \frac{\partial \boldsymbol{M}_j^*}{\partial \boldsymbol{B}^{(u)}}\boldsymbol{X}\right)\frac{\partial \boldsymbol{N}_j^*}{\partial \boldsymbol{A}^{(v)}}\boldsymbol{X} \; : \; j \in [\ell]\Bigg\}
\end{aligned}$$

### D.2.1 LOCAL PART

For the local part, we will prove that

$$\lim_{\varepsilon \to 0} \inf_{\widetilde{G} \in \mathcal{G}_{L'}(\Theta):\mathcal{D}_2(\widetilde{G},\widetilde{G}_*) \le \varepsilon} \|f_{\widetilde{G}} - f_{\widetilde{G}_*}\|_{L^2(\mu)}/\mathcal{D}_2(\widetilde{G}, \widetilde{G}_*) > 0.$$

We will prove this by contradiction: assume that the above claim does not hold. This means we can find a sequence of mixing measures $\widetilde{G}_n := \sum_{j=1}^{L'} \exp(c_{n,j})\delta_{\boldsymbol{B}_{n,j}\boldsymbol{A}_{n,j}}$ in $\widetilde{\mathcal{G}}_{L'}(\Theta)$ such that

$$\begin{cases} \mathcal{D}_2(\widetilde{G}_n, \widetilde{G}_*) \to 0, \\ \|f_{\widetilde{G}_n} - f_{\widetilde{G}_*}\|_{L^2(\mu)}/\mathcal{D}_2(\widetilde{G}_n, \widetilde{G}_*) \to 0. \end{cases}$$

as $n \to \infty$. We denote $\mathcal{V}_j^n := \mathcal{V}_j(\widetilde{G}_n)$ as a Voronoi cell of $\widetilde{G}_n$ generated by the $j$-th components of $\widetilde{G}_*$. Without loss of generality, we may assume that those Voronoi cells do not depend on the sample size, i.e., $\mathcal{V}_j = \mathcal{V}_j^n$. Therefore, the Voronoi loss can be rewritten as follows:

$$
\mathcal{D}_2(\widetilde{G}_n, \widetilde{G}_*) = \sum_{j=1}^{L} \Big| \sum_{i \in \mathcal{V}_j} \exp(c_{n,i}) - \exp(c_{j'}^*) \Big| + \sum_{j \in [L]: |\mathcal{V}_j| = 1} \sum_{i \in \mathcal{V}_j} \exp(c_{n,i}) (\|\Delta \boldsymbol{B}_{n,ij}\| + \|\Delta \boldsymbol{A}_{n,ij}\|)
$$
$$
+ \sum_{j \in [L]: |\mathcal{V}_j| > 1} \sum_{i \in \mathcal{V}_j} \exp(c_{n,i}) (\|\Delta \boldsymbol{B}_{n,ij}\|^2 + \|\Delta \boldsymbol{A}_{n,ij}\|^2),
$$

where $\Delta \boldsymbol{B}_{n,ij} := \boldsymbol{B}_{n,i} - \boldsymbol{B}_{j'}^*$ and $\Delta \boldsymbol{A}_{n,ij} := \boldsymbol{A}_{n,i} - \boldsymbol{A}_{j'}^*$ for all $i \in \mathcal{V}_{j'}$ and $j \in [L]$.

Since $\mathcal{D}_2(\widetilde{G}_n, \widetilde{G}_*) \to 0$, we have $\sum_{i \in \mathcal{V}_j} \exp(c_{n,i}) \to \exp(c_j^*)$, $\boldsymbol{B}_{n,i} \to \boldsymbol{B}_j^*$, and $\boldsymbol{A}_{n,i} \to \boldsymbol{A}_j^*$ for any $i \in \mathcal{V}_j$ and $j \in [L]$. Now, the proof of the local part can be divided into three main steps as follows:

**Step 1 - Decompose the difference between regression functions.**

Recall the notation,

$$
\boldsymbol{M} := \frac{\boldsymbol{C}_Q + \sigma_2(\boldsymbol{B})\sigma_1(\boldsymbol{A})}{\|\boldsymbol{C}_Q + \sigma_2(\boldsymbol{B})\sigma_1(\boldsymbol{A})\| + \tau_Q},
$$
$$
\boldsymbol{N} := \frac{\boldsymbol{C}_V + \sigma_2(\boldsymbol{B})\sigma_1(\boldsymbol{A})}{\|\boldsymbol{C}_V + \sigma_2(\boldsymbol{B})\sigma_1(\boldsymbol{A})\| + \tau_V},
$$

and their subscripts will follow the subscripts of the matrices $\boldsymbol{A}, \boldsymbol{B}$.

First, we define

$$
T_n(\boldsymbol{X}) := \left( \sum_{k=1}^{L} \exp\left( \boldsymbol{X}^\top m_{Q,k} \underbrace{\frac{\boldsymbol{C}_Q + \sigma_2(\boldsymbol{B}_k^*)\sigma_1(\boldsymbol{A}_k^*)}{\|\boldsymbol{C}_Q + \sigma_2(\boldsymbol{B}_k^*)\sigma_1(\boldsymbol{A}_k^*)\| + \tau_Q}}_{\boldsymbol{M}_k^*} \boldsymbol{X} + c_k^* \right) \right) \cdot [f_{\widetilde{G}_n}(\boldsymbol{X}) - f_{\widetilde{G}_*}(\boldsymbol{X})].
$$

Then, we can decompose the function $T_n(\mathbb{X})$ as follows:

$$
T_n(\boldsymbol{X}) = \sum_{j=1}^{L} \sum_{i \in \mathcal{V}_j} \exp(c_{n,i}) \Big[ \exp\left( \boldsymbol{X}^\top m_{Q,j} \boldsymbol{M}_{n,j} \boldsymbol{X} \right) (m_{V,j} \boldsymbol{N}_{n,j}) \boldsymbol{X} - \exp(\mathbb{X}^\top m_{Q,j} \boldsymbol{M}_j^* \mathbb{X})(m_{V,j} \boldsymbol{N}_j^*)\mathbb{X} \Big]
$$
$$
- \sum_{j=1}^{L} \sum_{i \in \mathcal{V}_j} \exp(c_{n,i}) \Big[ \exp\left( \boldsymbol{X}^\top m_{Q,j} \boldsymbol{M}_{n,j} \boldsymbol{X} \right) - \exp\left( \boldsymbol{X}^\top m_{Q,j} \boldsymbol{M}_j^* \boldsymbol{X} \right) \Big] f_{\widetilde{G}_n}(\boldsymbol{X})
$$
$$
+ \sum_{j=1}^{L} \Big( \sum_{i \in \mathcal{V}_j} \exp(c_{n,i}) - \exp(c_j^*) \Big) \exp\left( \boldsymbol{X}^\top m_{Q,j} \boldsymbol{M}_j^* \boldsymbol{X} \right) \Big[ (m_{V,j} \boldsymbol{N}_j^*)\boldsymbol{X} - f_{\widetilde{G}_n}(\boldsymbol{X}) \Big]
$$
$$
:= \widetilde{A}_n(\boldsymbol{X}) - \widetilde{B}_n(\boldsymbol{X}) + \widetilde{C}_n(\boldsymbol{X}). \tag{13}
$$

**Decompose $\widetilde{A}_n(\boldsymbol{X})$.** We denote,

$$
\widetilde{U}(\boldsymbol{X}; \boldsymbol{B}, \boldsymbol{A}) := \exp\left( \boldsymbol{X}^\top m_Q \frac{\boldsymbol{C}_Q + \sigma_2(\boldsymbol{B})\sigma_1(\boldsymbol{A})}{\|\boldsymbol{C}_Q + \sigma_2(\boldsymbol{B})\sigma_1(\boldsymbol{A})\| + \tau_Q} \boldsymbol{X} \right)
$$
$$
\widetilde{V}(\boldsymbol{X}; \boldsymbol{B}, \boldsymbol{A}) := \left( m_V \frac{\boldsymbol{C}_V + \sigma_2(\boldsymbol{B})\sigma_1(\boldsymbol{A})}{\|\boldsymbol{C}_V + \sigma_2(\boldsymbol{B})\sigma_1(\boldsymbol{A})\| + \tau_V} \right) \boldsymbol{X}
$$
$$
\widetilde{F}(\boldsymbol{X}; \boldsymbol{B}, \boldsymbol{A}) := \widetilde{U}(\boldsymbol{X}; \boldsymbol{B}, \boldsymbol{A})\widetilde{V}(\boldsymbol{X}; \boldsymbol{B}, \boldsymbol{A}),
$$

and for brevity of notation, we will use $\tilde{U}$ instead of $\widetilde{U}(\boldsymbol{X}; \boldsymbol{B}, \boldsymbol{A})$ with its subscript follows the subscripts of $\{\boldsymbol{B}, \boldsymbol{A}\}$, similarly for $\tilde{V}$ and $\widetilde{F}$.

We decompose $\widetilde{A}_n(\boldsymbol{X})$ based on the number of element in the Voronoi cells as follows:

$$\widetilde{A}_n(\boldsymbol{X}) = \sum_{j:|\mathcal{V}_j|=1}\sum_{i\in\mathcal{V}_j}\exp(c_{n,i})\Big[\widetilde{F}_{n,i}-\widetilde{F}_j^*\Big] + \sum_{j:|\mathcal{V}_j|>1}\sum_{i\in\mathcal{V}_j}\exp(c_{n,i})\Big[\widetilde{F}_{n,i}-\widetilde{F}_j^*\Big]$$

$$:= \widetilde{A}_{n,1}(\boldsymbol{X}) + \widetilde{A}_{n,2}(\boldsymbol{X}).$$

We apply the first-order Taylor expansion to $\widetilde{U}$ and $\widetilde{V}$,

$$\widetilde{U}_{n,i} = \widetilde{U}_j^* + \sum_{|\alpha|=1}(\Delta\boldsymbol{A}_{n,ij})^{\alpha_1}(\Delta\boldsymbol{B}_{n,ij})^{\alpha_2}\frac{\partial^{|\alpha|}\widetilde{U}}{\partial\boldsymbol{A}^{\alpha_1}\partial\boldsymbol{B}^{\alpha_2}}(\boldsymbol{X};\boldsymbol{B}_j^*,\boldsymbol{A}_j^*) + \widetilde{R}_{ij,1}(\boldsymbol{X}),$$

$$\widetilde{V}_{n,i} = \widetilde{V}_j^* + \sum_{|\alpha|=1}(\Delta\boldsymbol{A}_{n,ij})^{\alpha_1}(\Delta\boldsymbol{B}_{n,ij})^{\alpha_2}\frac{\partial^{|\alpha|}\widetilde{V}}{\partial\boldsymbol{A}^{\alpha_1}\partial\boldsymbol{B}^{\alpha_2}}(\boldsymbol{X};\boldsymbol{B}_j^*,\boldsymbol{A}_j^*) + \widetilde{R}_{ij,2}(\boldsymbol{X}),$$

for any $i$ and $j$ such that $i\in\mathcal{V}_j$ and $|\mathcal{V}_j|=1$. Here, the functions $\widetilde{R}_{ij,1}(\mathbb{X})$ and $\widetilde{R}_{ij,2}(\mathbb{X})$ denote the Taylor remainders. Plugging the above identities into $\widetilde{A}_{n,1}$ leads to

$$\widetilde{A}_{n,1}(\boldsymbol{X}) = \sum_{j:|\mathcal{V}_j|=1}\sum_{i\in\mathcal{V}_j}\frac{\exp(c_{n,i})}{\alpha!}\sum_{|\alpha|=1}\Bigg\{(\Delta\boldsymbol{A}_{n,ij})^{\alpha_1}(\Delta\boldsymbol{B}_{n,ij})^{\alpha_2}\frac{\partial^{|\alpha|}\widetilde{U}}{\partial\boldsymbol{A}^{\alpha_1}\partial\boldsymbol{B}^{\alpha_2}}(\boldsymbol{X};\boldsymbol{B}_j^*,\boldsymbol{A}_j^*)\widetilde{V}_j^*$$

$$+ (\Delta\boldsymbol{A}_{n,ij})^{\alpha_1}(\Delta\boldsymbol{B}_{n,ij})^{\alpha_2}\frac{\partial^{|\alpha|}\widetilde{V}}{\partial\boldsymbol{A}^{\alpha_1}\partial\boldsymbol{B}^{\alpha_2}}(\boldsymbol{X};\boldsymbol{B}_j^*,\boldsymbol{A}_j^*)\widetilde{U}_j^*\Bigg\} + \widetilde{R}_{n,1}(\boldsymbol{X})$$

$$= \sum_{j:|\mathcal{V}_j|=1}\sum_{|\alpha|=1}\Bigg\{\widetilde{M}_{n,j,\alpha_1,\alpha_2}\frac{\partial^{|\alpha|}\widetilde{U}}{\partial\boldsymbol{A}^{\alpha_1}\partial\boldsymbol{B}^{\alpha_2}}(\boldsymbol{X};\boldsymbol{B}_j^*,\boldsymbol{A}_j^*)\widetilde{V}_j^* + \widetilde{M}_{n,j,\alpha_1,\alpha_2}\frac{\partial^{|\alpha|}\widetilde{V}}{\partial\boldsymbol{A}^{\alpha_1}\partial\boldsymbol{B}^{\alpha_2}}(\boldsymbol{X};\boldsymbol{B}_j^*,\boldsymbol{A}_j^*)\widetilde{U}_j^*\Bigg\}$$

$$+ \widetilde{R}_{n,1}(\mathbb{X})$$

where $\alpha = (\alpha_1,\alpha_2) \in (\mathbb{N}^{r\times d},\mathbb{N}^{d\times r})$ and the function $\widetilde{R}_{n,1}(\mathbb{X})$ satisfies that $\widetilde{R}_{n,1}(\mathbb{X})/\mathcal{D}_2(\widetilde{G}_n,G_*)\to 0$. This fact is due to the uniform Lipschitz assumption of the function $\widetilde{F}$. The coefficients $\widetilde{M}_{n,j,\alpha_1,\alpha_2}$ are given by:

$$\widetilde{M}_{n,j,\alpha_1,\alpha_2} = \sum_{i\in\mathcal{V}_j}\frac{\exp(c_{n,i})}{\alpha!}(\Delta\boldsymbol{A}_{n,ij})^{\alpha_1}(\Delta\boldsymbol{B}_{n,ij})^{\alpha_2},$$

for any $|\alpha|=1$.

Moving to the function $\widetilde{A}_{n,2}(\boldsymbol{X})$, we perform the Taylor expansion up to the second order of $\widetilde{U}$ and $\widetilde{V}$ then plugging in $\widetilde{A}_{n,2}$ yields,

$$\widetilde{A}_{n,2}(\mathbb{X}) = \sum_{j:|\mathcal{V}_j|>1}\sum_{1\leq|\alpha|\leq2}\Bigg\{\widetilde{M}_{n,j,\alpha_1,\alpha_2}\frac{\partial^{|\alpha|}\widetilde{U}}{\partial\boldsymbol{A}^{\alpha_1}\partial\boldsymbol{B}^{\alpha_2}}(\boldsymbol{X};\boldsymbol{B}_j^*,\boldsymbol{A}_j^*)\widetilde{V}_j^* + \widetilde{M}_{n,j,\alpha_1,\alpha_2}\frac{\partial^{|\alpha|}\widetilde{V}}{\partial\boldsymbol{A}^{\alpha_1}\partial\boldsymbol{B}^{\alpha_2}}(\boldsymbol{X};\boldsymbol{B}_j^*,\boldsymbol{A}_j^*)\widetilde{U}_j^*\Bigg\}$$

$$+ \sum_{|\alpha|=1,|\beta|=1}\widetilde{M}_{n,j,\alpha_1,\alpha_2,\beta_1,\beta_2}\frac{\partial^{|\alpha|}\widetilde{U}}{\partial\boldsymbol{A}^{\alpha_1}\partial\boldsymbol{B}^{\alpha_2}}(\boldsymbol{X};\boldsymbol{B}_j^*,\boldsymbol{A}_j^*)\frac{\partial^{|\alpha|}\widetilde{V}}{\partial\boldsymbol{A}^{\beta_1}\partial\boldsymbol{B}^{\beta_2}}(\boldsymbol{X};\boldsymbol{B}_j^*,\boldsymbol{A}_j^*)$$

$$+ \widetilde{R}_{n,2}(\mathbb{X})$$

where the remainder $\widetilde{R}_{n,2}(\mathbb{X})$ satisfies that $\widetilde{R}_{n,2}(\mathbb{X})/\mathcal{D}_2(\widetilde{G}_n,G_*)) \to 0$. The coefficients $\widetilde{M}_{n,j,\alpha_1,\alpha_2}$ and $\widetilde{M}_{n,j,\alpha_1,\alpha_2,\beta_1,\beta_2}$ take the following forms:

$$\widetilde{M}_{n,j,\alpha_1,\alpha_2} = \sum_{i\in\mathcal{V}_j}\frac{\exp(c_{n,i})}{\alpha!}(\Delta\boldsymbol{A}_{n,ij})^{\alpha_1}(\Delta\boldsymbol{B}_{n,ij})^{\alpha_2},$$

for any $|\alpha|=2$ and

$$\widetilde{M}_{n,j,\alpha_1,\alpha_2,\beta_1,\beta_2} = \sum_{i\in\mathcal{V}_j}\frac{\exp(c_{n,i})}{\alpha!\beta!}(\Delta\boldsymbol{A}_{n,ij})^{\alpha_1+\alpha_2}(\Delta\boldsymbol{B}_{n,ij})^{\beta_1+\beta_2},$$

for any $|\alpha| = |\beta| = 1$.

The partial derivatives of $\widetilde{U}$ and $\widetilde{V}$ can be calculated as following (recall the $\boldsymbol{M}$, $\boldsymbol{N}$ term appear in $\widetilde{U}$ and $\widetilde{V}$):

$$\frac{\partial \widetilde{U}}{\partial \boldsymbol{A}^{(u)}} = \exp\left(\boldsymbol{X}^\top m_Q \boldsymbol{M} \boldsymbol{X}\right) \boldsymbol{X}^\top m_Q \frac{\partial \boldsymbol{M}}{\partial (\boldsymbol{A})^{(u)}} \boldsymbol{X}, \frac{\partial \widetilde{U}}{\partial \boldsymbol{B}^{(u)}} = \exp\left(\boldsymbol{X}^\top m_Q \boldsymbol{M} \boldsymbol{X}\right) \boldsymbol{X}^\top m_Q \frac{\partial \boldsymbol{M}}{\partial (\boldsymbol{B})^{(u)}} \boldsymbol{X},$$

$$\frac{\partial^2 \widetilde{U}}{\partial \boldsymbol{A}^{(u)} \partial \boldsymbol{A}^{(v)}} = \exp(\boldsymbol{X}^\top m_Q \boldsymbol{M} \boldsymbol{X}) m_Q^2 \left[ \left( \boldsymbol{X}^\top \frac{\partial^2 \boldsymbol{M}}{\partial (\boldsymbol{A})^{(u)} \partial (\boldsymbol{A})^{(v)}} \boldsymbol{X} \right) + \left( \boldsymbol{X}^\top \frac{\partial \boldsymbol{M}}{\partial (\boldsymbol{A})^{(u)}} \boldsymbol{X} \right) \left( \boldsymbol{X}^\top \frac{\partial \boldsymbol{M}}{\partial (\boldsymbol{A})^{(v)}} \boldsymbol{X} \right) \right],$$

$$\frac{\partial^2 \widetilde{U}}{\partial \boldsymbol{B}^{(u)} \partial \boldsymbol{B}^{(v)}} = \exp(\boldsymbol{X}^\top m_Q \boldsymbol{M} \boldsymbol{X}) m_Q^2 \left[ \left( \boldsymbol{X}^\top \frac{\partial^2 \boldsymbol{M}}{\partial (\boldsymbol{B})^{(u)} \partial (\boldsymbol{B})^{(v)}} \boldsymbol{X} \right) + \left( \boldsymbol{X}^\top \frac{\partial \boldsymbol{M}}{\partial (\boldsymbol{B})^{(u)}} \boldsymbol{X} \right) \left( \boldsymbol{X}^\top \frac{\partial \boldsymbol{M}}{\partial (\boldsymbol{B})^{(v)}} \boldsymbol{X} \right) \right],$$

$$\frac{\partial^2 \widetilde{U}}{\partial \boldsymbol{A}^{(u)} \partial \boldsymbol{B}^{(v)}} = \exp(\boldsymbol{X}^\top m_Q \boldsymbol{M} \boldsymbol{X}) m_Q^2 \left[ \left( \boldsymbol{X}^\top \frac{\partial^2 \boldsymbol{M}}{\partial (\boldsymbol{A})^{(u)} \partial (\boldsymbol{B})^{(v)}} \boldsymbol{X} \right) + \left( \boldsymbol{X}^\top \frac{\partial \boldsymbol{M}}{\partial (\boldsymbol{A})^{(u)}} \boldsymbol{X} \right) \left( \boldsymbol{X}^\top \frac{\partial \boldsymbol{M}}{\partial (\boldsymbol{B})^{(v)}} \boldsymbol{X} \right) \right],$$

$$\frac{\partial \widetilde{V}}{\partial \boldsymbol{A}^{(u)}} = m_V \frac{\partial \boldsymbol{N}}{\partial \boldsymbol{A}^{(u)}} \boldsymbol{X}, \quad \frac{\partial \widetilde{V}}{\partial \boldsymbol{B}^{(u)}} = m_V \frac{\partial \boldsymbol{N}}{\partial \boldsymbol{B}^{(u)}} \boldsymbol{X},$$

$$\frac{\partial^2 \widetilde{V}}{\partial \boldsymbol{A}^{(u)} \partial \boldsymbol{A}^{(v)}} = m_V \frac{\partial^2 \boldsymbol{N}}{\partial \boldsymbol{A}^{(u)} \partial \boldsymbol{A}^{(v)}} \boldsymbol{X}, \quad \frac{\partial^2 \widetilde{V}}{\partial \boldsymbol{B}^{(u)} \partial \boldsymbol{B}^{(v)}} = m_V \frac{\partial^2 \boldsymbol{N}}{\partial \boldsymbol{B}^{(u)} \partial \boldsymbol{B}^{(v)}} \boldsymbol{X}, \frac{\partial^2 \widetilde{V}}{\partial \boldsymbol{A}^{(u)} \partial \boldsymbol{B}^{(v)}} = m_V \frac{\partial^2 \boldsymbol{N}}{\partial \boldsymbol{A}^{(u)} \partial \boldsymbol{B}^{(v)}} \boldsymbol{X}.$$

Plugging these terms into $\widetilde{A}_{n,1}$ and $\widetilde{A}_{n,2}$, we obtain that,

$$
\widetilde{A}_{n,1}(\boldsymbol{X}) = \sum_{j:|\mathcal{V}_j|=1} \left\{ \sum_{|\alpha|=1} \left\{ \widetilde{M}_{n,j,\alpha_1,\alpha_2} \exp(m_{Q,j} \boldsymbol{X}^\top \boldsymbol{M}_j^* \boldsymbol{X}) m_{Q,j} \boldsymbol{X}^\top \frac{\partial \boldsymbol{M}_j^*}{\partial \boldsymbol{A}^{\alpha_1} \partial \boldsymbol{B}^{\alpha_2}} \boldsymbol{X} \widetilde{V}_j^* \right. \right.
$$

$$
+ \widetilde{M}_{n,j,\alpha_1,\alpha_2} m_{V,j} \frac{\partial \boldsymbol{N}_j^*}{\partial \boldsymbol{A}^{\alpha_1} \boldsymbol{B}^{\alpha_2}} \boldsymbol{X} \widetilde{U}_j \right\} + \widetilde{R}_{n,1}(\boldsymbol{X})
$$

$$
= \sum_{j:|\mathcal{V}_j|=1} \exp(m_{Q,j} \boldsymbol{X}^\top \boldsymbol{M}_j^* \boldsymbol{X}) \left[ \sum_{u=(1,1)}^{(d,d)} \widetilde{M}_{n,j,u,0} \left( m_{Q,j} \boldsymbol{X}^\top \frac{\partial \boldsymbol{M}_j^*}{\partial \boldsymbol{A}^u} \boldsymbol{X} \right) m_{V,j} \boldsymbol{N}_j^* \boldsymbol{X} \right.
$$

$$
+ \sum_{u=(1,1)}^{(d,d)} \widetilde{M}_{n,j,0,u} \left( m_{Q,j} \boldsymbol{X}^\top \frac{\partial \boldsymbol{M}_j^*}{\partial \boldsymbol{B}^u} \boldsymbol{X} \right) m_{V,j} \boldsymbol{N}_j^* \boldsymbol{X} + \sum_{u=(1,1)}^{(d,d)} \widetilde{M}_{n,j,u,0} m_{V,j} \frac{\partial \boldsymbol{N}_j^*}{\partial \boldsymbol{A}^u} \boldsymbol{X} + \widetilde{M}_{n,j,0,u} m_{V,j} \frac{\partial \boldsymbol{N}_j^*}{\partial \boldsymbol{B}^u} \boldsymbol{X} \right] + \widetilde{R}_{n,1}(\boldsymbol{X})
$$

$$
= \sum_{j:|\mathcal{V}_j|=1} \exp(m_{Q,j} \boldsymbol{X}^\top \boldsymbol{M}_j^* \boldsymbol{X}) \left[ \sum_{u=(1,1)}^{(d,d)} \left( \boldsymbol{X}^\top \frac{\partial \boldsymbol{M}_j^*}{\partial \boldsymbol{A}^u} \boldsymbol{X} \right) \boldsymbol{N}_j^* \boldsymbol{X} \times \widetilde{M}_{n,j,u,0} m_{Q,j} m_{V,j} \right.
$$

$$
+ \sum_{u=(1,1)}^{(d,d)} \left( \boldsymbol{X}^\top \frac{\partial \boldsymbol{M}_j^*}{\partial \boldsymbol{B}^u} \boldsymbol{X} \right) \boldsymbol{N}_j^* \boldsymbol{X} \times \widetilde{M}_{n,j,0,u} m_{Q,j} m_{V,j} + \sum_{u=(1,1)}^{(d,d)} \frac{\partial \boldsymbol{N}_j^*}{\partial \boldsymbol{A}^u} \boldsymbol{X} \times \widetilde{M}_{n,j,u,0} m_{V,j} + \frac{\partial \boldsymbol{N}_j^*}{\partial \boldsymbol{B}^u} \boldsymbol{X} \times \widetilde{M}_{n,j,0,u} m_{V,j} \right]
$$

$$
+ \widetilde{R}_{n,1}(\boldsymbol{X}),
$$

$$
\widetilde{A}_{n,2}(\boldsymbol{X}) = \sum_{j:|\mathcal{V}_j|>1} \exp(m_{Q,j} \boldsymbol{X}^\top \boldsymbol{M}_j^* \boldsymbol{X}) \Bigg[
$$

$$
\sum_u \left( \boldsymbol{X}^\top \frac{\partial \boldsymbol{M}_j^*}{\partial \boldsymbol{A}^{(u)}} \boldsymbol{X} \right) \boldsymbol{N}_j^* \boldsymbol{X} \widetilde{M}_{n,j,u,0} m_{Q,j} m_{V,j} + \sum_u \left( \boldsymbol{X}^\top \frac{\partial \boldsymbol{M}_j^*}{\partial \boldsymbol{B}^{(u)}} \boldsymbol{X} \right) \boldsymbol{N}_j^* \boldsymbol{X} \widetilde{M}_{n,j,0,u} m_{Q,j} m_{V,j}
$$

$$
+ \sum_u \frac{\partial \boldsymbol{N}_j^*}{\partial \boldsymbol{A}^{(u)}} \boldsymbol{X} \widetilde{M}_{n,j,u,0} m_{V,j} + \sum_u \frac{\partial \boldsymbol{N}_j^*}{\partial \boldsymbol{B}^{(u)}} \boldsymbol{X} \widetilde{M}_{n,j,0,u} m_{V,j}
$$

$$
+ \sum_{u,v} \left( \boldsymbol{X}^\top \frac{\partial^2 \boldsymbol{M}_j^*}{\partial \boldsymbol{A}^{(u)} \partial \boldsymbol{A}^{(v)}} \boldsymbol{X} \right) \boldsymbol{N}_j^* \boldsymbol{X} \widetilde{M}_{n,j,u+v,0} m_{Q,j} m_{V,j}
$$

$$
+ \sum_{u,v} \left( \boldsymbol{X}^\top \frac{\partial \boldsymbol{M}_j^*}{\partial \boldsymbol{A}^{(u)}} \boldsymbol{X} \boldsymbol{X}^\top \frac{\partial \boldsymbol{M}_j^*}{\partial \boldsymbol{A}^{(v)}} \boldsymbol{X} \right) \boldsymbol{N}_j^* \boldsymbol{X} \widetilde{M}_{n,j,u+v,0} m_{Q,j} m_{V,j}
$$

$$
+ \sum_{u,v} \left( \boldsymbol{X}^\top \frac{\partial^2 \boldsymbol{M}_j^*}{\partial \boldsymbol{B}^{(u)} \partial \boldsymbol{B}^{(v)}} \boldsymbol{X} \right) \boldsymbol{N}_j^* \boldsymbol{X} \widetilde{M}_{n,j,0,u+v} m_{Q,j} m_{V,j}
$$

$$
+ \sum_{u,v} \left( \boldsymbol{X}^\top \frac{\partial \boldsymbol{M}_j^*}{\partial \boldsymbol{B}^{(u)}} \boldsymbol{X} \boldsymbol{X}^\top \frac{\partial \boldsymbol{M}_j^*}{\partial \boldsymbol{B}^{(v)}} \boldsymbol{X} \right) \boldsymbol{N}_j^* \boldsymbol{X} \widetilde{M}_{n,j,0,u+v} m_{Q,j} m_{V,j}
$$

$$
+ \sum_{u,v} \left( \boldsymbol{X}^\top \frac{\partial^2 \boldsymbol{M}_j^*}{\partial \boldsymbol{A}^{(u)} \partial \boldsymbol{B}^{(v)}} \boldsymbol{X} \right) \boldsymbol{N}_j^* \boldsymbol{X} \widetilde{M}_{n,j,u,v} m_{Q,j} m_{V,j} + \sum_{u,v} \left( \boldsymbol{X}^\top \frac{\partial \boldsymbol{M}_j^*}{\partial \boldsymbol{A}^{(u)}} \boldsymbol{X} \boldsymbol{X}^\top \frac{\partial \boldsymbol{M}_j^*}{\partial \boldsymbol{B}^{(v)}} \boldsymbol{X} \right) \boldsymbol{N}_j^* \boldsymbol{X} \widetilde{M}_{n,j,u,v} m_{Q,j} m_{V,j}
$$

$$
+ \sum_{u,v} \frac{\partial^2 \boldsymbol{N}_j^*}{\partial \boldsymbol{A}^{(u)} \partial \boldsymbol{A}^{(v)}} \boldsymbol{X} \widetilde{M}_{n,j,u+v,0} m_{V,j} + \sum_{u,v} \frac{\partial^2 \boldsymbol{N}_j^*}{\partial \boldsymbol{B}^{(u)} \partial \boldsymbol{B}^{(v)}} \boldsymbol{X} \widetilde{M}_{n,j,0,u+v} m_{V,j} + \sum_{u,v} \frac{\partial^2 \boldsymbol{N}_j^*}{\partial \boldsymbol{A}^{(u)} \partial \boldsymbol{B}^{(v)}} \boldsymbol{X} \widetilde{M}_{n,j,u,v} m_{V,j}
$$

$$
+ \sum_{u,v} \left( \boldsymbol{X}^\top \frac{\partial \boldsymbol{M}_j^*}{\partial \boldsymbol{A}^{(u)}} \boldsymbol{X} \right) \frac{\partial \boldsymbol{N}_j^*}{\partial \boldsymbol{A}^{(v)}} \boldsymbol{X} \widetilde{M}_{n,j,u,0,v,0} m_{Q,j} m_{V,j} + \sum_{u,v} \left( \boldsymbol{X}^\top \frac{\partial \boldsymbol{M}_j^*}{\partial \boldsymbol{B}^{(u)}} \boldsymbol{X} \right) \frac{\partial \boldsymbol{N}_j^*}{\partial \boldsymbol{B}^{(v)}} \boldsymbol{X} \widetilde{M}_{n,j,0,u,0,v} m_{Q,j} m_{V,j}
$$

$$
+ \sum_{u,v} \left( \boldsymbol{X}^\top \frac{\partial \boldsymbol{M}_j^*}{\partial \boldsymbol{A}^{(u)}} \boldsymbol{X} \right) \frac{\partial \boldsymbol{N}_j^*}{\partial \boldsymbol{B}^{(v)}} \boldsymbol{X} \widetilde{M}_{n,j,u,0,0,v} m_{Q,j} m_{V,j} + \sum_{u,v} \left( \boldsymbol{X}^\top \frac{\partial \boldsymbol{M}_j^*}{\partial \boldsymbol{B}^{(u)}} \boldsymbol{X} \right) \frac{\partial \boldsymbol{N}_j^*}{\partial \boldsymbol{A}^{(v)}} \boldsymbol{X} \widetilde{M}_{n,j,0,u,v,0} m_{Q,j} m_{V,j} \Bigg]
$$

$$
+ \widetilde{R}_{n,2}(\boldsymbol{X}),
$$

here we use the notation $\frac{\partial \boldsymbol{M}_j^*}{\partial \boldsymbol{A}^u}$ to denote for the value of $\frac{\partial \boldsymbol{M}}{\partial \boldsymbol{A}^u}$ at $(\boldsymbol{A}_j^*, \boldsymbol{B}_j^*)$.

**Decompose $\widetilde{B}_n(\boldsymbol{X})$.** Moving to $\widetilde{B}_n(\boldsymbol{X})$, we can decompose this function as follows:

$$\widetilde{B}_n(\boldsymbol{X}) = \sum_{j:|\mathcal{V}_j|=1} \sum_{i \in \mathcal{V}_j} \exp(c_{n,i}) \Big[\widetilde{U}_{n,i} - \widetilde{U}_j^*\Big] f_{\widetilde{G}_n}(\boldsymbol{X}) + \sum_{j:|\mathcal{V}_j|>1} \sum_{i \in \mathcal{V}_j} \exp(c_{n,i}) \Big[\widetilde{U}_{n,i} - \widetilde{U}_j^*\Big] f_{\widetilde{G}_n}(\boldsymbol{X})$$

$$:= \widetilde{B}_{n,1}(\boldsymbol{X}) + \widetilde{B}_{n,2}(\boldsymbol{X}).$$

We perform the Taylor expansions up to the first order for $\widetilde{B}_{n,1}(\boldsymbol{X})$ and the second order for $\widetilde{B}_{n,2}(\boldsymbol{X})$ leads to

$$\widetilde{B}_{n,1}(\boldsymbol{X}) = \sum_{j:|\mathcal{V}_j|=1} \sum_{|\alpha|=1} \widetilde{M}_{n,j,\alpha_1,\alpha_2} \frac{\partial^{|\alpha|} \widetilde{U}_j^*}{\partial \boldsymbol{A}^{\alpha_1} \partial \boldsymbol{B}^{\alpha_2}} f_{\widetilde{G}_n}(\boldsymbol{X}) + \widetilde{R}_{n,3}(\boldsymbol{X}),$$

$$\widetilde{B}_{n,2}(\boldsymbol{X}) = \sum_{j:|\mathcal{V}_j|=1} \sum_{1 \le |\alpha| \le 2} \widetilde{M}_{n,j,\alpha_1,\alpha_2} \frac{\partial^{|\alpha|} \widetilde{U}_j^*}{\partial \boldsymbol{A}^{\alpha_1} \partial \boldsymbol{B}^{\alpha_2}} f_{\widetilde{G}_n}(\boldsymbol{X}) + \widetilde{R}_{n,4}(\boldsymbol{X}),$$

where the Taylor remainders $\widetilde{R}_{n,3}(\mathbb{X}), \widetilde{R}_{n,4}(\mathbb{X})$ satisfy that $\widetilde{R}_{n,3}(\mathbb{X})/\mathcal{D}_2(\widetilde{G}_n, G_*) \to 0$ and $\widetilde{R}_{n,4}(\mathbb{X})/\mathcal{D}_2(\widetilde{G}_n, G_*) \to 0$. Direct calculation leads to

$$\widetilde{B}_{n,1}(\boldsymbol{X}) = \sum_{j:|\mathcal{V}_j|=1} \exp(m_{Q,j} \boldsymbol{X}^\top \mathbf{Q}_j^* \boldsymbol{X}) \sum_{u=(1,1)}^{(d,d)} \left(\boldsymbol{X}^\top \frac{\partial \mathbf{Q}_j^*}{\partial \boldsymbol{A}^u} \boldsymbol{X}\right) f_{\tilde{G}_n}(\boldsymbol{X}) \widetilde{M}_{n,j,u,0} m_{Q,j}$$

$$+ \sum_u \left(\boldsymbol{X}^\top \frac{\partial \mathbf{Q}_j^*}{\partial \boldsymbol{B}^u} \boldsymbol{X}\right) f_{\tilde{G}_n}(\boldsymbol{X}) \widetilde{M}_{n,j,0,u} m_{Q,j} + \widetilde{R}_{n,3}(\boldsymbol{X}),$$

$$\widetilde{B}_{n,2}(\mathbb{X}) = \sum_{j:|\mathcal{V}_j|>1} \exp(m_{Q,j} \boldsymbol{X}^\top \boldsymbol{M}_j^* \boldsymbol{X}) \sum_{u=(1,1)}^{(d,d)} \left(\boldsymbol{X}^\top \frac{\partial \boldsymbol{M}_j^*}{\partial \boldsymbol{A}^u} \boldsymbol{X}\right) f_{\tilde{G}_n}(\boldsymbol{X}) \widetilde{M}_{n,j,u,0} m_{Q,j}$$

$$+ \sum_u \left(\boldsymbol{X}^\top \frac{\partial \boldsymbol{M}_j^*}{\partial \boldsymbol{B}^u} \boldsymbol{X}\right) f_{\tilde{G}_n}(\boldsymbol{X}) \widetilde{M}_{n,j,0,u} m_{Q,j}$$

$$+ \sum_{u,v} \left(\boldsymbol{X}^\top \frac{\partial^2 \boldsymbol{M}_j^*}{\partial \boldsymbol{A}^{(u)} \partial \boldsymbol{A}^{(v)}} \boldsymbol{X}\right) f_{\tilde{G}_n}(\boldsymbol{X}) \widetilde{M}_{n,j,u+v,0} m_{Q,j}$$

$$+ \sum_{u,v} \left(\boldsymbol{X}^\top \frac{\partial \boldsymbol{M}_j^*}{\partial \boldsymbol{A}^{(u)}} \boldsymbol{X} \boldsymbol{X}^\top \frac{\partial \boldsymbol{M}_j^*}{\partial \boldsymbol{A}^{(v)}} \boldsymbol{X}\right) f_{\tilde{G}_n}(\boldsymbol{X}) \widetilde{M}_{n,j,u+v,0} m_{Q,j}$$

$$+ \sum_{u,v} \left(\boldsymbol{X}^\top \frac{\partial^2 \boldsymbol{M}_j^*}{\partial \boldsymbol{B}^{(u)} \partial \boldsymbol{B}^{(v)}} \boldsymbol{X}\right) f_{\tilde{G}_n}(\boldsymbol{X}) \widetilde{M}_{n,j,0,u+v} m_{Q,j}$$

$$+ \sum_{u,v} \left(\boldsymbol{X}^\top \frac{\partial \boldsymbol{M}_j^*}{\partial \boldsymbol{B}^{(u)}} \boldsymbol{X} \boldsymbol{X}^\top \frac{\partial \boldsymbol{M}_j^*}{\partial \boldsymbol{B}^{(v)}} \boldsymbol{X}\right) f_{\tilde{G}_n}(\boldsymbol{X}) \widetilde{M}_{n,j,0,u+v} m_{Q,j}$$

$$+ \sum_{u,v} \left(\boldsymbol{X}^\top \frac{\partial^2 \boldsymbol{M}_j^*}{\partial \boldsymbol{A}^{(u)} \partial \boldsymbol{B}^{(v)}} \boldsymbol{X}\right) f_{\tilde{G}_n}(\boldsymbol{X}) \widetilde{M}_{n,j,u,v} m_{Q,j}$$

$$+ \sum_{u,v} \left(\boldsymbol{X}^\top \frac{\partial \boldsymbol{M}_j^*}{\partial \boldsymbol{A}^{(u)}} \boldsymbol{X} \boldsymbol{X}^\top \frac{\partial \boldsymbol{M}_j^*}{\partial \boldsymbol{B}^{(v)}} \boldsymbol{X}\right) f_{\tilde{G}_n}(\boldsymbol{X}) \widetilde{M}_{n,j,u,v} m_{Q,j}.$$

Putting all the above results together, we can represent the function $T_n(\boldsymbol{X})$ as follows:

$$Q_n(\mathbb{X}) = \widetilde{A}_{n,1} + \widetilde{A}_{n,2} - \widetilde{B}_{n,1} - \widetilde{B}_{n,2}$$

$$+ \sum_{j=1}^{L} \widetilde{N}_{n,j} \exp\left(\boldsymbol{X}^\top m_{Q,j} \boldsymbol{M}_j^* \boldsymbol{X}\right) \Big[(m_{V,j} \boldsymbol{N}_j^*) \mathbb{X} - f_{\widetilde{G}_n}(\mathbb{X})\Big], \tag{14}$$

where $\widetilde{N}_{n,j} := \sum_{i \in \mathcal{V}_j} \exp(c_{n,i}) - \exp(c_j^*)$ for any $j \in [L]$.

**Step 2 - Non-vanishing coefficients.** As indicated in equation (14), the ratio $Q_n(\mathbb{X})/\mathcal{D}_2(\widetilde{G}_n, G_*)$ can be expressed as a linear combination of the following independent functions:

$$\widetilde{U}_j^*(\boldsymbol{X}) \left( \boldsymbol{X}^\top \frac{\partial \boldsymbol{M}_j^*}{\partial \boldsymbol{A}^{(u)}} \boldsymbol{X} \right) \boldsymbol{N}_j^* \boldsymbol{X}, \quad \widetilde{U}_j^*(\boldsymbol{X}) \left( \boldsymbol{X}^\top \frac{\partial \boldsymbol{M}_j^*}{\partial \boldsymbol{B}^{(u)}} \boldsymbol{X} \right) \boldsymbol{N}_j^* \boldsymbol{X},$$

$$\widetilde{U}_j^*(\boldsymbol{X}) \frac{\partial \boldsymbol{N}_j^*}{\partial \boldsymbol{A}^{(u)}} \boldsymbol{X}, \quad \widetilde{U}_j^*(\boldsymbol{X}) \frac{\partial \boldsymbol{N}_j^*}{\partial \boldsymbol{B}^{(u)}} \boldsymbol{X},$$

$$\widetilde{U}_j^*(\boldsymbol{X}) \left( \boldsymbol{X}^\top \frac{\partial^2 \boldsymbol{M}_j^*}{\partial \boldsymbol{A}^{(u)} \partial \boldsymbol{A}^{(v)}} \boldsymbol{X} \right) \boldsymbol{N}_j^* \boldsymbol{X}, \quad \widetilde{U}_j^*(\boldsymbol{X}) \left( \boldsymbol{X}^\top \frac{\partial \boldsymbol{M}_j^*}{\partial \boldsymbol{A}^{(u)}} \boldsymbol{X} \boldsymbol{X}^\top \frac{\partial \boldsymbol{M}_j^*}{\partial \boldsymbol{A}^{(v)}} \boldsymbol{X} \right) \boldsymbol{N}_j^* \boldsymbol{X},$$

$$\widetilde{U}_j^*(\boldsymbol{X}) \left( \boldsymbol{X}^\top \frac{\partial^2 \boldsymbol{M}_j^*}{\partial \boldsymbol{B}^{(u)} \partial \boldsymbol{B}^{(v)}} \boldsymbol{X} \right) \boldsymbol{N}_j^* \boldsymbol{X}, \quad \widetilde{U}_j^*(\boldsymbol{X}) \left( \boldsymbol{X}^\top \frac{\partial \boldsymbol{M}_j^*}{\partial \boldsymbol{B}^{(u)}} \boldsymbol{X} \boldsymbol{X}^\top \frac{\partial \boldsymbol{M}_j^*}{\partial \boldsymbol{B}^{(v)}} \boldsymbol{X} \right) \boldsymbol{N}_j^* \boldsymbol{X},$$

$$\widetilde{U}_j^*(\boldsymbol{X}) \left( \boldsymbol{X}^\top \frac{\partial^2 \boldsymbol{M}_j^*}{\partial \boldsymbol{A}^{(u)} \partial \boldsymbol{B}^{(v)}} \boldsymbol{X} \right) \boldsymbol{N}_j^* \boldsymbol{X}, \quad \widetilde{U}_j^*(\boldsymbol{X}) \left( \boldsymbol{X}^\top \frac{\partial \boldsymbol{M}_j^*}{\partial \boldsymbol{A}^{(u)}} \boldsymbol{X} \boldsymbol{X}^\top \frac{\partial \boldsymbol{M}_j^*}{\partial \boldsymbol{B}^{(v)}} \boldsymbol{X} \right) \boldsymbol{N}_j^* \boldsymbol{X},$$

$$\widetilde{U}_j^*(\boldsymbol{X}) \frac{\partial^2 \boldsymbol{N}_j^*}{\partial \boldsymbol{A}^{(u)} \partial \boldsymbol{A}^{(v)}} \boldsymbol{X}, \quad \widetilde{U}_j^*(\boldsymbol{X}) \frac{\partial^2 \boldsymbol{N}_j^*}{\partial \boldsymbol{B}^{(u)} \partial \boldsymbol{B}^{(v)}}, \quad \widetilde{U}_j^*(\boldsymbol{X}) \frac{\partial^2 \boldsymbol{N}_j^*}{\partial \boldsymbol{A}^{(u)} \partial \boldsymbol{B}^{(v)}},$$

$$\widetilde{U}_j^*(\boldsymbol{X}) \left( \boldsymbol{X}^\top \frac{\partial \boldsymbol{M}_j^*}{\partial \boldsymbol{A}^{(u)}} \boldsymbol{X} \right) \frac{\partial \boldsymbol{N}_j^*}{\partial \boldsymbol{A}^{(v)}} \boldsymbol{X}, \quad \widetilde{U}_j^*(\boldsymbol{X}) \left( \boldsymbol{X}^\top \frac{\partial \boldsymbol{M}_j^*}{\partial \boldsymbol{B}^{(u)}} \boldsymbol{X} \right) \frac{\partial \boldsymbol{N}_j^*}{\partial \boldsymbol{B}^{(v)}} \boldsymbol{X},$$

$$\widetilde{U}_j^*(\boldsymbol{X}) \left( \boldsymbol{X}^\top \frac{\partial \boldsymbol{M}_j^*}{\partial \boldsymbol{A}^{(u)}} \boldsymbol{X} \right) \frac{\partial \boldsymbol{N}_j^*}{\partial \boldsymbol{B}^{(v)}} \boldsymbol{X}, \quad \widetilde{U}_j^*(\boldsymbol{X}) \left( \boldsymbol{X}^\top \frac{\partial \boldsymbol{M}_j^*}{\partial \boldsymbol{B}^{(u)}} \boldsymbol{X} \right) \frac{\partial \boldsymbol{N}_j^*}{\partial \boldsymbol{A}^{(v)}} \boldsymbol{X},$$

$$\widetilde{U}_j^*(\boldsymbol{X}) \boldsymbol{N}_j^* \boldsymbol{X}, \quad \widetilde{U}_j^*(\boldsymbol{X}) f_{\widetilde{G}_n}(\boldsymbol{X}),$$

$$\widetilde{U}_j^*(\boldsymbol{X}) \left( \boldsymbol{X}^\top \frac{\partial \boldsymbol{M}_j^*}{\partial \boldsymbol{A}^u} \boldsymbol{X} \right) f_{\tilde{G}_n}(\boldsymbol{X}), \quad \widetilde{U}_j^*(\boldsymbol{X}) \left( \boldsymbol{X}^\top \frac{\partial \boldsymbol{M}_j^*}{\partial \boldsymbol{B}^u} \boldsymbol{X} \right) f_{\tilde{G}_n}(\boldsymbol{X}),$$

$$\widetilde{U}_j^*(\boldsymbol{X}) \left( \boldsymbol{X}^\top \frac{\partial^2 \boldsymbol{M}_j^*}{\partial \boldsymbol{A}^{(u)} \partial \boldsymbol{A}^{(v)}} \boldsymbol{X} \right) f_{\tilde{G}_n}(\boldsymbol{X}), \quad \widetilde{U}_j^*(\boldsymbol{X}) \left( \boldsymbol{X}^\top \frac{\partial \boldsymbol{M}_j^*}{\partial \boldsymbol{A}^{(u)}} \boldsymbol{X} \boldsymbol{X}^\top \frac{\partial \boldsymbol{M}_j^*}{\partial \boldsymbol{A}^{(v)}} \boldsymbol{X} \right) f_{\tilde{G}_n}(\boldsymbol{X}),$$

$$\widetilde{U}_j^*(\boldsymbol{X}) \left( \boldsymbol{X}^\top \frac{\partial^2 \boldsymbol{M}_j^*}{\partial \boldsymbol{B}^{(u)} \partial \boldsymbol{B}^{(v)}} \boldsymbol{X} \right) f_{\tilde{G}_n}(\boldsymbol{X}), \quad \widetilde{U}_j^*(\boldsymbol{X}) \left( \boldsymbol{X}^\top \frac{\partial \boldsymbol{M}_j^*}{\partial \boldsymbol{B}^{(u)}} \boldsymbol{X} \boldsymbol{X}^\top \frac{\partial \boldsymbol{M}_j^*}{\partial \boldsymbol{B}^{(v)}} \boldsymbol{X} \right) f_{\tilde{G}_n}(\boldsymbol{X}),$$

$$\widetilde{U}_j^*(\boldsymbol{X}) \left( \boldsymbol{X}^\top \frac{\partial^2 \boldsymbol{M}_j^*}{\partial \boldsymbol{A}^{(u)} \partial \boldsymbol{B}^{(v)}} \boldsymbol{X} \right) f_{\tilde{G}_n}(\boldsymbol{X}), \quad \widetilde{U}_j^*(\boldsymbol{X}) \left( \boldsymbol{X}^\top \frac{\partial \boldsymbol{M}_j^*}{\partial \boldsymbol{A}^{(u)}} \boldsymbol{X} \boldsymbol{X}^\top \frac{\partial \boldsymbol{M}_j^*}{\partial \boldsymbol{B}^{(v)}} \boldsymbol{X} \right) f_{\tilde{G}_n}(\boldsymbol{X}),$$

for any indices $1 \le j \le L$ and $u = (u_1, u_2), v = (v_1, v_2)$ with $1 \le u_1, v_1, u_2, v_2 \le d$.

We will show that at least one of the coefficients of these independent functions does not go to 0 as $n \to \infty$. Assume by contrary that all these coefficients of these linear independent functions go to 0 when $n \to \infty$. From equation (14), we have that $\widetilde{M}_{n,j,\alpha_1,\alpha_2}/\mathcal{D}_2(\widetilde{G}_n, G_*)$, $\widetilde{M}_{n,j,\alpha_1,\beta_1,\alpha_2,\beta_2}/\mathcal{D}_2(\widetilde{G}_n, G_*)$, and $\widetilde{N}_{n,j}/\mathcal{D}_2(\widetilde{G}_n, G_*)$ go to 0 for all the coefficients $\alpha_1, \alpha_2, \beta_1, \beta_2 \in \mathbb{N}^{d \times d}$ satisfying that $1 \le |\alpha_1| + |\beta_1| + |\alpha_2| + |\beta_2| \le 2$.

Since $\widetilde{N}_{n,j}/\mathcal{D}_2(\widetilde{G}_n, G_*) \to 0$, we find that for any $j \in [L]$

$$\frac{|\sum_{i \in \mathcal{V}_j} \exp(c_{n,i}) - \exp(c_j^*)|}{\mathcal{D}_2(\widetilde{G}_n, \widetilde{G}_*)} = \frac{|\widetilde{N}_{n,j}|}{\mathcal{D}_2(\widetilde{G}_n, \widetilde{G}_*)} \to 0.$$

Taking the summation of these limits over $j \in [L]$ yields

$$\frac{\sum_{j=1}^L |\sum_{i \in \mathcal{V}_j} \exp(c_{n,i}) - \exp(c_j^*)|}{\mathcal{D}_2(\widetilde{G}_n, \widetilde{G}_*)} \to 0. \tag{15}$$

Now, for any index $j \in [L]$ such that $|\mathcal{V}_j| = 1$, the limits $\widetilde{M}_{n,j,e_u,0_d}/\mathcal{D}_2(\widetilde{G}_n, \widetilde{G}_*) \to 0$ lead to $\frac{\sum_{i \in \mathcal{V}_j} \exp(c_{n,i}) \|\Delta \boldsymbol{A}_{n,ij}\|_1}{\mathcal{D}_2(\widetilde{G}_n, \widetilde{G}_*)} \to 0$ as $n \to \infty$. Because the $\ell_1$-norm and $\ell_2$-norm are equivalent, this

result implies that

$$\frac{\sum_{j:|\mathcal{V}_j|=1}\sum_{i\in\mathcal{V}_j}\exp(c_{n,i})\|\Delta\boldsymbol{A}_{n,ij}\|}{\mathcal{D}_2(\widetilde{G}_n,\widetilde{G}_*)} \to 0.$$

Similarly, since $\widetilde{M}_{n,j,0_d,e_u}/\mathcal{D}_2(\widetilde{G}_n,\widetilde{G}_*) \to 0$, we also have that $\frac{\sum_{j:|\mathcal{V}_j|=1}\sum_{i\in\mathcal{V}_j}\exp(c_{n,i})\|\Delta\boldsymbol{B}_{n,ij}\|}{\mathcal{D}_2(\widetilde{G}_n,\widetilde{G}_*)} \to 0$. Thus, we obtain

$$\frac{\sum_{j:|\mathcal{V}_j|=1}\sum_{i\in\mathcal{V}_j}\exp(c_{n,i})(\|\Delta\boldsymbol{A}_{n,ij}\|+\|\Delta\boldsymbol{B}_{n,ij}\|)}{\mathcal{D}_2(\widetilde{G}_n,\widetilde{G}_*)} \to 0 \qquad (16)$$

Moving to indices $j \in [L]$ such that their corresponding Voronoi cells have more than one element, i.e., $|\mathcal{V}_j| > 1$. The limits $\widetilde{M}_{n,j,2e_u,0_d}/\mathcal{D}_2(\widetilde{G}_n,\widetilde{G}_*) \to 0$ and $\widetilde{M}_{n,j,0_d,2e_u}/\mathcal{D}_2(\widetilde{G}_n,\widetilde{G}_*) \to 0$ induces that

$$\frac{\sum_{j:|\mathcal{V}_j|>1}\sum_{i\in\mathcal{V}_j}\exp(c_{n,i})(\|\Delta\boldsymbol{A}_{n,ij}\|^2+\|\Delta\boldsymbol{B}_{n,ij}\|^2)}{\mathcal{D}_2(\widetilde{G}_n,G_*)} \to 0 \qquad (17)$$

By putting the results in equations (15), (16), and (17) together, we arrive at $1 = \frac{\mathcal{D}_2(\widetilde{G}_n,\widetilde{G}_*)}{\mathcal{D}_2(\widetilde{G}_n,\widetilde{G}_*)} \to 0$ as $n \to \infty$, which is a contradiction. As a consequence, at least one of the coefficients of the linear independent functions in $T_n(\boldsymbol{X})/\mathcal{D}_2(\widetilde{G}_n,G_*)$ does not go to 0 as $n \to \infty$.

**Step 3 - Application of the Fatou's lemma.** We define $\widetilde{m}_n$ to be the maximum of the absolute values of the coefficients of the linear independent functions in $T_n(\boldsymbol{X})/\mathcal{D}_2(\widetilde{G}_n,G_*)$. As at least one of these coefficients does not go to 0, it indicates that $1/\widetilde{m}_n \not\to \infty$ as $n \to \infty$.

Since $\|f_{\widetilde{G}_n} - f_{\widetilde{G}_*}\|_{L^2(\mu)}/\mathcal{D}_2(\widetilde{G}_n,\widetilde{G}_*) \to 0$ as $n \to \infty$, we obtain $\|f_{\widetilde{G}_n} - f_{\widetilde{G}_*}\|_{L^2(\mu)}/(\widetilde{m}_n\mathcal{D}_2(\widetilde{G}_n,G_*)) \to 0$. An application of the Fatou's lemma leads to:

$$0 = \lim_{n\to\infty}\frac{\|f_{\widetilde{G}_n}-f_{\widetilde{G}_*}\|_{L^2(\mu)}}{\widetilde{m}_n\mathcal{D}_2(\widetilde{G}_n,G_*)} \geq \int \liminf_{n\to\infty}\frac{\left|f_{\widetilde{G}_n}(\boldsymbol{X})-f_{\widetilde{G}_*}(\boldsymbol{X})\right|}{\widetilde{m}_n\mathcal{D}_2(\widetilde{G}_n,G_*)}d\mu(\boldsymbol{X}) \geq 0.$$

This inequality implies that $\liminf_{n\to\infty}\frac{\left|f_{\widetilde{G}_n}(\boldsymbol{X})-f_{\widetilde{G}_*}(\boldsymbol{X})\right|}{\widetilde{m}_n\mathcal{D}_2(\widetilde{G}_n,G_*)} = 0$ for almost surely $\boldsymbol{X}$. As $n \to \infty$, we denote

$$\frac{\widetilde{M}_{n,j,\alpha_1,\alpha_2}}{\widetilde{m}_n\mathcal{D}_2(\widetilde{G}_n,G_*)} \to \tilde{\tau}_{j,\alpha_1,\alpha_2}, \quad \frac{\widetilde{M}_{n,j,\alpha_1,\alpha_2,\beta_1,\beta_2}}{\widetilde{m}_n\mathcal{D}_2(\widetilde{G}_n,G_*)} \to \tilde{\xi}_{j,\alpha_1,\alpha_2,\beta_1,\beta_2}, \quad \frac{\widetilde{N}_{n,j}}{\widetilde{m}_n\mathcal{D}_2(\widetilde{G}_n,G_*)} \to \tilde{\lambda}_{0,j},$$

for any indices $j \in [L]$ and any coefficients $\alpha_1, \alpha_2, \beta_1, \beta_2$ such that $1 \leq |\alpha_1|+|\alpha_2|+|\beta_1|+|\beta_2| \leq 2$. Here, at least one element of the set $\{\tilde{\tau}_{j,\alpha_1,\alpha_2}, \tilde{\xi}_{j,\alpha_1,\alpha_2,\beta_1,\beta_2}, \tilde{\lambda}_{0,j}\}$ is different from 0. Given the above notations, the limit $\liminf_{n\to\infty}\frac{\left|f_{\widetilde{G}_n}(\boldsymbol{X})-f_{\widetilde{G}_*}(\boldsymbol{X})\right|}{\tilde{m}_n\mathcal{D}_2(\widetilde{G}_n,G_*)} = 0$ implies that,

$$0 = \liminf_{n\to\infty}\frac{T_n(\boldsymbol{X})}{\tilde{m}_n\mathcal{D}_2(\widetilde{G}_n,G_*)}$$

$$= \liminf_{n\to\infty}\frac{\widetilde{A}_{n,1}+\widetilde{A}_{n,2}-\widetilde{B}_{n,1}-\widetilde{B}_{n,2}+\sum_{j=1}^{L}\widetilde{N}_{n,j}\exp\left(\boldsymbol{X}^\top m_{Q,j}\boldsymbol{M}_j^*\boldsymbol{X}\right)\left[(m_{V,j}\boldsymbol{N}_j^*)\mathbb{X}-f_{\widetilde{G}_n}(\mathbb{X})\right]}{\tilde{m}_n\mathcal{D}_2(\widetilde{G}_n,G_*)},$$

for almost surely $\mathbb{X}$. For example, we look at the limit of $\widetilde{A}_{n,1}/\tilde{m}_n \mathcal{D}_2(\widetilde{G}_n, G_*)$,

$$
\liminf_{n \to \infty} \frac{\widetilde{A}_{n,1}}{\tilde{m}_n \mathcal{D}_2(\widetilde{G}_n, G_*)} = \sum_{j:|\mathcal{V}_j|=1} \left( \exp(m_{Q,j} \boldsymbol{X}^\top \boldsymbol{M}_j^* \boldsymbol{X}) \left[ \sum_{u=(1,1)}^{(d,d)} \left( \boldsymbol{X}^\top \frac{\partial \boldsymbol{M}_j^*}{\partial \boldsymbol{A}^u} \boldsymbol{X} \right) \boldsymbol{N}_j^* \boldsymbol{X} \times \widetilde{\tau}_{j,u,0} m_{Q,j} m_{V,j} \right. \right.
$$

$$
+ \sum_{u=(1,1)}^{(d,d)} \left( \boldsymbol{X}^\top \frac{\partial \boldsymbol{M}_j^*}{\partial \boldsymbol{B}^u} \boldsymbol{X} \right) \boldsymbol{N}_j^* \boldsymbol{X} \times \widetilde{\tau}_{j,0,u} m_{Q,j} m_{V,j} + \sum_{u=(1,1)}^{(d,d)} \frac{\partial \boldsymbol{N}_j^*}{\partial \boldsymbol{A}^u} \boldsymbol{X} \times \widetilde{\tau}_{j,u,0} m_{V,j}
$$

$$
\left. \left. + \frac{\partial \boldsymbol{N}_j^*}{\partial \boldsymbol{B}^u} \boldsymbol{X} \times \widetilde{\tau}_{j,0,u} m_{V,j} \right] \right).
$$

From the equation $0 = \liminf_{n \to \infty} \frac{T_n(\boldsymbol{X})}{\tilde{m}_n \mathcal{D}_2(\widetilde{G}_n, G_*)}$ and the linear independence of the functions imply that all the coefficients $\{\tilde{\tau}_{j,\alpha_1,\alpha_2}, \tilde{\xi}_{j,\alpha_1,\alpha_2,\beta_1,\beta_2}, \tilde{\lambda}_{0,j}\}$ are 0. It is a contradiction. As a consequence, we obtain that

$$
\lim_{\varepsilon \to 0} \inf_{\widetilde{G} \in \mathcal{G}_{L'}(\Theta) : \mathcal{D}_3(\widetilde{G}, \widetilde{G}_*) \leq \varepsilon} \|f_{\widetilde{G}} - f_{\widetilde{G}_*}\|_{L^2(\mu)} / \mathcal{D}_2(\widetilde{G}_n, G_*) > 0.
$$

### D.2.2 GLOBAL PART

The result of the local part implies that there exists a positive constant $\varepsilon'$ such that

$$
\inf_{\widetilde{G} \in \mathcal{G}_{L'}(\Theta) : \mathcal{D}_2(\widetilde{G}, \widetilde{G}_*) \leq \varepsilon'} \|f_{\widetilde{G}} - f_{\widetilde{G}_*}\|_{L^2(\mu)} / \mathcal{D}_2(\widetilde{G}, \widetilde{G}_*) > 0.
$$

Therefore the remaining part is to prove

$$
\inf_{\widetilde{G} \in \mathcal{G}_{L'}(\Theta) : \mathcal{D}_2(\widetilde{G}, \widetilde{G}_*) > \varepsilon'} \|f_{\widetilde{G}} - f_{\widetilde{G}_*}\|_{L^2(\mu)} / \mathcal{D}_2(\widetilde{G}, \widetilde{G}_*) > 0.
$$

We assume by contradiction that the above claim does not hold. It means there exists a sequence of measures $\widetilde{G}'_n := \sum_{j=1}^{L} \exp(c_{n,j}) \delta_{(\boldsymbol{B}_{n,j}, \boldsymbol{A}_{n,j})}$ in $\widetilde{\mathcal{G}}_{L'}(\Theta)$ such that

$$
\begin{cases} \mathcal{D}_2(\widetilde{G}, \widetilde{G}_*) > \varepsilon' \\ \|f_{\widetilde{G}'_n} - f_{\widetilde{G}_*}\|_{L^2(\mu)} / \mathcal{D}_2(\widetilde{G}, \widetilde{G}_*) \to 0 \end{cases}
$$

as $n \to \infty$, which implies that $\|f_{\widetilde{G}'_n} - f_{\widetilde{G}_*}\|_{L^2(\mu)} \to 0$ as $n \to \infty$.

Given that $\Theta$ is a compact set, there exists a mixing measure $\widetilde{G}'$ in $\widetilde{\mathcal{G}}_{L'}(\Theta)$ such that one of the $\widetilde{G}'_n$'s subsequences converges to $\widetilde{G}'$. Since $\mathcal{D}_2(\widetilde{G}'_n, \widetilde{G}_*) > \varepsilon'$, we obtain that $\mathcal{D}_2(\widetilde{G}', \widetilde{G}_*) > \varepsilon'$. An application of the Fatou's lemma leads to

$$
0 = \lim_{n \to \infty} \|f_{\widetilde{G}'_n} - f_{\widetilde{G}_*}\|_{L^2(\mu)} \geq \int \liminf_{n \to \infty} \left\| f_{\widetilde{G}'_n}(\boldsymbol{X}) - f_{\widetilde{G}_*}(\boldsymbol{X}) \right\|^2 d\mu(\boldsymbol{X}).
$$

The above inequality indicates that $f_{\widetilde{G}'} = f_{\widetilde{G}_*}$ for almost surely $\mathbb{X}$. From the identifiability property that we will prove shortly below, we deduce that $\widetilde{G}' \equiv \widetilde{G}_*$. It follows that $\mathcal{D}_2(\widetilde{G}', \widetilde{G}_*) = 0$, which is opposed to the fact that $\mathcal{D}_2(\widetilde{G}', \widetilde{G}_*) > \varepsilon' > 0$. Hence, the proof is completed. **Proof for the identifiability property.** We will prove that if $f_{\widetilde{G}}(\boldsymbol{X}) = f_{\widetilde{G}_*}(\boldsymbol{X})$ for almost surely $\boldsymbol{X}$, then $\widetilde{G} \equiv \widetilde{G}_*$. To ease the presentation, for any mixing measure $\widetilde{G} = \sum_{j=1}^{\tilde{L}} \exp(c_j) \delta_{(\boldsymbol{B}_j, \boldsymbol{A}_j)} \in \mathcal{G}_{L'}(\Theta)$, we denote

$$
\mathrm{softmax}_G(u) = \frac{\exp(u)}{\sum_{k=1}^{L} \exp\left( \boldsymbol{X}^\top m_{Q,k} \frac{\boldsymbol{C}_Q + \sigma_2(\boldsymbol{B}_k)\sigma_1(\boldsymbol{A}_k)}{\|\boldsymbol{C}_Q + \sigma_2(\boldsymbol{B}_k)\sigma_1(\boldsymbol{A}_k)\| + \tau_Q} \boldsymbol{X} + c_k \right)},
$$

where $u \in \left\{ \boldsymbol{X}^\top m_{Q,j} \frac{\boldsymbol{C}_Q + \sigma_2(\boldsymbol{B}_j)\sigma_1(\boldsymbol{A}_j)}{\|\boldsymbol{C}_Q + \sigma_2(\boldsymbol{B}_j)\sigma_1(\boldsymbol{A}_j)\| + \tau_Q} \boldsymbol{X} + c_j : j \in [\tilde{L}] \right\}$.

The equation $f_{\widetilde{G}}(\mathbb{X}) = f_{\widetilde{G}_*}(\mathbb{X})$ indicates that

$$\sum_{j=1}^{\tilde{L}} \text{softmax}\left( \boldsymbol{X}^\top m_{Q,j} \frac{\boldsymbol{C}_Q + \sigma_2(\boldsymbol{B}_j)\sigma_1(\boldsymbol{A}_j)}{\|\boldsymbol{C}_Q + \sigma_2(\boldsymbol{B}_j)\sigma_1(\boldsymbol{A}_j)\| + \tau_Q} \boldsymbol{X} + c_j \right) \left( m_{V,j} \frac{\boldsymbol{C}_V + \sigma_2(\boldsymbol{B}_j)\sigma_1(\boldsymbol{A}_j)}{\|\boldsymbol{C}_V + \sigma_2(\boldsymbol{B}_j)\sigma_1(\boldsymbol{A}_j)\| + \tau_V} \right) \boldsymbol{X}$$

$$= \sum_{j=1}^{L} \text{softmax}\left( \boldsymbol{X}^\top m_{Q,j} \frac{\boldsymbol{C}_Q + \sigma_2(\boldsymbol{B}_j^*)\sigma_1(\boldsymbol{A}_j^*)}{\|\boldsymbol{C}_Q + \sigma_2(\boldsymbol{B}_j^*)\sigma_1(\boldsymbol{A}_j^*)\| + \tau_Q} \boldsymbol{X} + c_j^* \right) \left( m_{V,j} \frac{\boldsymbol{C}_V + \sigma_2(\boldsymbol{B}_j^*)\sigma_1(\boldsymbol{A}_j^*)}{\|\boldsymbol{C}_V + \sigma_2(\boldsymbol{B}_j^*)\sigma_1(\boldsymbol{A}_j^*)\| + \tau_V} \right) \boldsymbol{X}$$

$$(18)$$

That equation implies that $\tilde{L} = L$. As a consequence, we find that

$$\left\{ \text{softmax}(\boldsymbol{X}^\top m_{Q,j} \frac{\boldsymbol{C}_Q + \sigma_2(\boldsymbol{B}_j)\sigma_1(\boldsymbol{A}_j)}{\|\boldsymbol{C}_Q + \sigma_2(\boldsymbol{B}_j)\sigma_1(\boldsymbol{A}_j)\| + \tau_Q} \boldsymbol{X} + c_j) : j \in [\tilde{L}] \right\}$$

$$= \left\{ \text{softmax}(\boldsymbol{X}^\top m_{Q,j} \frac{\boldsymbol{C}_Q + \sigma_2(\boldsymbol{B}_j^*)\sigma_1(\boldsymbol{A}_j^*)}{\|\boldsymbol{C}_Q + \sigma_2(\boldsymbol{B}_j^*)\sigma_1(\boldsymbol{A}_j^*)\| + \tau_Q} \boldsymbol{X} + c_j^*) : j \in [L] \right\}$$

for almost surely $\mathbb{X}$. By relabelling the indices, we can assume without loss of generality that for any $j \in [L]$

$$\text{softmax}(\boldsymbol{X}^\top m_{Q,j} \frac{\boldsymbol{C}_Q + \sigma_2(\boldsymbol{B}_j^*)\sigma_1(\boldsymbol{A}_j^*)}{\|\boldsymbol{C}_Q + \sigma_2(\boldsymbol{B}_j^*)\sigma_1(\boldsymbol{A}_j^*)\| + \tau_Q} \boldsymbol{X} + c_j^*) = \text{softmax}(\boldsymbol{X}^\top m_{Q,j} \frac{\boldsymbol{C}_Q + \sigma_2(\boldsymbol{B}_j)\sigma_1(\boldsymbol{A}_j)}{\|\boldsymbol{C}_Q + \sigma_2(\boldsymbol{B}_j)\sigma_1(\boldsymbol{A}_j)\| + \tau_Q} \boldsymbol{X} + c_j),$$

for almost surely $\mathbb{X}$. Given the invariance to translation of the softmax function, the equation (18) leads to

$$\sum_{j=1}^{\tilde{L}} \exp(c_j) \exp\left( \boldsymbol{X}^\top m_{Q,j} \frac{\boldsymbol{C}_Q + \sigma_2(\boldsymbol{B}_j^*)\sigma_1(\boldsymbol{A}_j^*)}{\|\boldsymbol{C}_Q + \sigma_2(\boldsymbol{B}_j^*)\sigma_1(\boldsymbol{A}_j^*)\| + \tau_Q} \boldsymbol{X} \right) \left( m_{V,j} \frac{\boldsymbol{C}_V + \sigma_2(\boldsymbol{B}_j)\sigma_1(\boldsymbol{A}_j)}{\|\boldsymbol{C}_V + \sigma_2(\boldsymbol{B}_j)\sigma_1(\boldsymbol{A}_j)\| + \tau_V} \right) \boldsymbol{X}$$

$$= \sum_{j=1}^{L} \exp(c_j^*) \exp\left( \boldsymbol{X}^\top m_{Q,j} \frac{\boldsymbol{C}_Q + \sigma_2(\boldsymbol{B}_j^*)\sigma_1(\boldsymbol{A}_j^*)}{\|\boldsymbol{C}_Q + \sigma_2(\boldsymbol{B}_j^*)\sigma_1(\boldsymbol{A}_j^*)\| + \tau_Q} \boldsymbol{X} \right) \left( m_{V,j}^* \frac{\boldsymbol{C}_V + \sigma_2(\boldsymbol{B}_j^*)\sigma_1(\boldsymbol{A}_j^*)}{\|\boldsymbol{C}_V + \sigma_2(\boldsymbol{B}_j^*)\sigma_1(\boldsymbol{A}_j^*)\| + \tau_V} \right) \boldsymbol{X},$$

$$(19)$$

for almost surely $\mathbb{X}$.

Now, the index set $[L]$ can be partitioned into $\tilde{m}$ subsets $\tilde{K}_1, \tilde{K}_2, \ldots, \tilde{K}_{\tilde{m}}$ where $\tilde{m} \le L$, such that $\exp(c_j) = \exp(c_{j'}^*)$ for any indices $j, j' \in \tilde{K}_i$ and $i \in [\tilde{m}]$. Thus, equation (19) can be rewritten as follows:

$$\sum_{i=1}^{\tilde{m}} \sum_{j \in \tilde{K}_i} \exp(c_j^*) \exp\left( \boldsymbol{X}^\top m_{Q,j} \frac{\boldsymbol{C}_Q + \sigma_2(\boldsymbol{B}_j^*)\sigma_1(\boldsymbol{A}_j^*)}{\|\boldsymbol{C}_Q + \sigma_2(\boldsymbol{B}_j^*)\sigma_1(\boldsymbol{A}_j^*)\| + \tau_Q} \boldsymbol{X} \right) \left( m_{V,j} \frac{\boldsymbol{C}_V + \sigma_2(\boldsymbol{B}_j)\sigma_1(\boldsymbol{A}_j)}{\|\boldsymbol{C}_V + \sigma_2(\boldsymbol{B}_j)\sigma_1(\boldsymbol{A}_j)\| + \tau_V} \right) \boldsymbol{X}$$

$$= \sum_{i=1}^{\tilde{m}} \sum_{j \in \tilde{K}_i} \exp(c_j^*) \exp\left( \boldsymbol{X}^\top m_{Q,j} \frac{\boldsymbol{C}_Q + \sigma_2(\boldsymbol{B}_j^*)\sigma_1(\boldsymbol{A}_j^*)}{\|\boldsymbol{C}_Q + \sigma_2(\boldsymbol{B}_j^*)\sigma_1(\boldsymbol{A}_j^*)\| + \tau_Q} \boldsymbol{X} \right) \left( m_{V,j} \frac{\boldsymbol{C}_V + \sigma_2(\boldsymbol{B}_j^*)\sigma_1(\boldsymbol{A}_j^*)}{\|\boldsymbol{C}_V + \sigma_2(\boldsymbol{B}_j^*)\sigma_1(\boldsymbol{A}_j^*)\| + \tau_V} \right) \boldsymbol{X},$$

for almost surely $\mathbb{X}$. The above equation implies that

$$\left\{ \frac{\boldsymbol{C}_V + \sigma_2(\boldsymbol{B}_j)\sigma_1(\boldsymbol{A}_j)}{\|\boldsymbol{C}_V + \sigma_2(\boldsymbol{B}_j)\sigma_1(\boldsymbol{A}_j)\| + \tau_V} : j \in \tilde{K}_i \right\} = \left\{ \frac{\boldsymbol{C}_V + \sigma_2(\boldsymbol{B}_j^*)\sigma_1(\boldsymbol{A}_j^*)}{\|\boldsymbol{C}_V + \sigma_2(\boldsymbol{B}_j^*)\sigma_1(\boldsymbol{A}_j^*)\| + \tau_V} : j \in \tilde{K}_i \right\},$$

for any $i \in [\tilde{m}]$ and for almost surely $\mathbb{X}$. Since the activation functions $\sigma_1$ and $\sigma_2$ are algebraically independent, the above result indicates that

$$\sum_{i=1}^{\tilde{m}} \sum_{j \in \tilde{K}_i} \exp(c_j)\delta_{(\boldsymbol{B}_j, \boldsymbol{A}_j)} = \sum_{i=1}^{\tilde{m}} \sum_{j \in \tilde{K}_i} \exp(c_j^*)\delta_{(\boldsymbol{B}_j^*, \boldsymbol{A}_j^*)}.$$

As a consequence, $\widetilde{G} \equiv \widetilde{G}_*$ and the proof is completed.

### D.3    PROOF OF PROPOSITION 1

We recall that $(\boldsymbol{X}_1, \boldsymbol{Y}_1), (\boldsymbol{X}_2, \boldsymbol{Y}_2), \ldots, (\boldsymbol{X}_n, \boldsymbol{Y}_n) \in \mathbb{R}^d \times \mathbb{R}^d$ are i.i.d. samples from the following regression model:

$$\boldsymbol{Y}_i = f_{G_*}(\boldsymbol{X}_i) + \varepsilon_i, \quad i = 1, 2, \ldots, n,$$

where the Gaussian noises $\varepsilon_1, \ldots, \varepsilon_n$ are i.i.d. and satisfy that $\mathbb{E}[\varepsilon_i | \boldsymbol{X}_i] = 0$ and $\text{Var}(\varepsilon_i | \boldsymbol{X}_i) = \sigma^2 I_{\bar{d}}$ for all $i \in [n]$ and $f_{G_*}(.)$ admits the following form:

$$f_{G_*}(\boldsymbol{X}) := \sum_{j=1}^L \frac{\exp\left(\boldsymbol{X}^\top m_{Q,j} \frac{\boldsymbol{C}_Q + \sigma_2(\boldsymbol{W}_{2,j}^* \boldsymbol{B}_j^*)\sigma_1(\boldsymbol{W}_{1,j}^* \boldsymbol{A}_j^*)}{\|\boldsymbol{C}_Q + \sigma_2(\boldsymbol{W}_{2,j}^* \boldsymbol{B}_j^*)\sigma_1(\boldsymbol{W}_{1,j}^* \boldsymbol{A}_j^*)\| + \tau_Q} \boldsymbol{C}_K \boldsymbol{X} + c_j^*\right)}{S(\boldsymbol{X})}$$

$$\times \left( m_{V,j} \frac{\boldsymbol{C}_V + \sigma_2(\boldsymbol{W}_{2,j}^* \boldsymbol{B}_j^*)\sigma_1(\boldsymbol{W}_{1,j}^* \boldsymbol{A}_j^*)}{\|\boldsymbol{C}_V + \sigma_2(\boldsymbol{W}_{2,j}^* \boldsymbol{B}_j^*)\sigma_1(\boldsymbol{W}_{1,j}^* \boldsymbol{A}_j^*)\| + \tau_V} \right) \boldsymbol{X}, \quad (20)$$

where

$$S(\boldsymbol{X}) = \sum_{k=1}^L \exp\left(\boldsymbol{X}^\top m_{Q,k} \frac{\boldsymbol{C}_Q + \sigma_2(\boldsymbol{W}_{2,j}^* \boldsymbol{B}_j^*)\sigma_1(\boldsymbol{W}_{1,j}^* \boldsymbol{A}_j^*)}{\|\boldsymbol{C}_Q + \sigma_2(\boldsymbol{W}_{2,j}^* \boldsymbol{B}_j^*)\sigma_1(\boldsymbol{W}_{1,j}^* \boldsymbol{A}_j^*)\| + \tau_Q} \boldsymbol{C}_K \boldsymbol{X} + c_k^*\right). \quad (21)$$

The least-square estimator $G_n$ takes the following form:

$$G_n := \arg\min_{G \in \mathcal{G}_{L'}(\Theta)} \sum_{i=1}^n \|\boldsymbol{Y}_i - f_G(\boldsymbol{X}_i)\|^2,$$

From the Gaussianity assumption of $\varepsilon_i | \boldsymbol{X}_i$ for all $i \in [n]$, we have $\boldsymbol{Y}_i | \boldsymbol{X}_i \sim \mathcal{N}(f_{G_*}(\boldsymbol{X}_i), \sigma^2 I_d)$ for all $i \in [n]$. Therefore, the least square estimator $G_n$ is indeed equivalent to a maximum likelihood estimator with respect to the data $Y_1 | \boldsymbol{X}_1, \ldots, Y_n | \boldsymbol{X}_n$, which has the following form:

$$G_n \in \arg\max_{G \in \mathcal{G}_{L'}(\Theta)} \frac{1}{n} \sum_{i=1}^n \log(p(\boldsymbol{Y}_i | f_G(\boldsymbol{X}_i), \sigma^2 I_{\bar{d}}))$$

where $p(\boldsymbol{Y}_i | f_G(\boldsymbol{X}_i), \sigma^2 I_d)$ stands for multivariate Gaussian distribution with mean $f_G(\boldsymbol{X})$ and covariance matrix $\sigma^2 I_d$.

Next, we will denote some useful notations to help us quantify the distance between the two density functions, $p(\boldsymbol{Y} | f_{G_n}(\mathbb{X}), \sigma^2 I_d)$ and $p(\boldsymbol{Y} | f_{G_*}(\mathbb{X}), \sigma^2 I_d)$. Recall the set of all mixing measure $\mathcal{G}_L(\Theta)$ and denote the set of regression function $\mathcal{F}_L := \{f_G(x) : G \in \mathcal{G}_L(\Theta)\}$. Let $\mathcal{P}_L$ denotes the set of conditional density of all mixing measures in $\mathcal{G}_L(\Theta)$, and we further denote,

$$\tilde{\mathcal{P}}_L(\Theta) := \{p_{(G+G_*)/2}(\boldsymbol{Y} | \mathbb{X}) : G \in \mathcal{G}_L(\Theta)\},$$
$$\tilde{\mathcal{P}}_L^{1/2}(\Theta) := \{p_{(G+G_*)/2}^{1/2}(\boldsymbol{Y} | \mathbb{X}) : G \in \mathcal{G}_L(\Theta)\}.$$

Also, for each $\delta > 0$, we define the Hellinger ball in $\tilde{\mathcal{P}}_L^{1/2}(\Theta)$ centered around the conditional density $p_{G_*}$,

$$\tilde{\mathcal{P}}_L^{1/2}(\Theta, \delta) := \{p^{1/2} \in \tilde{\mathcal{P}}_L^{1/2}(\Theta) : h(p, p_{G_*}) \leq \delta\}.$$

To measure the size of the above sets, we leverage the following quantity from van de Geer (2000):

$$\mathcal{J}(\delta, \tilde{\mathcal{P}}_L^{1/2}(\Theta, \delta)) := \int_{\delta^2/2^{13}}^\delta H_B^{1/2}(t, \tilde{\mathcal{P}}_L^{1/2}(\Theta, t), \| \cdot \|_{\mathcal{L}^2(\mu)}) dt \vee \delta, \quad (22)$$

where $H_B(t, \tilde{\mathcal{P}}_L^{1/2}(\Theta, t), \| \cdot \|_{\mathcal{L}^2(\mu)})$ denotes the bracketing entropy of $\tilde{\mathcal{P}}_L^{1/2}(\Theta, t)$ under $\mathcal{L}^2$-norm, while $t \vee \delta = \max(t, \delta)$. By adapting the proof argument of Theorem 7.4 and Theorem 9.2 in van de Geer (2000), we have the following lemma.

**Lemma 2.** *Consider the function $\Psi(\delta) \geq \mathcal{J}(\delta, \mathcal{P}_{HL}^{1/2}(\Theta, \delta))$ such that $\Psi(\delta)/\delta^2$ is a non-increasing function of $\delta$. Then, for some universal constant $c$ and sequence $(\delta_n)$ such that $\sqrt{n}\delta_n^2 \geq c\Psi(\delta_n)$, we have that*

$$\mathbb{P}\left(\mathbb{E}_{\mathbb{X}}[h(p_{\hat{G}_n}(\cdot|\mathbb{X}), p_{G_*}(\cdot|\mathbb{X}))] > \delta\right) \leq c\exp\left(-\frac{n\delta^2}{\nu^2}\right)$$

*for all $\delta \geq \delta_n$.*

The goal is to upper bound the bracketing entropy for any $0 < \epsilon \leq 1/2$, i.e.,

$$H_B(\epsilon, \tilde{\mathcal{P}}_{HL}^{1/2}(\Theta, \epsilon), \|\cdot\|_{\mathcal{L}^2(\mu)}) \lesssim \log(1/\epsilon). \tag{23}$$

We will prove this bound later. Given this bound, it follows that,

$$\mathcal{J}_B(\delta, \tilde{\mathcal{P}}_{HL}^{1/2}(\Theta, \delta)) \lesssim \int_{\delta^2/2^{13}}^{\delta} \log(1/t)dt \vee \delta. \tag{24}$$

Consider $\Psi(\delta) = \delta \cdot [\log(1/\delta)]^{1/2}$, the it is obvious that $\Psi(\delta)/\delta^2$ is non-increasing function of $\delta$. In addition, equation 24 implies that $\Psi(\delta) \geq \mathcal{J}_B(\delta, \tilde{\mathcal{P}}_{HL}^{1/2}(\Theta, \delta))$. By choosing $\delta_n = \sqrt{\log(n)/n}$, we have $\sqrt{n}\delta_n^2 \geq c\Psi(\delta_n)$ for some universal constant $c$. An application of Lemma 2 leads us to the conclusion of Proposition 1:

$$h(p(\mathbf{Y}|g_{\bar{G}_n}(\mathbb{X}), \sigma^2 I_d), p(\mathbf{Y}|g_{\bar{G}_*}(\mathbb{X}), \sigma^2 I_d)) = \mathcal{O}(\sqrt{\log(n)/n}), \tag{25}$$

where $h$ denotes the Hellinger distance. Hellinger distance between two multivariate normal distributions has the following closed-form,

$$h(p(\mathbf{Y}|g_{\bar{G}_n}(\mathbb{X}), \sigma^2 I_d), p(\mathbf{Y}|g_{\bar{G}_*}(\mathbb{X}), \sigma^2 I_d)) = 1 - \exp\left\{-\frac{1}{8\sigma^2}\|g_{\bar{G}_n}(\mathbb{X}) - g_{\bar{G}_*}(\mathbb{X})\|^2\right\}.$$

Therefore, for sufficient large $n$, there exists some universal constant $C$ such that

$$\|g_{\bar{G}_n}(\mathbb{X}) - g_{\bar{G}_*}(\mathbb{X})\|^2 \leq 8\sigma^2 \log\left(\frac{1}{1 - C\log(n)/n}\right) \leq 16\sigma^2 C\log(n)/n.$$

As a consequence, we have

$$\|g_{\bar{G}_n}(\mathbb{X}) - g_{\bar{G}_*}(\mathbb{X})\| = \mathcal{O}(\sqrt{\log(n)/n}),$$

or $\|g_{\bar{G}_n} - g_{\bar{G}_*}\|_{L^2(\mu)} = \mathcal{O}_P(\sqrt{\log(n)/n})$. This concludes the proof of this proposition.

Now we go back to prove the upper bound at equation 23. First we derive an upper bound for the multivariate Gaussian density $p_G(\cdot|\mathbb{X})$. Since the variance effect $\sigma^2$ is fixed, we have

$$p_G(\mathbf{Y}|\mathbb{X}) = \frac{1}{(2\pi\sigma^2)^{d/2}} \exp\left(-\frac{\|\mathbf{Y} - f_G(\mathbb{X})\|^2}{2\sigma^2}\right) \leq \frac{1}{(2\pi\sigma^2)^{d/2}}.$$

Because the input space $\mathcal{X}$ and parameter space $\Theta$ are both bounded, there exists a constant $M$ such that $\|f_G(\mathbb{X})\| \leq M$ for $G \in \mathcal{G}_L$ and $\mathbb{X} \in \mathcal{X}$. Thus, for any $\|\mathbf{Y}\| \geq 2M$, we have $\frac{\|\mathbf{Y} - g_G(\mathbb{X})\|^2}{2\sigma^2} \geq \frac{\|\mathbf{Y}\|^2}{8\sigma^2}$, which leads to

$$p(\mathbf{Y}|g_G(\mathbb{X}), \sigma^2 I_{\bar{d}}) = \frac{1}{(2\pi\sigma^2)^{d/2}} \exp\left(-\frac{\|\mathbf{Y} - g_G(\mathbb{X})\|^2}{2\sigma^2}\right) \leq \frac{1}{(2\pi\sigma^2)^{d/2}} \exp\left(-\frac{\|\mathbf{Y}\|^2}{8\sigma^2}\right).$$

We define the integrable function

$$K(\mathbf{Y}|\mathbb{X}) = \begin{cases} (2\pi\sigma^2)^{-d/2} & \text{for } \|\mathbf{Y}\| \leq 2M, \\ (2\pi\sigma^2)^{-d/2} \exp\left(-\frac{\|\mathbf{Y}\|^2}{8\sigma^2}\right) & \text{for } \|\mathbf{Y}\| > 2M, \end{cases}$$

and thus $p(\mathbf{Y}|g_G(\mathbb{X}), \sigma^2 I_d) \leq K(\mathbf{Y}|\mathbb{X})$ for all $\mathbf{Y}$ and $\mathbb{X} \in \mathcal{X}$.

Let $\tau < \epsilon$ and $\{e_1, \ldots, e_n\}$ be the $\tau$-cover of $\mathcal{P}_L(\Theta)$ under $\ell_1$-norm such that the covering number $N := N(\tau, \mathcal{P}_L(\Theta), \|\cdot\|_1)$. Next, we construct the brackets of the form $[L_i(\boldsymbol{Y}|\mathbb{X}), U_i(\boldsymbol{Y}|\mathbb{X})]$, for $1 \leq i \leq N$ with

$$L_i(\boldsymbol{Y}|\mathbb{X}) := \max\{e_i(\boldsymbol{Y}|\mathbb{X}) - \tau, 0\},$$
$$U_i(\boldsymbol{Y}|\mathbb{X}) := \max\{e_i(\boldsymbol{Y}|\mathbb{X}) + \tau, K(\boldsymbol{Y}|\mathbb{X})\}.$$

It is straightforward to check that $\mathcal{P}_L(\Theta) \subset \bigcup_{i=1}^{N}[L_i(\boldsymbol{Y}|\mathbb{X}), U_i(\boldsymbol{Y}|\mathbb{X})]$ and $U_i(\boldsymbol{Y}|\mathbb{X}) - L_i(\boldsymbol{Y}|\mathbb{X}) \leq \min\{\eta, K(\boldsymbol{Y}|\mathbb{X})\}$. From this, we can achieve the following upper bound

$$\|U_i - L_i\|_1 = \int_{\|\boldsymbol{Y}\| \leq 2M} |U_i(\boldsymbol{Y}|\mathbb{X}) - L_i(\boldsymbol{Y}|\mathbb{X})| d(\mathbb{X}, \boldsymbol{Y}) + \int_{\|\boldsymbol{Y}\| > 2M} |U_i(\boldsymbol{Y}|\mathbb{X}) - L_i(\boldsymbol{Y}|\mathbb{X})| d(\mathbb{X}, \boldsymbol{Y})$$

$$\leq K\tau + \exp\left(-\frac{K^2}{2\sigma^2}\right) \leq K'\tau,$$

where $K := \max\{2M, \sqrt{8\sigma^2}\} \log(1/\tau)$ and $K'$ be a positive constant. From the definition of bracket entropy, given that $H_B(K'\tau, \mathcal{P}_L(\Theta), \|\cdot\|_1)$ is the logarithm of the smallest number of bracket of size $K'\tau$ necessary to cover $\mathcal{P}_L(\Theta)$, we have

$$H_B(K'\tau, \mathcal{P}_L(\Theta), \|\cdot\|_1) \leq \log(N) = \log N(\tau, \mathcal{P}_L(\Theta), \|\cdot\|_1). \tag{26}$$

Now we want to bound the covering number $N$. Specifically, we denote $\Delta = \{\boldsymbol{B}_j, \boldsymbol{A}_j, c_j : (\boldsymbol{B}_j, \boldsymbol{A}_j, c_j) \in \Theta\}$. Because the parameter space $\Theta$ is compact, $\Delta$ is also a compact set. Thus, we can find $\tau$-covers $\Delta_\tau$ such that $|\Delta_\tau| \leq \mathcal{O}(\tau^{-2rdL-L})$.

For each mixing measure $G = \sum_{j=1}^{L} \exp(c_j)\delta_{(\boldsymbol{B}_j, \boldsymbol{A}_j)} \in \mathcal{G}_L$, we consider an other mixing measure as follows,

$$G' = \sum_{j=1}^{L} \exp(c'_j)\delta_{(\boldsymbol{B}'_j, \boldsymbol{A}'_j)}, \tag{27}$$

where $(c'_j, \boldsymbol{B}'_j, \boldsymbol{A}'_j) \in \Delta_\tau$ is the closest to $(c_j, \boldsymbol{B}_j, \boldsymbol{A}_j)$. We define some following intermediate functions to alleviate the bound:

$$f_1(\boldsymbol{X}) := \sum_{j=1}^{L} \text{softmax}\left(\boldsymbol{X}^\top m_{Q,j} \frac{\boldsymbol{C}_Q + \sigma_2(\boldsymbol{B}_j)\sigma_1(\boldsymbol{A}_j)}{\|\boldsymbol{C}_Q + \sigma_2(\boldsymbol{B}_j)\sigma_1(\boldsymbol{A}_j)\| + \tau_Q}\boldsymbol{X} + c_j\right)\left(m_{V,j}\frac{\boldsymbol{C}_V + \sigma_2(\boldsymbol{B}'_j)\sigma_1(\boldsymbol{A}'_j)}{\|\boldsymbol{C}_V + \sigma_2(\boldsymbol{B}'_j)\sigma_1(\boldsymbol{A}'_j)\| + \tau_V}\right)\boldsymbol{X},$$

$$f_2(\boldsymbol{X}) := \sum_{j=1}^{L} \text{softmax}\left(\boldsymbol{X}^\top m_{Q,j} \frac{\boldsymbol{C}_Q + \sigma_2(\boldsymbol{B}'_j)\sigma_1(\boldsymbol{A}'_j)}{\|\boldsymbol{C}_Q + \sigma_2(\boldsymbol{B}'_j)\sigma_1(\boldsymbol{A}'_j)\| + \tau_Q}\boldsymbol{X} + c'_j\right)\left(m_{V,j}\frac{\boldsymbol{C}_V + \sigma_2(\boldsymbol{B}'_j)\sigma_1(\boldsymbol{A}'_j)}{\|\boldsymbol{C}_V + \sigma_2(\boldsymbol{B}'_j)\sigma_1(\boldsymbol{A}'_j)\| + \tau_V}\right)\boldsymbol{X},$$

Now, we have,

$$\|f - f_1\|_\infty \leq \sum_{j=1}^{L} \sup_{\mathbb{X} \in \mathcal{X}} \text{softmax}\left(\boldsymbol{X}^\top m_{Q,j} \frac{\boldsymbol{C}_Q + \sigma_2(\boldsymbol{B}_j)\sigma_1(\boldsymbol{A}_j)}{\|\boldsymbol{C}_Q + \sigma_2(\boldsymbol{B}_j)\sigma_1(\boldsymbol{A}_j)\| + \tau_Q}\boldsymbol{X} + c_j\right) \cdot m_{V,j}\|\mathbf{V}_j\boldsymbol{X} - \mathbf{V}'_j\boldsymbol{X}\|$$

$$\leq \sum_{j=1}^{L} \sup_{\mathbb{X} \in \mathcal{X}} m_{V,j}\|\mathbf{V}_j\boldsymbol{X} - \mathbf{V}'_j\boldsymbol{X}\|$$

$$\leq \sum_{j=1}^{L} \sup_{\mathbb{X} \in \mathcal{X}} m_{V,j}\left\|\left(\frac{\boldsymbol{C}_V + \sigma_2(\boldsymbol{B}_j)\sigma_1(\boldsymbol{A}_j)}{\|\boldsymbol{C}_V + \sigma_2(\boldsymbol{B}_j)\sigma_1(\boldsymbol{A}_j)\| + \tau_V} - \frac{\boldsymbol{C}_V + \sigma_2(\boldsymbol{B}'_j)\sigma_1(\boldsymbol{A}'_j)}{\|\boldsymbol{C}_V + \sigma_2(\boldsymbol{B}'_j)\sigma_1(\boldsymbol{A}'_j)\| + \tau_V}\right)\boldsymbol{X}\right\|$$

$$\lesssim \sum_{j=1}^{L} \sup_{\mathbb{X} \in \mathcal{X}} \|(\boldsymbol{B}_j, \boldsymbol{A}_j) - (\boldsymbol{B}'_j, \boldsymbol{A}'_j)\| \cdot \|\mathbb{X}\|$$

$$\lesssim \sum_{j=1}^{L} \tau \cdot B \lesssim \tau, \tag{28}$$

where the second last inequality holds due to the fact that the input space is bounded: $\|X\| \leq B$ for all $\mathbb{X} \in \mathcal{X}$.

Similarly, we also have,

$$
\begin{aligned}
\|f_1 - f_2\|_\infty &\leq \sum_{j=1}^{L} \sup_{\mathbb{X} \in \mathcal{X}} \left| \mathrm{softmax}\left( X^\top m_{Q,j} \frac{C_Q + \sigma_2(B_j)\sigma_1(A_j)}{\|C_Q + \sigma_2(B_j)\sigma_1(A_j)\| + \tau_Q} X + c_j \right) \right. \\
&\qquad \left. - \mathrm{softmax}\left( X^\top m_{Q,j} \frac{C_Q + \sigma_2(B_j')\sigma_1(A_j')}{\|C_Q + \sigma_2(B_j')\sigma_1(A_j')\| + \tau_Q} X + c_j' \right) \right| \cdot m_{V,j} \|V_j' X\| \\
&\lesssim \sum_{j=1}^{L} \sup_{\mathbb{X} \in \mathcal{X}} \left| \mathrm{softmax}\left( X^\top m_{Q,j} \frac{C_Q + \sigma_2(B_j)\sigma_1(A_j)}{\|C_Q + \sigma_2(B_j)\sigma_1(A_j)\| + \tau_Q} X + c_j \right) \right. \\
&\qquad \left. - \mathrm{softmax}\left( X^\top m_{Q,j} \frac{C_Q + \sigma_2(B_j')\sigma_1(A_j')}{\|C_Q + \sigma_2(B_j')\sigma_1(A_j')\| + \tau_Q} X + c_j' \right) \right| \\
&\lesssim \sum_{j=1}^{L} \|(B_j, A_j, c_j) - (B_j', A_j', c_j')\| \\
&\lesssim \tau.
\end{aligned}
\tag{29}
$$

Therefore, by triangle inequality, we have $\|f - f_2\|_\infty \lesssim \tau$. Then, by noting that the Gaussian density function $f(x) = (2\pi\sigma^2)^{-d/2} \exp\left(-\|x\|^2/2\sigma^2\right)$ is a global Lipschitz function, we have,

$$
\|p(Y|f_G(\mathbb{X}), \sigma^2 I_d) - p(Y|f_{G'}(\mathbb{X}), \sigma^2 I_d)\|_1 \lesssim \|g_G(\mathbb{X}) - g_{\bar{G}}(\mathbb{X})\|_\infty \lesssim \tau.
\tag{30}
$$

From the definition of covering number, we get

$$
\log N(\tau, \mathcal{P}_{HL}(\Theta), \|\cdot\|_1) \leq |\Delta_\tau| \leq \mathcal{O}(\tau^{-2rdL-L})
\tag{31}
$$

From equations 26 and 31, we have

$$
H_B(\tau, \mathcal{P}_L(\Theta), \|\cdot\|_1) \lesssim \log(1/\xi).
$$

By choosing $\xi = \epsilon/2$, we achieve that

$$
H_B(\epsilon, \mathcal{P}_{HL}(\Theta), h) \lesssim \log(1/\epsilon).
$$

This concludes our proof.

## E  USE OF LARGE LANGUAGE MODELS

In this paper, large language models were utilized exclusively for editorial assistance, such as grammar correction and spelling improvements, and were not applied to content creation, data analysis, or experimental design.

