# OpenReview forum: "DoRAN: Stabilizing Weight-Decomposed Low-Rank Adaptation via  Noise Injection and Auxiliary Networks"
_ICLR.cc/2026/Conference — Submitted to ICLR 2026_

### Official Review · Reviewer_vatj · 2025-10-19

**Soundness:** 2
**Presentation:** 3
**Contribution:** 3
**Rating:** 6
**Confidence:** 4

**Summary:**

This paper proposes DoRAN, a stable and flexible extension of DoRA for parameter-efficient fine-tuning. It introduces a learnable noise term (τ) to stabilize normalization and a hypernetwork to generate low-rank matrices with cross-layer parameter sharing. Theoretical analysis shows that DoRAN improves optimization stability and sample efficiency, while experiments on both vision (ViT-B/16) and language (LLaMA-7B/13B) tasks demonstrate consistent gains over LoRA and DoRA with minimal parameter overhead.

**Strengths:**

1. The paper provides a clear and well-motivated improvement over DoRA by introducing a learnable noise term (τ) that stabilizes normalization and enables smooth interpolation between DoRA and LoRA behaviors.
2. The proposed hypernetwork design allows cross-layer parameter sharing, which helps reduce redundancy to some extent and may promotes diversity in learned representations.
3. The paper includes theoretical analysis that offers useful insights into the stability and convergence behavior of the proposed method, providing a clear theoretical perspective in addition to empirical evidence.

**Weaknesses:**

1. The experimental results need clearer attribution. It is essential to specify which results are taken from prior work and which are reproduced by the authors. If the baseline numbers are reproduced, detailed descriptions of their experimental settings should be provided to ensure fairness and reproducibility.
2. For the image classification experiments, both the proposed method and baseline results appear to have significantly fewer data points than prior related work [1, 2, 3, 4], which raises concerns about the reliability or completeness of the evaluation.
3. The experiments should include results on more recent and competitive large language models, such as LLaMA3-8B or Qwen-2.5-7B, to demonstrate broader applicability and scalability.
4. The paper should include a comprehensive comparison of computational efficiency between DoRAN, LoRA, and DoRA.

------

[1] MLAE: Masked LoRA Experts for Visual Parameter-Efficient Fine-Tuning. https://arxiv.org/abs/2405.18897

[2] Neural Prompt Search. https://arxiv.org/abs/2206.04673

[3] Towards Efficient Visual Adaption via Structural Re-parameterization. https://arxiv.org/abs/2302.08106

[4] Sensitivity-Aware Visual Parameter-Efficient Fine-Tuning. https://arxiv.org/abs/2303.08566

**Questions:**

1. In Appendix B.4 (Table 5), why are experiments not conducted on all projection layers (QUERY, KEY, VALUE, UP, and DOWN) as in the original DoRA, and why is KEY ablated instead? Please report DoRAN results when applied to the full set of projections (Q/K/V/up/down) and compare them directly to the corresponding results in the original DoRA paper to ensure completeness and fairness.
2. Please provide the training dynamics and final values of the learnable noise term (τ). In particular, report per-layer (and, if applicable, per-head) trajectories over training, the final distributions, and their correlations with optimization stability and performance. A brief analysis (e.g., plots of τ over steps, summary statistics, and variance across random seeds) would substantially strengthen the paper.

---

> ### Author Response · Authors · 2025-11-20
> **Response to reviewer Reviewer vatj (part 1)**
>
> Thank you for the constructive. We would like to address your concerns as follows:
>
> **W1: The paper needs clearer attribution on the results of the baselines.**
>
> **Answer:**
>
> We clarify that the baseline results for the vision tasks are taken from [1], except for DoRA, which we implement from scratch and reproduce on the vision benchmarks, and those for the language task are taken from [2]. To ensure a fair comparison, we matched the experimental settings across all baselines and our method. In particular, we applied the same data augmentation strategy, used identical training protocols, and set the rank $r$ in DoRAN to be the same as in DoRA and LoRA. We appreciate the reviewer’s suggestion and have made these details explicit in the revised version.
>
> **W2: Regarding the image classification experiments.**
>
> **Answer:** We assume the reviewer’s concern about “significantly fewer data points” refers to the VTAB-1K setting. We emphasize that this is the standard low-data evaluation protocol used in prior work—including [1]—where only 1,000 labeled examples per task are provided. This is not a limitation of our evaluation but an intentional benchmark designed to test data-efficient fine-tuning. To provide a more complete assessment, we also include experiments on the FGVC benchmark, which uses the full datasets rather than the 1K subset. As discussed in the main paper, the two benchmarks are complementary: VTAB-1K evaluates generalization in data-scarce regimes, while FGVC evaluates fine-grained recognition with full data. Together, they offer a comprehensive picture of DoRAN’s performance.
>
> **W3: The experiments should include results on more competitive large language models, such as LLaMA3-8B or Qwen-2.5-7B, to demonstrate broader applicability and scalability.**
>
> Thank you for the suggestion. To further assess the effectiveness of DoRAN on recent and competitive base models, we performed an additional experiment on LLaMA3-8B. As presented in the table below, DoRAN continues to offer clear benefits at this scale, achieving an average improvement of +0.6% over vanilla DoRA. We appreciate the reviewer’s input and have incorporated this new experiment into the revised manuscript.
>
> | Method | BoolQ | PIQA | SIQA | HellSwag | WinoGrande | ARC-e | ARC-c | OBQA | AVG |
> |---|---|---|---|---|---|---|---|---|---|
> | DoRA | **73.5** | 87.9 | 79 | 93.4 | 83.7 | 89.8 | **78.9** | 84.4 | 83.8 |
> | DoRAN | 72.9 | **88.7** | **80.5** | **94.7** | **84.1** | **90.3** | 78 | **85.6** | **84.4** |
>
> **W4: The paper should include a comprehensive comparison of computational efficiency between DoRAN, LoRA, and DoRA.**
>
> **Answer:** We appreciate the reviewer’s feedback. For completeness, besides the ratio of trainable parameters, we also present FLOPs and runtime comparisons for low-rank approaches including LoRA, DoRA, and DoRAN in the table below, as well as in the revision. Although DoRAN adds small hypernetwork modules that increase FLOPs during training, its runtime and proportion of trainable parameters remain almost the same as DoRA’s because these hypernetworks are lightweight and efficient to compute. Meanwhile, DoRAN delivers notable performance improvements (for example, +2.9% over LoRA and +2.06% over DoRA on LLaMA-7B). At inference time, all low-rank matrices are merged, so LoRA, DoRA, and DoRAN share identical FLOPs and latency, making DoRAN both practical and efficient for PEFT.
>
> |   Model   | Method | #Params(%) | GFLOPs | Runtimes | Performance |
> |:---------:|:------:|:----------:|:------:|:--------:|:-----------:|
> |  LLaMA-7B |  LoRA  |    0.25    |  6.63  |   4.17   |    74.09    |
> |           |  DoRA  |    0.25    |  25.42 |   5.22   |    74.94    |
> |           |  DoRAN  |    0.26    |  60.05 |   5.25   |    77.01    |
> | LLaMA-13B |  LoRA  |     0.2    |  12.88 |   7.25   |    80.18    |
> |           |  DoRA  |     0.2    |  49.58 |   7.37   |    80.05    |
> |           |  DoRAN |    0.21    | 117.21 |   7.48   |    80.76    |

---

> ### Author Response · Authors · 2025-11-20
> **Response to reviewer Reviewer vatj (part 2)**
>
> **Q1: The paper should include experiments where DoRAN is applied to all projection layers (QUERY, KEY, VALUE, UP, and DOWN)**
>
> **Answer:** In our main experiments, we apply DoRAN to the QUERY and VALUE projections to remain aligned with our theoretical analysis. In Appendix B.4, we explored extending DoRAN to the UP and DOWN projection layers to assess the broader applicability of the method, while still restricting the primary evaluation to QUERY and VALUE for theoretical consistency. We appreciate the reviewer’s suggestion, and in response, we have conducted additional ablation experiments where DoRAN is applied to the full set of projection layers. The results of these extended experiments are shown in the table below, and are also included in the revised manuscript.
>
> | Model | Method | BoolQ | PIQA | SIQA | HellSwag | WinoGrande | ARC-e | ARC-c | OBQA | AVG |
> |---|---|---|---|---|---|---|---|---|---|---|
> | LLaMA-7B | DoRA | 69.7 | 82.3 | **79.5** | **86.3** | 81.0 | 81.9 | 66.2 | 79.2 | 78.2 |
> |  | DoRAN | **70.2** | **83.8** | 78.6 | 85.0 | **81.1** | **82.0** | **66.7** | **80.4** | **78.5** |
> | LLaMA-13B | DoRA | **72.4** | 85.1 | 80.1 | 91.9 | 84.3 | **84.8** | 70.9 | 82.8 | 81.5 |
> |  | DoRAN | 72.2 | **86.1** | **80.4** | **92.9** | **84.9** | 84.6 | **71.8** | **83.8** | **82.1** |
>
> **Q2: Final values of $\tau$ and its dynamic:**
>
> **Answer:** Following the reviewer’s suggestion, we added violin plots of the learned $\tau$ values across the 12 ViT layers—both after the first 10 epochs and at the end of training. These plots are now included in the appendix of the revised manuscript (Figure 4 and 5).
>
> At epoch 10, as shown in the Figure 4 of the appendix, except the last layer, the distribution of $\tau$ values is relatively uniform across layers: both the mean and variance remain relatively similar throughout the depth of the network. This indicates that, early in training, the model has not yet differentiated the roles of different layers in terms of how strongly they should rely on direction learning versus norm control.
>
> By contrast, in the final epoch, as shown in the Figure 5 of the appendix, clear structural patterns emerge. The early layers exhibit larger variance in $\tau$, suggesting that these layers learn more diverse or specialized transformations and therefore benefit from a wider range of direction-vs-norm tradeoffs. The later layers, in comparison, converge to much tighter $\tau$ distributions, suggesting that they operate in a more homogeneous regime.
>
> Moreover, the mean $\tau$ values increase steadily over training. This trend shows that the model gradually shifts toward stronger direction learning rather than relying solely on norm control, differentiating DoRAN from DoRA and highlighting the additional flexibility introduced by the $\tau$ parameter. Overall, these results support our claim that $\tau$ adapts meaningfully across depth and provides structural benefits.
>
> Last but not least, we sincerely appreciate your constructive feedback, which greatly supports our revisions.
>
> **References:**
>
> [1] V-PETL Bench: A Unified Visual Parameter-Efficient Transfer Learning Benchmark. NeurIPS, 2024.
>
> [2] DoRA: Weight-Decomposed Low-Rank Adaptation. ICML, 2024.

---

> > ### Comment · Reviewer_vatj · 2025-11-23
> >
> > Thank you for the detailed response. Most of my questions have been clarified. However, I would like to understand how DoRAN performs compared with some of the strongest existing methods in the vision domain. I would appreciate it if the authors could reproduce the results of the methods I mentioned, such as MLAE, under your own experimental setup and provide a direct comparison. It would also be very helpful if the authors could report the full hyperparameter configurations used for reproducing MLAE, including settings such as the expert drop rate and the scale coefficient, consistent with those described in the original paper.

---

> ### Author Response · Authors · 2025-11-25
>
> Dear Reviewer vatj,
>
> Thanks for your response. We are glad to hear that most of your questions have been clarified. Next, let us address your concerns regarding how DoRAN performs compared with baselines in the vision domain, particularly MLAE.
>
> **The configurations for comparing with MLAE:**
> Following the reviewer’s suggestion, we conducted additional experiments to compare DoRAN with MLAE. To ensure a fair comparison among low-rank adaptation methods reported in our paper, we emphasize that we apply low-rank fine-tuning to the query and value projection weights for MLAE in both vision and language tasks. This differs from the original implementations of the MLAE paper, which apply low-rank updates to all the projection matrices including the query, key, and value weights.
>
> Because the original MLAE paper does not provide hyperparameters and optimization configurations on the FGVC benchmark, we adopt a dropout rate of 0.5 and set the coefficient initialization value to 1.0 for MLAE on this benchmark. All other training settings—including the learning rate, weight decay, and other optimization configurations—are kept consistent with those used in our proposed method. For the VTAB-1K experiments, we follow the hyperparameters and optimizer settings reported in the original paper.
>
> For the Commonsense Reasoning task, we reuse the FGVC hyperparameters for reproducing MLAE: a dropout rate of 0.5 and a coefficient initialization value of 1.0.
>
> **Comparision results:** The results are summarized in the tables below. These results show that although MLAE achieves a 0.5\% performance gain over DoRAN on FGVC, DoRAN surpasses MLAE by 1.7\% on the VTAB-1K benchmark and further outperforms it on the language tasks by 0.91\% for LLaMA-7B and 1.04\% for LLaMA-13B. This demonstrates that DoRAN is competitive with state-of-the-art approaches including MLAE, while also exhibiting strong flexibility and applicability across multiple domains.
>
> **Table 1: Image Classification Results on FGVC Datasets**
>
> | Method        | CUB-200-2011 | NABirds | Oxford Flowers | Stanford Dogs | Stanford Cars | Average |
> |---------------|--------------|---------|----------------|----------------|----------------|---------|
> | LoRA          | 84.6         | 78.2    | 98.9           | 85.1           | 77.1           | 84.78   |
> | DoRA          | 87.3         | 80.0    | 99.1           | 87.6           | 81.9           | 87.18   |
> | MLAE          | **89.5**         | **85.5**    | 99.2           | **91.2**           | **84.4**           | **90.0**    |
> | **DoRAN**     | 88.5     | 85.3| **99.2**       | 90.8       | 83.7           | 89.5 |
>
> --------------------------------------
>
>
>
> **Table 2: Image Classification results on the VTAB-1K dataset**
>
> | **Method** | **CIFAR100** | **Caltech101** | **DTD** | **Flower102** | **Pets** | **SVHN** | **Sun397** | **Camelyon** | **EuroSAT** | **Resisc45** | **Retinopathy** | **Clevr-Count** | **Clevr-Dist** | **DMLab** | **KITTI** | **dSpr-Loc** | **dSpr-ori** | **sNORB-Azim** | **sNORB-Ele** | **AVG** |
> |-----------|--------------|----------------|---------|----------------|----------|----------|------------|--------------|-------------|--------------|------------------|------------------|----------------|-----------|-----------|--------------|--------------|----------------|----------------|--------|
> | LoRA      | 67.1 | 91.4 | 69.4 | 98.2 | 90.4 | 85.3 | 54.0 | 84.9 | 95.3 | 84.4 | 73.6 | 82.9 | 69.2 | 49.8 | 78.5 | 75.7 | 47.1 | 31.0 | **44.0** | 72.2 |
> | DoRA      | 67.9 | 90.4 | 70.6 | 99.0 | 90.2 | 89.6 | 54.6 | 83.9 | 95.5 | 85.3 | 75.9 | 80.8 | 69.8 | 50.5 | 80.9 | 79.1 | 47.7 | 32.5 | 39.6 | 72.8 |
> | MLAE      | 71.1 | 91.4 | **72.7** | 99.1 | **91.8** | 87.8 | **55.7** | 83.0 | 95.7 | 83.7 | 75.8 | 81.3 | 69.1 | 50.6 | 79.9 | 71.9 | 44.9 | 28.7 | 41.2 | 72.4 |
> | **DoRAN** | **70.2** | **92.4** | 71.9 | **99.1** | 91.3 | **90.3** | 55.0 | **86.9** | **96.4** | **87.4** | **76.4** | **84.0** | **70.0** | **53.2** | **81.9** | **79.9** | **49.2** | **35.6** | 42.0 | **74.4** |
>
> -------------------------------------------
>
> *(To be continued)*

---

> ### Author Response · Authors · 2025-11-25
>
> **Table 3: Performance on the Commonsense Reasoning Task**
>
> | Model | Method | **#Params (\%)** | BoolQ | PIQA | SIQA | HellaSwag | WinoGrande | ARC-e | ARC-c | OBQA | Average |
> |-------|--------|------------------|--------|--------|--------|-------------|-------------|---------|---------|--------|----------|
> | **LLaMA-7B** | LoRA | 0.25 | 67.2 | 79.4 | 76.6 | 78.3 | 78.4 | 77.1 | 61.5 | 74.2 | 74.09 |
> | ~ | DoRA | 0.25 | 67.22 | 79.98 | 76.66 | 80.66 | **79.72** | 79.5 | 61.01 | 74.8 | 74.94 |
> | ~ | MLAE | 0.25 | 69.02 | **81.18** | 77.02 | 78.8 | 80.51 | **80.6** | 63.05 | 78.6 | 76.1 |
> | ~ | **DoRAN** | 0.26 | **69.82** | 81.01 | **77.89** | **83.45** | 79.56 | 79.76 | **65.02** | **79.6** | **77.01** |
> | **LLaMA-13B** | LoRA | 0.2 | 71.7 | 82.4 | 79.6 | **90.4** | **83.6** | 83.1 | 68.5 | 82.1 | 80.18 |
> | ~ | DoRA | 0.2 | **72.2** | 83.19 | **80.81** | 88.92 | 81.93 | 82.95 | **69.37** | 81 | 80.05 |
> | ~ | MLAE | 0.2 | 70.12 | 83.13 | 79.22 | 89.04 | 81.77 | 83.25 | 67.83 | 83.4 | 79.72 |
> | ~ | **DoRAN** | 0.21 | 71.8 | **84.5** | 80.6 | 89.79 | 83.19 | **83.29** | 68.69 | **84.2** | **80.76** |
>
> **Future direction.** To further highlight the flexibility of DoRAN, we note that DoRAN can be naturally combined with MLAE. In MLAE, the stochastic masking mechanism is implemented as
>
> $$\Delta W = M  \odot  \Lambda  \odot  \varepsilon$$
>
> Equivalently, for each layer $l \in [L]$, the forward computation can be written as
>
> $$Y = Y_{\text{pretrained}} + Y_{\text{MLAE}},  Y_{\text{MLAE}} = x A^\top \text{diag}(\text{Dropout}_{p}(\lambda))B$$
>
> Importantly, MLAE only introduces a masking mechanism — it does not modify the architecture of the underlying low-rank matrices. This makes it straightforward to incorporate MLAE into DoRAN. A natural combination strategy is to first generate the low-rank matrices $A$ and $B$ using the DoRAN hypernetworks, and then apply MLAE’s masked update within DoRAN’s normalized forward pass:
>
> $$Y_{\text{DoRAN + MLAE}} = x \frac{W_0 + W'}{||W_0 + W'||+\tau}, W' = A^\top \text{diag}(\text{Dropout}_{p}(\lambda))B$$
>
> We believe this hybrid approach could leverage the strengths of both DoRAN and MLAE. Exploring this direction is promising and we leave a thorough investigation of its benefits for future work.
>
> Best regards,
>
> The Authors

---

> > ### Comment · Reviewer_vatj · 2025-11-26
> >
> > Thank you very much for your response and for the considerable effort put into the additional experiments.
> > Could you share the specific hyperparameters used to reproduce MLAE in your LLaMA experiments, such as the expert drop rate?
> >
> > Moreover, your idea of combining MLAE with DoRAN is highly insightful. I would recommend citing this work in the manuscript and incorporating the supplementary experimental results into the appendix.
> >
> > Overall, I maintain my positive evaluation of the paper.

---

> > > ### Author Response · Authors · 2025-11-26
> > >
> > > Dear Reviewer vatj,
> > >
> > > Thanks for your response. Below is the hyperparameter configuration we used to reproduce MLAE in our additional experiments.
> > >
> > > ------------------------
> > > **The configurations for comparing with MLAE:** Following the reviewer’s suggestion, we conducted additional experiments to compare DoRAN with MLAE. To ensure a fair comparison among low-rank adaptation methods reported in our paper, we emphasize that we apply low-rank fine-tuning to the query and value projection weights for MLAE in both vision and language tasks. This differs from the original implementations of the MLAE paper, which apply low-rank updates to all the projection matrices including the query, key, and value weights.
> > >
> > > Because the original MLAE paper does not provide hyperparameters and optimization configurations on the FGVC benchmark, we adopt a dropout rate of 0.5 and set the coefficient initialization value to 1.0 for MLAE on this benchmark. All other training settings—including the learning rate, weight decay, and other optimization configurations—are kept consistent with those used in our proposed method. For the VTAB-1K experiments, we follow the hyperparameters and optimizer settings reported in the original paper.
> > >
> > > For the Commonsense Reasoning task, we reuse the FGVC hyperparameters for reproducing MLAE: a dropout rate of 0.5 and a coefficient initialization value of 1.0.
> > >
> > > ------------------------
> > >
> > > We have cited the work [1] and added these experiments to Appendix C in the revised manuscript. We hope that our responses completely address your concerns regarding our paper. Please feel free to let us know if you have any further comments. **Otherwise, we hope that the reviewer can consider increasing your final rating, which we really appreciate.**
> > >
> > > **References**
> > >
> > > [1] Junjie Wang, Guangjing Yang, Wentao Chen, Huahui Yi, Xiaohu Wu, Zhouchen Lin, and Qicheng Lao. Mlae: Masked lora experts for visual parameter-efficient fine-tuning. arxiv preprint arxiv 2405.18897, 2024.

---

### Official Review · Reviewer_S9Kw · 2025-10-26

**Soundness:** 2
**Presentation:** 3
**Contribution:** 1
**Rating:** 4
**Confidence:** 4

**Summary:**

This paper proposes DoRAN method that extends DoRA to improve training stability and sample efficiency. The main contributions include:

1. Stabilized weight decomposition with learnable noise, mitigating DoRA’s instability.

2. Hypernetwork-based low-rank reparameterization enabling parameter coupling across layers.

3. Theoretical and empirical proof of improved sample efficiency and fine-tuning robustness.

**Strengths:**

1. Provide theoretic analysis on gradients to show that the learnable noise $\tau$ can ensure bounded gradients

**Weaknesses:**

1. Compared to other LoRA variants, one major issue with DoRA-like approaches is their high computational cost in both GPU memory and training time. The primary reason is that computing the normalization vector $\\|W_0+BA\\|_c$ requires a full matrix $BA\in \mathbb{R}^{m\times n}$, even though it has low rank as $A\in \mathbb{R}^{r \times n}$ and $B\in \mathbb{R}^{m\times r}$ with $r \ll \mathrm{min}(m,n)$. This inefficiency can be observed by conducting the natural language generation (NLG) task on the MetaMathQA dataset; see [PiSSA] for experimental details.

2. Theoretically, DoRA may suffer from instability when $\\|W_0+BA\\|_c$ becomes small, yet the authors provide no empirical evidence to support this argument. It is possible that the observed instability arises from the unbounded scaling vector $m$.

3. The motivation for parameter sharing across layers is not well justified. Even if such sharing has potential benefits, it remains unclear why a simple one-layer MLP should be used to generate $A,B$.

4. Similar to DoRA, a recent method, [DeLoRA], also introduces a normalization term: $$ y=W_0x+ B\Sigma A x$$ where $\Sigma\in \mathbb{R}^{r\times r}$ is a positive diagonal matrix which normalize the rows of $A$ and the columns of $B$. Theoretically, DeLoRA could suffer from the same instability discussed in this paper; however, it has been found empirically stable over a wide range of learning rates.

5. The paper claims that the proposed approach reduces to LoRA when $\tau$ is large. This claim is incorrect. For large $\tau$, the proposed approach becomes $$W=\frac{m}{\tau} (W_0+BA)$$ which fundamentally differs from LoRA’s formulation, $$W=W_0+\frac{\alpha}{\gamma}BA.$$ In LoRA, $BA$ often has much smaller norm than $W_0$, ensuring that pre-trained knowledge is largely preserved. In contrast, the proposed approach applies a learnable scaling to $W_0$, potentially causing a large weight shift $\Delta W:=W - W_0$.

6. The proposed method needs additional experiments to validate its effectiveness, such as NLG evaluations on the MetaMathQA dataset (as in [PiSSA]), tests on non-LLaMA models, and experiments with larger base model.

[PiSSA] Meng et al. PiSSA: Principal singular values and singular vectors adaptation of large language models. NeurIPS 2024.

[DeLoRA] Bini et al. DeLoRA: Decoupling angles and strength in low-rank adaptation. ICLR 2025.

**Questions:**

N/A

---

> ### Author Response · Authors · 2025-11-20
> **Response to Reviewer S9Kw (part 1)**
>
> Thank you for your constructive feedback. We appreciate the opportunity to clarify and address your concerns as outlined below. We hope that our responses address your concerns, and finally convince you to raise your score.
>
> **W1: Vanilla DoRA requires computing the norm of a full matrix, and can be computationally inefficient in practice. Henceforth, DoRAN may inherit this weakness.**
>
> **Answer:** We acknowledge that the original DoRA formulation incurs additional computational overhead due to the normalization term, which requires normalizing the full matrix $||W_0+BA||$. This limitation affects DoRA and all methods that operate on its normalized parameterization.
>
> However, the reviewer’s concern does not undermine the core contribution of DoRAN, because our work addresses a different and orthogonal limitation of DoRA. The motivation behind DoRAN is not to reduce DoRA’s computational cost, but to improve DoRA’s training stability and sample inefficiency. Our theoretical analysis and empirical results show that DoRAN directly resolves these issues through its stabilization term $\tau$ and shared hypernetwork, yielding significantly improved performance—even outperforming full fine-tuning in some language settings. Thus, while DoRAN inherits DoRA’s computational characteristics, it substantially improves DoRA’s effectiveness, making it a strictly more capable and robust method. Computational efficiency and training effectiveness are distinct concerns, and we directly targets the latter. We also acknowledge that approaches aimed at improving DoRA’s computational footprint are complementary to our contribution and can be integrated with DoRAN in future work. We have clarified this distinction and the motivation behind DoRAN in the revised version.
>
> **W2, W4: lack practical evidence of instabilities arise from a small norm $||W_0+BA||_c$, hence lack practical motivation for introducing $\tau$.**
>
> **Answer:** We would like to clarify that addressing the small norm $||W_0 + BA||_c$ is **not** the sole and major motivation of introducing the noise term $\tau$. In fact, the noise term $\tau$ has two additional benefits that we have mentioned in the original version, including:
>
> + **Improved sample efficiency:** In Theorem 1, we demonstrate that the convergence rates of estimating low-rank matrices in DoRA are significantly slow, indicating that we need exponentially many data points to estimate these matrices. The main reason for this issue is the interaction among low-rank matrices expressed via the partial differential equation in Eq.(5). To overcome this problem, we introduce a shared structure with added noise in Section 4.2. Under this setting, the previous low-rank matrix interaction does not occur, that is, the partial differential equation in Eq.(5) no longer holds true. As a result, we show in Theorem 2 that the sample complexity of estimating low-rank matrices is substantially reduced from exponential order to polynomial order. This highlights the benefits of the noise term in improving the model's sample efficiency.
>
> + **Training stability through well-behaved gradient dynamics:** As shown in our gradient analysis, $\tau$ stabilizes training in *two* complementary ways. **First,** it prevents singularities when the normalization term becomes small, which is the scenario highlighted in the reviewer comment. **Second,** as shown in the *gradient analysis* section, $\tau$ ensures that the *parallel component of the gradient is preserved*, rather than being suppressed as in vanilla DoRA. This enables the low-rank factors (A) and (B) to co-adapt both their *direction* and *norm*, leading to significantly more stable optimization. This addresses a source of instability that cannot be attributed solely to the scaling vector; it is inherent to the normalization structure of DoRA.
>
> Together, these motivations indicate that $\tau$ is not simply a safeguard against small norms, but a principled design choice that improves both the *stability* and *sample efficiency* of DoRA-style parameterizations.
>
> **W3: The design of parameter-sharing is not well-motivated, and it's unclear why a shared network should be used to generate $A, B$.**
>
> **Answer:** We first clarify that we do *not* use a shared hypernetwork to generate $A$ and $B$, since their roles fundamentally differ, so each has its own hypernetwork. Additionally, we do not apply any cross-layer sharing mechanism. What is shared is the *input and first layer* of the hypernetworks across the query and value projections. This sharing mechanism between the query and value project matrices is motivated by our theoretical analysis, specifically Section 4.2, which shows that sharing structure between the query and value projections significantly improves *sample efficiency*, improving the rate from exponential to polynomial.

---

> > ### Comment · Reviewer_S9Kw · 2025-11-27
> >
> > Thanks the authors for further explanation. I understand that the main contributions of this work are to solve some technique issues of DoRA.  Although some empirical evidences show that DoRA-like method can achieve better performance than LoRA, it has some potential drawbacks which is hard to address due to the nature of model parameterization.
> >
> > 1. Since DoRA or DoRAN involves $W=m\frac{W_0+B^\top A}{\\|W_0+B^\top A\\|+\tau}$, then the weight update $\Delta W=W-W_0$ is **likely to be full rank**. Then, all the nice regularization properties of low-rank update are lost. For LLM applications, it could be possible that more model forgetting happens while little extra adaptation performance is achieved. Another example is the sample efficiency issue mentioned in this work. Does LoRA also suffers from the same issue or it is only associated with the DoRA-like parameterization?
> >
> > 2. It needs much more GPU memory. Since LLM is one major application of LoRA variants,  and the current situation is that affordable computational resource is way behind the model scale. Taking 70B model for example, if one can do fine tuning with relatively cheap hardware, why not spend more time on hyper-parameter tuning?

---

> ### Author Response · Authors · 2025-11-20
> **Response to Reviewer S9Kw (part 2)**
>
> **W5: for large $\tau$, the update formular is not equivalent to LoRA.**
>
> **Answer:** We agree with the reviewer that, when $\tau$ is extremely large, the update approaches
>
> $$W \approx \frac{m}{\tau}(W_0 + BA),$$
>
> which is *not* identical to LoRA. However, in the main text, we explicitly describe this regime as *LoRA-like* to highlight that the parameterization transitions toward a linear low-rank form, rather than to claim exact equivalence.
>
> We also agree with the reviewer that when $\tau$ is extremely large and the previous approximation holds, the term $\frac{m}{\tau}$ could be too far from $1$, which could in principle induce a large shift from the pretrained weights. However, this limiting case is *not encountered in practice*, since $\tau$ never becomes large enough for this approximation to dominate and stay in a moderate range of less than $0.5$ in our experiments. Thus, while the reviewer is correct about the exact mathematical limit, this regime is not observed empirically, and DoRAN behaves as intended in all realistic training settings.
>
> **W6: Limited experimental designs to validate the effectiveness of DoRAN:**
>
> **Answer:** We would like to clarify the scope of our experiments and the rationale behind our choices. Since DoRAN is designed as an extension of DoRA, we closely follow DoRA’s original experimental protocol for language tasks to ensure a fair and consistent comparison. In addition, to address concerns about generality beyond LLaMA-style architectures, we include experiments on FGVC and VTAB-1K, following the vision-focused parameter-efficient fine-tuning benchmarks at [1]. These experiments evaluate DoRAN on non-LLaMA models, specifically the Vision Transformer (ViT) architecture, demonstrating that DoRAN extends naturally to modalities and backbones beyond language.
>
> Regarding MetaMathQA, we have not included results on this benchmark in the current version. However, in response to the reviewer’s suggestion, we have expanded our comparison against PiSSA, the method evaluated on MetaMathQA, on our current settings in revised manuscript (Table 1 for VTAB-1K and Table 2 for FGVC) and the table below (for FGVC dataset).
>
> | Method   | #Params (%) | CUB  | NAB  | Flowers | Dogs | Cars | Avg   |
> |----------|-------------|------|------|---------|------|------|-------|
> | PiSSA    | 0.55        | 87.1 | 80.5 | 99.1   | 84.8| 82.3 |  86.7    |
> | DoRAN    | 0.66        | 88.5 | 85.3 | 99.2    | 90.8 | 83.7 | 89.5  |
>
> Moreover, based on your suggestion, to evaluate the method on more competitive models, we have conducted an additional experiment to compare DoRAN agaisnt DoRA on LLaMA3-8B. The results are presented below and have been included in the revision (Table 8).
>
> | Method | BoolQ | PIQA | SIQA | HellSwag | WinoGrande | ARC-e | ARC-c | OBQA | AVG |
> |---|---|---|---|---|---|---|---|---|---|
> | DoRA | **73.5** | 87.9 | 79 | 93.4 | 83.7 | 89.8 | **78.9** | 84.4 | 83.8 |
> | DoRAN | 72.9 | **88.7** | **80.5** | **94.7** | **84.1** | **90.3** | 78| **85.6** | **84.4** |
>
> We agree that further evaluations—such as experiments on MetaMathQA, additional non-LLaMA models, and larger base models—would strengthen the paper, and we consider these valuable directions for future work.
>
> Lastly, we truly appreciate your feedback, it has played an important role in strengthening our revision.
>
> **References:**
>
> [1] V-PETL Bench: A Unified Visual Parameter-Efficient Transfer Learning Benchmark. NeurIPS, 2024.

---

> > ### Author Response · Authors · 2025-11-25
> > **Looking forward to your response**
> >
> > Dear Reviewer S9Kw,
> >
> > We would like to thank you again for your thorough review, and we hope that our response successfully addresses your concerns regarding our manuscript. However, since the discussion period is expected to end soon, please feel free to let us know if you have any further concerns. We would be willing to address any additional concerns from you. Otherwise, we hope that you will consider increasing your final rating.
> >
> > Thank you so much for your time and effort spent on the paper, we really appreciate it!
> >
> > Best regards,
> >
> > The Authors

---

> > ### Comment · Reviewer_S9Kw · 2025-11-27
> >
> > - W5
> >
> > From the reviewer's viewpoint, it is far from a LoRA-like method even when $\tau$ is large. In fact, there are some fundamental differences. LoRA preserves the pretrained weight $W_0$ and only adds low-rank update, which impose a strong regularization, especially when the rank is small. However, DoRA-like parameterization involves full-rank update. It is more likely involving weight overwrite, although one may get better adaptation performance.
> >
> > - W6
> >
> > Thank you for providing further experiments on Vision Benchmarks. However, it does not address my main concerns about DoRA-like methods, which is the scalability to larger LLM (e.g., 70B) and diversity of base models (non-Llama family).

---

> ### Author Response · Authors · 2025-11-30
> **Response to Reviewer S9Kw (part 3)**
>
> Dear Reviewer S9Kw,
>
> Thanks for your response. Let us address your remaining concerns as follows.
>
> **Q1:**
>
> **Regarding the rank of DoRA and DoRAN:** We agree with the reviewer that the update of DoRA and DoRAN can in general be full rank, and this property is inherited by DoRAN as well. However, this property is not necessarily inherently detrimental. Instead, it gives the method additional flexibility for adaptation.
>
> While low rank can indeed provide useful regularization, recent works have also shown that choosing a rank that is too small can severely limit expressiveness when the true underlying update has higher effective rank. This is precisely why prior methods such as AdaLoRA [2] and HiRA [3] introduce mechanisms to adapt the rank instead of fixing it a priori. In a similar spirit, although DoRA does not explicitly search for an “optimal” rank, the fact that it is not constrained to a specific rank allows it to better approximate the true update when higher expressiveness is beneficial. In other words, DoRA trades a hard rank constraint for additional degrees of freedom, which can be advantageous in practice rather than a weakness.
>
> **In terms of forgetting,** full-rank updates do not automatically imply more forgetting. In our setting, the update is still generated through low-rank factors $A,B$ and is modulated by the normalization term in the denominator, which acts as an implicit regularizer. Empirically, in our experiments on LLM finetuning, we do not observe increased forgetting for DoRA/DoRAN relative to standard low-rank approaches at comparable adaptation performance.
>
> **Regarding the sample efficiency.** We confirm that the sample-efficiency issue is not unique to DoRA; standard LoRA also suffers from the same limitation. Truong et al. [1, Theorem 5.1] show that estimating low-rank matrices in LoRA to a given accuracy can require an exponential number of samples. In Theorem 1 of our manuscript, we prove that DoRA inherits a similar exponential sample-complexity behavior.
>
> To address this, we introduce DoRAN, which is specifically designed to improve sample efficiency. Concretely, DoRAN (i) injects noise into the denominator of DoRA’s weight decomposition, acting as an adaptive regularizer that mitigates instabilities, and (ii) replaces static low-rank matrices with auxiliary networks that generate them dynamically, encouraging shared structure between the query and value matrices. As shown in Theorem 2, these modifications improve the sample complexity of DoRAN from exponential to polynomial order, thereby directly tackling the sample-efficiency issue raised by the reviewer.
>
> **Q2:**
>
> Thank you for the insightful comment. We agree that incorporating the norm component in DoRA and DoRAN can, in principle, increase GPU memory consumption compared to standard LoRA.
>
> However, following Section 4.3 of the original DoRA paper [4], we adopt the same practical trick of **detaching the norm term** $|| W_0 + B^\top A ||$ from the gradient computation graph. Concretely, the norm is treated as a constant during backpropagation, so gradients are not propagated through this term. This design substantially reduces the memory footprint of the computation graph while retaining the benefits of the reparameterization. The original DoRA work reports that this detachment yields **no noticeable degradation in accuracy** [4], and we observe the same behavior in our experiments. As a result, only the magnitude scalar and the term $W_0 + B^\top A$ remain in the gradient graph, which keeps the additional memory overhead moderate relative to vanilla LoRA.
>
> Regarding the reviewer’s point that, given limited hardware, one might instead invest more effort in hyperparameter tuning: we view our method as **complementary** to such tuning rather than a substitute. Even with extensive hyperparameter search, standard LoRA remains constrained by its parameterization. In contrast, DoRAN aims to improve the *intrinsic* sample efficiency and stability of DoRA-type updates. In settings where resources are tight (e.g., fine-tuning large LLMs on relatively small or noisy datasets), improving the underlying parameterization can yield gains that are difficult to achieve by hyperparameter tuning alone.
>
> **References**
>
> [1] Tuan Truong, Chau Nguyen, Huy Nguyen, Minh Le, Trung Le, and Nhat Ho. RepLoRA: Reparameterizing low-rank adaptation via the perspective of mixture of experts. In ICML, 2025.
>
> [2] Qingru Zhang, Minshuo Chen, Alexander Bukharin, Nikos Karampatziakis, Pengcheng He, Yu Cheng, Weizhu Chen, Tuo Zhao. AdaLoRA: Adaptive Budget Allocation for Parameter-Efficient Fine-Tuning. In ICLR, 2023.
>
> [3] Huang, Qiushi, et al. HiRA: Parameter-efficient hadamard high-rank adaptation for large language models. In ICLR, 2025.
>
> [4] Liu, Shih-Yang, et al. Dora: Weight-decomposed low-rank adaptation. In ICML, 2024.

---

> ### Author Response · Authors · 2025-11-30
> **Response to Reviewer S9Kw (part 4)**
>
> **W5:**
>
> We thank the reviewer for pointing out this potential source of confusion. As demonstrated in the previous response, our use of the term *“LoRA-like”* was only intended to convey that, **in the limit of large $\tau$**, the resulting update can be seen as a direction update with a scaling factor. It was *not* meant to suggest that DoRA preserves $W_0$ in the same way as standard LoRA, nor that it inherits all of LoRA’s regularization properties. We apologize for the misleading wording.
>
> To avoid this confusion, we have removed the phrase *“LoRA-like”* from the revised manuscript. Instead, we now explicitly emphasize that $\tau$ controls a trade-off between **norm control** and **directional updating**, without framing the method as LoRA-style for large $\tau$.
>
> Moreover, the regime of this direction update with a scaling factor only happens **in theory for very large $\tau$**, and is **not** observed in practice. As shown in Figure 5 of the revision which visualize the distributions of all $\tau$ across layers of the models, the learned $\tau$ values typically remain below 2, so the theoretical large-$\tau$ behavior raised in the comment does not occur in our empirical settings.
>
> **W6:**
>
> Thank you for raising this point. We acknowledge that our current experiments do not fully address scalability to very large LLMs (e.g., 70B) or to a broader set of base architectures beyond the LLaMA family.
>
> Due to computational and budget constraints, we are currently unable to run experiments on models at the 70B scale. For the commonsense reasoning setup, we therefore restrict our study to LLaMA models—LLaMA-7B, LLaMA-13B, and LLaMA3-8B—in order to ensure a fair and controlled comparison, as the original DoRA paper also evaluates exclusively on these LLaMA variants. Our goal in this work is to analyze and improve DoRA under the same conditions, so we align our experimental protocol accordingly.
>
> That said, we agree that demonstrating robustness across (i) larger parameter scales and (ii) non-LLaMA architectures would further strengthen the claims. Conceptually, our method is a local reparameterization of weight matrices and does not rely on properties specific to the LLaMA family, so we do not foresee any fundamental obstacle to applying it to other LLM architectures. Moreover, our additional experiments on diverse vision backbones provide some evidence that the proposed mechanisms are not tied to a single model family.
>
> We have added a discussion of this limitation and explicitly highlighted large-scale (e.g., 70B) and non-LLaMA evaluations as an important direction for future work.

---

### Official Review · Reviewer_Tvs9 · 2025-10-30

**Soundness:** 2
**Presentation:** 3
**Contribution:** 2
**Rating:** 6
**Confidence:** 4

**Summary:**

This paper proposes DoRAN, a Parameter-Efficient Fine-Tuning (PEFT) method that improves upon Weight-Decomposed Low-Rank Adaptation (DoRA). The authors identify two key limitations in DoRA: potential optimization instability when weight norms are small, and inefficient parameterization by learning independent low-rank matrices for each layer. DoRAN introduces two main components: (i) a learnable noise term injected into the denominator of the weight normalization step, which smoothly interpolate between DoRA's directional updates and LoRA's linear scaling; and (ii) the use of auxiliary networks (hypernetworks) to generate the low-rank adaptation matrices. This second component induces shared structure across layers and attention heads, aiming to improve parameter and sample efficiency. The authors provide a theoretical analysis, framing the method through a Mixture-of-Experts (MoE) perspective, to show that DoRAN achieves a better (polynomial) sample complexity compared to the exponential rate of non-shared structures. The method is evaluated on image classification and commonsense reasoning tasks, demonstrating performance gains over DoRA and LoRA.

**Strengths:**

Although the idea of interpolating between LoRA and DoRA's weight approximation is simple, it proves effective experimentally.

Using hyper-networks to predict the weight update is also a straightforward strategy to induce correlations across layers.

The authors theoretically support DoRAN's strategy which further supports experimental findings.

**Weaknesses:**

Algorithmic novelty is limited as interpolating between LoRA and DoRA is a straightforward strategy and using hyper-networks for weight prediction has already been studied for PEFT [1]

Comparison with other sota PEFT algorithms is not performed. Since DoRA can theoretically be applied on top of other recent PEFT algorithms [2,3] having a study of whether the DoRAN strategy further improves these variants would have been a strong addition to the paper.

The paper only quantifies the increase in trainable parameters. There is no discussion of the practical impact on training time, memory, or FLOPs introduced by the hypernetwork's forward pass at every step.

[1] Mahabadi, Rabeeh Karimi, et al. "Parameter-efficient multi-task fine-tuning for transformers via shared hypernetworks." Proceedings of the 59th Annual Meeting of the Association for Computational Linguistics and the 11th International Joint Conference on Natural Language Processing, 2021.

[2] Meng, Fanxu, Zhaohui Wang, and Muhan Zhang. "Pissa: Principal singular values and singular vectors adaptation of large language models." Advances in Neural Information Processing Systems, 2024.

[3] Zhang, Qingru, et al. "Adalora: Adaptive budget allocation for parameter-efficient fine-tuning." ICLR, 2023

**Questions:**

Hyper-network predicted weights and interpolation coefficients prove effective for vision tasks but gains due to the hypernetwork are limited for commonsense reasoning results in Table 3. Do the authors have an explanation for this observation ? Are correlated weights less important for language tasks ?

---

> ### Author Response · Authors · 2025-11-20
> **Response to Reviewer Tvs9 (part 1)**
>
> Thank you for your thoughtful feedback. We are pleased to address your concerns in detail as follows.
>
> **W1: algorithmic novelty is limited since interpolating between LoRA and DoRA is straightforward, and the hypernetworks have been widely used in PEFT**
>
> **Answer:** We appreciate the reviewer’s feedback and the opportunity to clarify the distinct contributions of our work. While interpolating between LoRA and DoRA may appear straightforward at a high level, our method is not merely a linear interpolation nor a direct application of existing hypernetwork-based PEFT approaches such as [1]. Instead, DoRAN introduces two complementary components - (i) the stabilization term $\tau$ and (ii) a shared hypernetwork - that together address key limitations in DoRA and result in demonstrably improved sample efficiency and training stability.
>
> First, although prior works (including [1]) employ hypernetworks for weight generation, our use of a *shared hypernetwork between the query and value projection matrices* is fundamentally different. This design choice is theoretically motivated. In particular, as shown in Section 4.2, this shared structure substantially enhances the sample efficiency from an exponential rate to a polynomial rate, and this theoretical claim is consistently supported in the empirical results.
>
> Second, the stabilization term $\tau$ goes well beyond a straightforward interpolation between LoRA-like scaling and DoRA updates. As motivated by our sample-efficiency theoretical analysis and the gradient analysis, $\tau$ plays a crucial role in ensuring well-behaved optimization dynamics: it prevents denominator singularities and allows the low-rank matrices to co-adapt both their direction and norm by retaining the parallel component of the gradient, which was otherwise be discarded in DoRA. This mechanism is essential for stabilizing DoRA training and is not present in LoRA, DoRA, or hypernetwork-based PEFT.
>
> Overall, the novelty of DoRAN lies not in isolated components but in the joint design of $\tau$ and the shared hypernetwork, which target complementary limitations and together produce a stable, parameter-efficient, and theoretically grounded PEFT method. We have clarified this distinction and expanded the discussion in the revised manuscript.
>
> **W2: lack comparisons with SoTA PEFT methods such as PiSSA and AdaLoRA:**
>
> **Answer:** We appreciate the reviewer’s suggestion. In response, we have included additional comparisons with PiSSA [2] in the revised manuscript (Table 1 for VTAB-1K and Table 2 for FGVC) and the table below (for FGVC dataset). We agree with the reviewer that DoRA can, in principle, be applied on top of PiSSA, and the same holds for DoRAN. Specifically, without altering the hypernetwork architecture, DoRAN can also be integrated with PiSSA by making minor adjustments to the initialization of the hypernetwork parameters. Recall that PiSSA begins by computing the SVD decomposition $W_0 = USV^\top$ and uses the following forward pass:
>
> $$W' = (W_{res}+AB)x$$
>
> where $A, B$ are initialized with
>
> $$A = U_{[:, :r]}\sqrt{S_{[:r, :r]}}, B = \sqrt{S_{[:r, :r]}}V^\top_{[:, :r]},$$
>
> and the frozen residual is given by $W_{res} = U_{[:, r:]}S_{[r:, r:]}V^\top_{[:, r:]}$. When applying DoRAN on top of PiSSA, we may initialize $W_{res}$ the same as PiSSA. For the adaptation matrices, consider the auxiliary networks implementation:
>
> $$B = W_2^B\sigma(W_1^B B'), A = W_2^A\sigma(W_1^A A')$$
>
> Theoretically, there are multiple possible ways to initialize the auxiliary networks based on the idea of PiSSA. For example, a simple choice is to set  $A' = B' = W_1^A = W_2^A = I$, where $I$ denotes the identity matrix, and set
>
> $$W_2^A = c U_{[:, :r]}\sqrt{S_{[:r, :r]}}, W_2^B = c \sqrt{S_{[:r, :r]}}V^\top_{[:, :r]},$$
>
> where $c = \frac{1}{\sigma(1)}$ is a normalizing constant ensuring that the initial weight $W'$ matches the pretrained weight $W_0$. The reviewer’s suggestion indeed highlights an interesting direction, which we consider promising for future exploration.
>
> | Method   | #Params (%) | CUB  | NAB  | Flowers | Dogs | Cars | Avg   |
> |----------|-------------|------|------|---------|------|------|-------|
> | PiSSA    | 0.55        | 87.1 | 80.5 | 99.1   | 84.8| 82.3 |  86.7    |
> | DoRAN    | 0.66        | 88.5 | 85.3 | 99.2    | 90.8 | 83.7 | 89.5  |

---

> ### Author Response · Authors · 2025-11-20
> **Response to Reviewer Tvs9 (part 2)**
>
> **W3: Lack discussion on the training time, memory, and FLOPs.**
>
> **Answer:** We thank the reviewer for the valuable feedback. In the current version, we report the ratio of learnable parameters to enable a fair comparison across PEFT methods. For completeness, we additionally provide a comparison of FLOPs and runtimes for low-rank–based approaches, including LoRA, DoRA, and DoRAN, in the table below. During
> training, although DoRAN introduces additional hypernetwork modules that increase FLOPs, its runtimes and the ratio of learnable parameters remain nearly identical to those of DoRA because all the hypernetworks are relatively small so these computations can be done efficiently. At the same time, DoRAN achieves substantial performance gains (e.g., +2.9% over LoRA and +2.06% over DoRA on LLaMA-7B). At inference time, all low-rank matrices are merged, so LoRA, DoRA, and DoRAN have identical FLOPs and latency, making DoRAN a practical and efficient solution for PEFT. Based on your suggestion, we have included this table in the revision for completeness of the experiments.
>
> |   Model   | Method | #Params(%) | GFLOPs | Runtimes | Performance |
> |:---------:|:------:|:----------:|:------:|:--------:|:-----------:|
> |  LLaMA-7B |  LoRA  |    0.25    |  6.63  |   4.17   |    74.09    |
> |           |  DoRA  |    0.25    |  25.42 |   5.22   |    74.94    |
> |           |  DoRAN  |    0.26    |  60.05 |   5.25   |    77.01    |
> | LLaMA-13B |  LoRA  |     0.2    |  12.88 |   7.25   |    80.18    |
> |           |  DoRA  |     0.2    |  49.58 |   7.37   |    80.05    |
> |           |  DoRAN |    0.21    | 117.21 |   7.48   |    80.76    |
>
> **Q1: The gains due to the hypernetwork on the commonsense reasoning task are limited compared to the gains for vision tasks.**
>
> **Answer:** Although the improvement of DoRAN over vanilla DoRA on commonsense reasoning tasks appears smaller than the gains observed in vision experiments, the +2.3% average increase is in fact substantial within this evaluation setting. As shown in Table 3, the performance differences among strong PEFT baselines are typically only within a narrow 1–3% band, making a 2.3% improvement both meaningful and competitive. Notably, *DoRAN is the only parameter-efficient method that surpasses full fine-tuning (FFT)* on these benchmarks, further underscoring its effectiveness.
>
> **References:**
>
> [1] Parameter-efficient multi-task fine-tuning for transformers via shared hypernetworks. ACL, 2021.
>
> [2] PiSSA: Principal Singular Values and Singular Vectors Adaptation of Large Language Models. NeurIPS, 2024.

---

### Official Review · Reviewer_yTBF · 2025-10-30

**Soundness:** 2
**Presentation:** 3
**Contribution:** 2
**Rating:** 4
**Confidence:** 4

**Summary:**

This paper proposes DoRAN (Weight-Decomposed Low-Rank Adaptation with Noise Injection and Auxiliary Networks), a new variant of DoRA for parameter-efficient fine-tuning (PEFT) of large models. DoRAN aims to improve both stability and sample efficiency in low-rank adaptation. It introduces two main innovations: (1) a learnable noise injection term in the normalization denominator to stabilize gradients and preserve magnitude information adaptively; and (2) auxiliary hyper-networks that dynamically generate low-rank matrices across layers, enabling structural consistency and cross-layer parameter coupling. Theoretical analysis connects DoRAN to a Mixture-of-Experts framework and claims improved sample complexity from exponential to polynomial order. Experiments on both vision and language benchmarks show consistent improvements over LoRA and DoRA with minimal parameter overhead. Overall, the paper combines noise-based regularization and hypernetwork-driven generation into a unified PEFT framework targeting more stable and efficient fine-tuning.

**Strengths:**

- The authors provide a rigorous convergence analysis by reinterpreting DoRAN through the Mixture-of-Experts lens. Theoretical results establish improved sample complexity and offer insight into how shared parameterization and noise injection contribute to stability—an uncommon depth of analysis for PEFT papers.


- Experiments on diverse benchmarks (VTAB-1K, FGVC, LLaMA-7B/13B) demonstrate consistent improvements over prior PEFT baselines with minimal added parameters. Results support the claim that DoRAN improves both stability and sample efficiency, and the empirical section aligns well with the theoretical motivation.


- The paper is well-organized and clear in exposition. Mathematical derivations are transparent, figures are well-labeled, and the narrative effectively connects theoretical and empirical parts. Compared with prior DoRA-related works, this manuscript shows strong technical writing and coherent argumentation.

**Weaknesses:**

(1) **Lack of ablation on the noise injection mechanism.** The paper attributes DoRAN’s performance gain partly to the learnable noise term in the normalization denominator, but there is no ablation isolating its contribution. It remains unclear whether the improvement stems from the proposed stabilization or simply from regularization effects. A variant without the noise term or with fixed noise magnitude would clarify its real impact.

(2) **Limited analysis of the auxiliary hyper-network.** Although the auxiliary network is claimed to enable dynamic low-rank generation and cross-layer consistency, the paper does not show how these properties manifest in practice. No visualization or diagnostic (e.g., rank diversity, parameter correlation across layers) is provided. This makes it difficult to assess whether the hypernetwork truly improves representation capacity or merely adds parameter redundancy.

**Questions:**

- Could the authors provide an ablation study isolating the contribution of the noise term in the normalization denominator? For instance, comparing (1) no noise, (2) fixed Gaussian noise, and (3) learnable noise injection would clarify whether the improvement comes from true stabilization or from generic regularization effects. Including convergence or gradient variance plots would further strengthen this claim.

- It would be helpful to better understand how the auxiliary network influences cross-layer representations. Can the authors visualize or quantify how the generated low-rank matrices differ across layers, e.g., via rank spectra or inter-layer cosine similarit? Such analysis would make the "cross-layer consistency" argument more convincing and clarify whether the hyper-network provides structural benefits or just extra capacity.

---

> ### Author Response · Authors · 2025-11-20
> **Response to Reviewer yTBF**
>
> Thank you for your thoughtful feedback. We are pleased to address your concerns in detail as follows. We hope that our responses address your concerns, and finally convince you to raise your score.
>
> **W1, Q1: Ablations on DoRAN with (1) no noise, (2) fixed Gaussian noise, and (3) learnable noise injection**
>
> **Answer:** We would like to clarify that we have also evaluated DoRA–$\tau$ (i.e., DoRA with a learnable noise $\tau$ and not using auxilliary networks), which corresponds to category (3) mentioned in the reviewer’s first question. Following the reviewer’s suggestions, we additionally conducted two ablations on FGVC datasets:
>
> - DoRAN without $\tau$, using only the hypernetwork component (i), and
>
> - DoRAN with fixed Gaussian noise instead of learnable $\tau$ and use the hypernetwork component (ii).
>
> We appreciate these suggestions and have incorporated the additional ablation results into the *appendix of the revised version (Table 5)*, which is also the table below. As the results show, although DoRAN with Gaussian noise and without noise offer slight improvements, they still fall short of a learnable
> $\tau$, which provides an extra degree of freedom to balance direction learning and norm control (as discussed in the Gradient Analysis section). Furthermore, the inclusion of hypernetworks yields a substantial performance gain. Together, these resullts underscore the effectiveness and complementary contributions of both components.
>
> | Method                 | CUB | NAB | Flowers | Dogs | Cars | Avg  |
> |------------------------|-----|-----|---------|------|------|------|
> | Vanilla DoRA           | 87.3| 80  | 99.1    | 87.6 | 81.9 | 87.2|
> | DoRA–$\tau$          | 88.3| 83.4| 99.2    | 90.2 | 82.9 | 88.8 |
> | DoRAN w/o $\tau$ (i)          | 87.9| 85.4| 99.2    | 90.5 | 83.5 | 89.3 |
> | DoRAN + Gaussian $\tau$ (ii)       | 88.3| 85.2| 99.0    | 90.5 | 83.3 | 89.3 |
> | **DoRAN**              | **88.5**| **85.3**| **99.2**| **90.8**| **83.7**| **89.5** |
>
> **W2, Q2: Limited analysis on whether the cross-layer hypernetworks truly improves representation capacity**
>
> **Answer:** First, we recall the architechture of the auxiliary networks. Our auxiliary network is only partially shared: the first layer is shared between the query and value projection matrices, while the second layers are distinctive. As shown in the Section 4.2 in our theoretical analysis, we showed that the sharing mechanism of the hypernetwork between the query and value projections substantially reduces the model sample complexity from an exponential order to a polynomial order. We also clarify that we do not perform any cross-layer sharing: each layer maintains its own independent hypernetwork, and therefore generates its own independent low-rank matrices. We sincerely apologize for the earlier use of the term “cross-layers,” which may have caused misunderstanding regarding sharing across layers. We have corrected this wording in the revised manuscript.
>
> **References:**
>
> [1] On the Relationship between Self-Attention and Convolutional Layers. ICLR, 2020.
>
> [2] How Do Vision Transformers Work?. ICLR, 2022.
>
> [3] Are Sixteen Heads Really Better Than One?. NeurIPS, 2019.

---

> > ### Author Response · Authors · 2025-11-25
> > **Looking forward to your response**
> >
> > Dear Reviewer yTBF,
> >
> > We would like to thank you again for your thorough review, and we hope that our response successfully addresses your concerns regarding our manuscript. However, since the discussion period is expected to end soon, please feel free to let us know if you have any further concerns. We would be willing to address any additional concerns from you. Otherwise, we hope that you will consider increasing your final rating.
> >
> > Thank you so much for your time and effort spent on the paper, we really appreciate it!
> >
> > Best regards,
> >
> > The Authors

---

### Author Response · Authors · 2025-11-20
**General Response**

Dear Area Chairs and Reviewers,

We sincerely thank you for your constructive feedback and thoughtful reviews, which have significantly strengthened our manuscript. We also greatly appreciate the reviewers’ positive recognition of our work, including the following endorsement:

**Contribution:**

- The authors provide a **rigorous** convergence analysis by reinterpreting DoRAN through the Mixture-of-Experts lens. Theoretical results establish improved sample complexity and offer insight into how shared parameterization and noise injection contribute to stability—**an uncommon depth of analysis for PEFT papers** (Reviewers yTBF, vatj).

- The idea of interpolating between LoRA and DoRA's weight approximation is **simple yet effective** (Reviewer Tvs9).

- Provide theoretical analysis on gradients to show that the learnable noise can **ensure bounded gradients** (Reviewer S9Kw).

- The paper provides a **clear and well-motivated improvement** over DoRA. The proposed hypernetwork design helps **reduce redundancy** and may **promote diversity** in learned representations (Reviewer vatj).

**Soundness:**

- Experiments on **diverse benchmarks** demonstrate **consistent improvements** over prior PEFT baselines with minimal added parameters. Results support the claim that **DoRAN improves both stability and sample efficiency** (Reviewer yTBF).

- The authors theoretically support DoRAN's strategy, which further supports experimental findings (Reviewer Tvs9).

- Provide theoretical analysis on gradients to show that the learnable noise $\tau$ can ensure bounded gradients (Reviewer S9Kw).

- The paper includes theoretical analysis that **offers useful insights** into the stability and convergence behavior of the proposed method, providing a clear theoretical perspective in addition to empirical evidence (Reviewer vatj).

**Presentation:**

- The paper is **well-organized and clear** in exposition. Mathematical derivations are **transparent**, figures are well-labeled, and the narrative **effectively connects theoretical and empirical parts**. Compared with prior DoRA-related works, this manuscript shows **strong technical writing** and **coherent argumentation** (Reviewer yTBF).

We deeply appreciate the reviewers’ positive feedback and thoughtful suggestions. We will address each reviewer’s specific concerns in our detailed responses below and describe the corresponding revisions made based on their valuable recommendations.

Sincerely,

Authors

---

### Author Response · Authors · 2025-12-03
**Summary After Author–Reviewer Discussion**

Dear Area Chairs and Senior Area Chairs,

We hope you are doing well, and we would like to express our sincere appreciation for your efforts under these difficult circumstances. We recognize that reviewing the reverted discussion threads involves extra work, and we are truly grateful for your dedication to preserving a fair and accurate evaluation process.

To facilitate your assessment, we include below a brief summary of the key points raised in the reviews and discussions of our paper before the leakage incident. In particular, we outline the main components of our rebuttal as they stood prior to November 27, 2025, to support your evaluation and final decision:

## **1. Motivation and Contribution**

**I. Motivation.**

(i) We propose DoRAN, a PEFT method built on top of DoRA that addresses its training instability. Instead of normalizing the adapted weight exactly as in DoRA, we introduce a learnable noise term that smooths the normalization and preserves useful gradient direction information, resulting in more stable optimization.

(ii) To improve sample efficiency, we generate the low-rank adapter matrices using shared hypernetworks. These hypernetworks are shared across the query and value projections, enabling substantial parameter sharing with minimal loss of expressivity.

**Feedback from reviewers.**

(i) Although the idea of interpolating between LoRA and DoRA's weight approximation is simple, it proves effective experimentally. Using hyper-networks to predict the weight update is also a straightforward strategy to induce correlations across layers. (Reviewer Tvs9)

(ii) The paper provides a clear and well-motivated improvement over DoRA by introducing a learnable noise term ($\tau$) that stabilizes normalization and enables smooth interpolation between DoRA and LoRA behaviors. (Reviewer vatj)

(iii) The proposed hypernetwork design allows cross-layer parameter sharing, which helps reduce redundancy to some extent and may promotes diversity in learned representations. (Reviewer vatj)



**II. Contributions.**

Our main contributions are two-fold:

(i) We provide a theoretical analysis showing that our design improves sample efficiency, tightening the convergence rate from exponential order to polynomial order.

(ii) We demonstrate across diverse experimental settings that DoRAN achieves state-of-the-art parameter efficiency, consistently outperforming LoRA, DoRA, and other PEFT baselines on both vision and language tasks.


**Feedback from reviewers.**

(i) The authors provide a rigorous convergence analysis by reinterpreting DoRAN through the Mixture-of-Experts lens. Theoretical results establish improved sample complexity and offer insight into how shared parameterization and noise injection contribute to stability—an uncommon depth of analysis for PEFT papers. (Reviewer yTBF)

(ii) Provide theoretic analysis on gradients to show that the learnable noise $\tau$ can ensure bounded gradients. (Reviewer S9Kw)

(iii) The paper includes theoretical analysis that offers useful insights into the stability and convergence behavior of the proposed method, providing a clear theoretical perspective in addition to empirical evidence. (Reviewer vatj)


## **2. Theoretical Results**

We first rigorously derive **a novel relation between DoRA and MoE**. Next, through this connection, we demonstrate that **our proposed method DoRAN is substantially more sample efficient than the standard DoRA**. In particular, the sample comlexity of estimating low-rank matrices in DoRAN is of a polynomial order on the sample size, while that DoRA admits a significantly higher sample complexity of an exponential order.

**Feedback from reviewers.**

Multiple reviewers (yTBF, vatj) agree that our theoretical results **offer useful insight** into how shared parameterization and noise injection contribute to stability—**an uncommon depth of analysis for PEFT papers**.

---

> ### Author Response · Authors · 2025-12-03
>
> ## **3. Empirical Results**
>
> We evaluate DoRAN on three benchmarks: two vision benchmarks (VTAB-1K and FGVC) and one language benchmark. Across all settings, DoRAN consistently outperforms LoRA, DoRA, and other PEFT methods. In particular:
>
> * **VTAB-1K:** DoRAN surpasses DoRA and LoRA on all domains, with gains over DoRA of **+1.1\%** on Natural, **+1.6\%** on Specialized, **+1.8\%** on Structured, and **+2.4\%** overall.
> * **FGVC:** DoRAN improves performance by **+2.3\%** over DoRA and **+4.7\%** over LoRA overall.
> * **Commonsense reasoning:** DoRAN outperforms DoRA by **+1.9\%** on LLaMA-7B and **+0.2\%** on LLaMA-13B.
>
> Moreover, DoRAN is markedly more sample-efficient than DoRA in the low-data regime: when only **1\%** of the training data is used, DoRAN exceeds DoRA by more than **20\%**, highlighting its superior sample efficiency over vanilla DoRA.
>
> **Feedback from reviewers.**
>
> (i) Experiments on diverse benchmarks (VTAB-1K, FGVC, LLaMA-7B/13B) demonstrate consistent improvements over prior PEFT baselines with minimal added parameters. Results support the claim that DoRAN improves both stability and sample efficiency, and the empirical section aligns well with the theoretical motivation. (Reviewer yTBF)
>
> (ii) The authors theoretically support DoRAN's strategy which further supports experimental findings. (Reviewer Tvs9)
>
> ## **4. Main concerns from reviewres.**
>
> **I. Reviewer yTBF: Score 4**
>
> **Reviewer concerns:** The reviewer requested: (i) ablation study and (ii) analysis on cross-layer hypernetworks
>
> **Our response:**
>
> (i) We clarified that our original evaluation of DoRA in the paper used learnable noise $\tau$ without auxiliary networks. In the rebuttal and revised manuscript, we further included two additional ablations—DoRAN without $\tau$ and DoRAN with fixed Gaussian noise. The results show that DoRAN outperforms all alternative designs on the FGVC dataset.
>
> (ii) We corrected wording issues that led to a misunderstanding regarding the use of hypernetworks in DoRAN. The reviewer assumed that a single hypernetwork was shared across all layers; we clarified that each layer employs its own dedicated hypernetwork, and we have revised the manuscript accordingly.
>
> **II. Reviewer Tvs9: Positive feedback (Score 6)**
>
> **Reviewer concerns:** The reviewer suggested: (i) clarify algorithm novelty more detail, (ii) compare to other methods, and (iii) provide detail of computational information
>
> **Our response:**
>
> (i) We clarified that DoRAN is not a simple interpolation of LoRA and DoRA. Its two key proposed components - the stabilization term $\tau$ and shared hypernetwork - address DoRA’s limitations. We also provided theoretical analysis and experiments to demonstrate their combined effect drives DoRAN’s improved performance.
>
> (ii) We compared our method with PiSSA, one method the reviewer proposed to compare, on vision tasks and show that DoRAN still **outperform this method** in all tasks. Moreover, we also show and propose a strategy that applying our designs in DoRAN on PiSSA, which we consider promising for future exploration.
>
> (iii) We reported learnable-parameter ratios in the original paper and, following the reviewer’s suggestion, we additionally compared FLOPs and runtimes for LoRA, DoRA, and DoRAN. Despite slightly higher training FLOPs, DoRAN maintains similar runtime and parameter ratio, achieves notable performance gains, and shares identical FLOPs and latency with other methods at inference.

---

> ### Author Response · Authors · 2025-12-03
>
> **III. Reviewer S9Kw: Score 4**
>
> **Reviewer concerns:** The reviewer concerned:
>
> (i) High computational cost of DoRA-like approaches
>
> (ii) The noise term is not well-motivated
>
> (iii) The design of parameter-sharing is not well-motivated
>
> (iv) Incorrect claim about LoRA equivalent
>
> (v) Insufficient validation for the effectiveness of DoRAN
>
> **Our response:**
>
> (i) We admit that DoRA’s normalization adds extra compute cost, and DoRAN inherits this, but our method targets different problems of DoRA, which are instability and sample inefficiency. DoRAN’s stabilization term and shared hypernetwork directly address these issues. Empirically, DoRAN improves performance and sometimes surpasses full fine-tuning, even though computational efficiency remains a separate problem for future work.
>
> (ii) We clarify that the noise term $\tau$ serves different purposes, not only stabilizing small norms. $\tau$ improves sample efficiency (it avoids interaction at Eq.(5)), prevents singularities, and retains gradient directionality for stable co-adaptation (section 3.2).
>
> (iii) We clarify that the matrices $A$ and $B$ each have their own hypernetwork between key and value matrices and no cross-layer sharing is used (section 3.1). Only the input and first layer are shared between query/value projection hypernetworks. This design is theoretically motivated (see Section 4.2), and our analysis shows it improves sample efficiency from exponential to polynomial.
>
> (iv) We thank the reviewer for highlighting this confusion. “LoRA-like” referred only to the behavior under very large $\tau$, not to preserving $W_0$ or LoRA’s regularization. We removed the term to avoid confusion and clarified that $\tau$ controls the balance between norm control and directional updates. Empirically, $\tau$ stays below 2 in all layers (as shown in Fig. 5), confirming theoretical large-behavior raised in the comment does not occur in our empirical settings.
>
> (v) Our experiments have not yet included 70B-scale models due to compute limits, but we thoroughly evaluated LLaMA-7B/13B and LLaMA3-8B for fair and controlled comparisons to DoRA. We show generality beyond LLaMA by adding ViT experiments (FGVC, VTAB-1K) and expanded comparisons, including PiSSA and additional LLaMA3-8B results. These results consistently show DoRAN’s strong performance, and we acknowledge larger-scale evaluations as future work.
>
> **IV. Reviewer vatj: Positive feedback (Score 6)**
>
> **Reviewer concerns:** The reviewer suggested: (i) provide clearer attribution on results, (ii) compare methods on more competitive LLMs, (iii) provide detail of computational information, (iv) provide ablation study when DoRAN is applied to all projection matrices, (v) Analyse $\tau$ values, and (vi) add MLAE methods to experiments
>
> **Our response:**
>
> (i) We additionally provided complete details of all hyperparameters used to train or reproduce the methods on both vision tasks and commonsense reasoning tasks, ensuring that all experiments constitute a fair and consistent comparison.
>
> (ii) We incorporated the reviewer’s suggestion to include experiments on LLaMA3-8B in both the rebuttal and the revised manuscript, and we demonstrate that DoRAN consistently outperforms DoRA in this setting.
>
> (iii) We reported learnable-parameter ratios in the original paper and, following the reviewer’s suggestion, we additionally compared FLOPs and runtimes for LoRA, DoRA, and DoRAN. Despite slightly higher training FLOPs, DoRAN maintains similar runtime and parameter ratio, achieves notable performance gains, and shares identical FLOPs and latency with other methods at inference.
>
> (iv) We provided experiments where DoRAN is applied to all projection layers (QUERY, KEY, VALUE, UP, and DOWN) in rebutal and our revised manuscript and demonstrate that DoRAN still outperform DoRA in commonsense reasoning tasks.
>
> (v) We provided plots of $\tau$ across 12 ViT layers (Figures 4–5, appendix). Early in training, $\tau$ is uniform across layers; by the final epoch, early layers show higher variance while later layers converge, and the mean $\tau$ increases steadily. This shows $\tau$ adapts meaningfully across depth, supporting DoRAN’s structural benefits.
>
> (vi) We incorporated MLAE experiments on all vision and commonsense reasoning tasks in the rebuttal and revised manuscript. DoRAN outperforms MLAE on most tasks (VTAB-1K and commonsense reasoning), with only a minimal loss on FGVC (0.5\%). We provide detailed hyperparameters and implementation details for fair comparison, and also outline how our design can be applied to MLAE, leaving a full investigation of its benefits for future work.
>
> After discussion, the reviewer confirmed that all questions were addressed. They also highlighted that the idea of combining MLAE with DoRAN is highly insightful and chose to maintain a positive evaluation of the paper.

---

### Meta-Review · Area_Chair_ZEo2 · 2026-01-06

**Summary:**

This paper proposes a new variant of DoRA by injecting noise and replacing static low-rank matrices with auxiliary networks. The reviewer evaluations are borderline. After carefully reading the paper and the reviews, the AC agrees with a central concern raised by Reviewer S9Kw (detailed below). This concern remained unresolved after the rebuttal, and the reviewer was not convinced by the authors’ response.

While this issue may partially stem from a misunderstanding, it nevertheless indicates that the current manuscript does not present its core idea with sufficient clarity or justification. As a result, the AC recommends REJECTION.

**Reviewer Concerns:**

The paper received four reviews, all with borderline scores. Positive feedback highlights the clear motivation and the theoretical discussion of noise injection for numerical stability.

However, several weaknesses were consistently identified: limited novelty (Reviewer Tvs9); computational efficiency concerns (Reviewers Tvs9, vatj), less convincing experimental validation, including missing strong baselines (Reviewers Tvs9, S9Kw), missing comparisons (Reviewer vatj), and missing ablation studies (Reviewer yTBF). Part of the weakness on experiments have been addressed during the rebuttal.

A particularly critical concern was raised by Reviewer S9Kw, which the AC also supports. In essence, injecting a small amount of noise to improve the numerical stability of division operations is a neat and lightweight idea, but by itself it is insufficient for a top-tier venue such as ICLR. Consequently, the core contribution effectively becomes learning two auxiliary networks 𝑔(𝐴) and 𝑔(𝐵) to represent the original matrices A and B. While simple, this introduces additional learnable components and complexity. The authors appear to be aware of this issue and attempt to mitigate it by sharing parameters for A; however, as the reviewer noted, this design choice is somewhat unintuitive and was not convincingly justified. When the auxiliary module becomes the primary component, it raises a fundamental question about whether the method truly improves or meaningfully modifies LoRA/DoRA.

Given these unresolved concerns, the AC concludes that although the paper shows some potential, it is not sufficiently mature in its current form.

**Reviewer Scores:**

The initial reviewer scores were 4/4/6/6. During the rebuttal period, Reviewer S9Kw (score 4) actively participated in the discussion but maintained their score. Another reviewer with score 6 also chose to keep their original assessment.

---

### Decision · Program_Chairs · 2026-01-26

Reject